# Integrase anchors viral RNA to the HIV-1 capsid interior

Matthew R. Singer[1], Zhen Li[2], Juan S. Rey[3], Joshua Hope[4], Florian Chenavier[1], Nicola J. Cook[1], Emma Punch[1], Jamie Smith[1], Zhiyu Zhou[1], Sarah Maslen[5], Laura Masino[6], Andrea Nans[6], Mark Skehel[5], Ian A. Taylor[7], Giulia Zanetti[8,9], Peijun Zhang[4,10], Juan R. Perilla[3✉], Alan N. Engelman[2✉] & Peter Cherepanov[1,11✉]

HIV-1 integrase (IN) promotes encapsulation of viral genomic RNA into mature viral cores, and this function is a target for ongoing antiretroviral drug development efforts[1–3]. Here we determined the cryogenic electron microscopy (cryo-EM) structure of a primate lentiviral IN in a complex with RNA, revealing a linear filament made of IN octamer repeat units, each comprising a pair of asymmetric homotetramers. The assembly is stabilized through IN–RNA interactions involving mainly the IN C-terminal domains and RNA backbone. The spacing and orientation of the IN filament repeat units closely matched those of consecutive capsid (CA) hexamers within the mature CA lattice. Using cryo-EM images of native purified HIV-1 cores, we refined the structure of the IN filament as it propagates along the luminal side of the CA lattice. Each IN tetramer within the filament nestled in a CA hexamer, engaging closely with the major homology regions. Substitutions of residues involved in IN–CA contacts yielded eccentric virions with RNA nucleoids located outside of the cores. Collectively, our results establish the structural basis for the HIV-1 IN–RNA interaction and reveal that IN forms an RNA-binding module on the luminal side of the mature CA lattice.

In addition to its canonical role of orchestrating the insertion of viral DNA into the host genome, HIV-1 integrase (IN) has an essential function in viral morphogenesis[1,4,5] (reviewed previously[6]). The HIV-1 core contains viral ribonucleoproteins (RNPs; predominantly nucleocapsid (NC) protein bound to two copies of genomic RNA) encased within a closed lattice of capsid (CA) protein. Numerous examples of amino acid substitutions in IN, or allosteric IN inhibitors (ALLINIs), preclude the incorporation of viral RNPs into cores during maturation, yielding an eccentric virion morphology with RNPs situated outside of comparatively electron-lucent or 'empty' CA shells[1–4,7]. The failure to encapsulate RNPs results in a pronounced defect of reverse transcription, the signature class II phenotype associated with many HIV-1 IN mutant viruses[5,8]. Although HIV-1 IN has been shown to interact with viral genomic RNA in virio[1,2], how the viral protein binds to and retains viral RNA inside mature cores remains unclear.

## Formation of ordered IN–RNA co-polymers

We tested recombinant HIV-1 IN protein for interaction with a range of RNA constructs using biolayer interferometry (BLI). In agreement with published observations[1], HIV-1 IN bound to a 57-mer oligonucleotide spanning the HIV-1 transactivation response (RNA^TAR) element that

folds into a long hairpin structure[9], as well as to a range of shorter RNAs, including those composed solely of GA or UC dinucleotides that do not adopt stable tertiary structures (Extended Data Fig. 1). Elevated salt concentrations inhibit the IN–RNA interaction[1], and HIV-1 IN quickly aggregates in vitro in buffers containing <1 M NaCl (Extended Data Fig. 2a). To overcome this technical hurdle, we screened a panel of primate lentiviral INs, identifying IN from simian immunodeficiency virus (SIV) from talapoin monkeys (SIVtal) as having favourable biochemical properties. Sharing around 52% amino acid sequence identity with HIV-1 IN, SIVtal IN displayed improved solubility (Extended Data Fig. 2a,b). Similar to HIV-1 IN[2,10–12] and maedi-visna virus (MVV) IN[13], SIVtal IN formed stable tetramers under a wide range of concentrations (Extended Data Fig. 2a). Finally, SIVtal IN showed robust interaction with RNA oligonucleotides in the BLI assay (Extended Data Fig. 1). To evaluate the effects of RNA binding on the IN structure, we used hydrogen–deuterium exchange coupled with mass spectrometry (HDX–MS). We incubated HIV-1 and SIVtal INs in the absence or presence of RNA^TAR and transferred the mixtures into deuterated buffer. RNA significantly diminished isotope exchange over extended portions of the IN amino acid sequences, including regions within the N-terminal domain (NTD), catalytic core domain (CCD) and C-terminal domain (CTD) (Extended Data Fig. 2c).

[1]Chromatin Structure & Mobile DNA Laboratory, The Francis Crick Institute, London, UK. [2]Department of Cancer Immunology and Virology, Dana-Farber Cancer Institute, Boston, MA, USA. [3]Department of Chemistry and Biochemistry, University of Delaware, Newark, DE, USA. [4]Division of Structural Biology, Nuffield Department of Medicine, The University of Oxford, Oxford, UK. [5]Proteomics Science Technology Platform, The Francis Crick Institute, London, UK. [6]Structural Biology Technology Platform, The Francis Crick Institute, London, UK. [7]Macromolecular Structure Laboratory, The Francis Crick Institute, London, UK. [8]Membrane Architecture Laboratory, The Francis Crick Institute, London, UK. [9]Institute of Structural and Molecular Biology, Birkbeck College, London, UK. [10]Diamond Light Source, Harwell Science and Innovation Campus, Didcot, UK. [11]Department of Infectious Disease, Imperial College London, London, UK. ✉e-mail: JPerilla@udel.edu; Alan_Engelman@dfci.harvard.edu; Peter.Cherepanov@crick.ac.uk

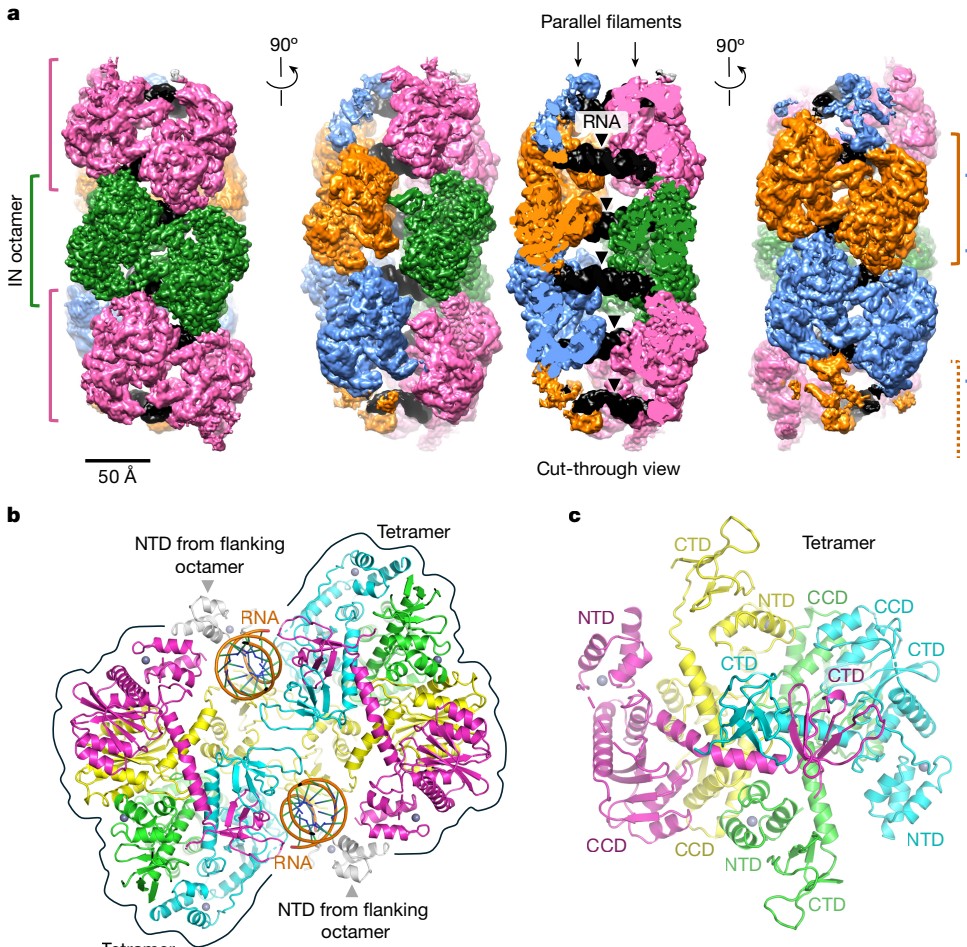

**a**

Parallel filaments

RNA

Cut-through view

IN octamer

50 Å

**b**

NTD from flanking octamer

Tetramer

RNA

RNA

NTD from flanking octamer

Tetramer

**c**

Tetramer

CTD

NTD

CCD

CTD

CCD

CTD

NTD

NTD

CCD

CCD

NTD

CTD

**Fig. 1 | The structure of the SIVtal IN–RNA^TAR complex. a**, Cryo-EM reconstruction of the SIVtal IN–RNA complex shown in three orthogonal views. The structure comprises two IN filaments bridged by RNA duplexes (black). Octameric IN repeat units are distinguished by alternating colours and delineated with square brackets. **b**, Cartoon representation of a single octameric IN repeat unit bound to two RNA molecules. IN subunits are coloured green, cyan, magenta and yellow to indicate structurally equivalent chains within opposing tetramers. A-form duplex RNA chains are shown in the cartoon with an orange phosphodiester backbone. The NTDs from flanking octameric units are included in the model and shown in grey. Grey spheres represent Zn^{2+} ions coordinated within the NTDs. **c**, View of a single IN tetramer along its approximate two-fold symmetry axis. Canonical IN domains—NTD, CCD and CTD—are labelled. The two-fold symmetry is disrupted by the lateral displacement of the CTD dimer (formed by IN subunits in cyan and magenta). The images of cryo-EM maps (**a**) and structural models (**b**,**c**) were created using UCSF Chimera and PyMOL, respectively.

Imaging SIVtal IN vitrified in the presence of RNA^TAR on a transmission electron microscope revealed the formation of elongated polymers (Extended Data Fig. 3a). Processing single-particle images enabled us to compute a three-dimensional (3D) reconstruction of the IN–RNA co-polymer at an overall resolution of 3.7 Å (Fig. 1a, Extended Data Fig. 3b and Extended Data Table 1). The averaged structure comprises two identical linear filaments running parallel to one another and joined by RNA duplexes shared between them (Fig. 1a and Supplementary Video 1). Each sister filament represents a chain of IN octamers formed around a pair of RNA^TAR chains. This assembly depends on the ability of RNA^TAR to adopt an extended duplex, multiple copies of which rigidly tether the parallel IN filaments. Concordantly, SIVtal IN formed single filaments (chains of octamers) in the presence of single-stranded oligo-GA ribooligonucleotides, while ordered IN structures did not form in the absence of RNA (Extended Data Fig. 3a). Although fortuitous formation of the RNA^TAR-tethered double-IN filament enabled high-resolution structure determination, we would not expect such extended long-range arrangements to occur with the 9 kb viral RNA genome.

Masked refinement centred on a filament repeat unit resulted in 3D reconstruction of an IN octamer associated with two copies of RNA^TAR at an overall resolution of 3.3 Å. The local resolution ranged from 3 Å throughout the bulk of the protein regions to about 4 Å for the RNA component (Extended Data Fig. 3c), which was defined less well presumably due to averaging of the two opposing binding orientations. The IN octamer features two-fold symmetry and comprises two identical homotetramers sharing a pair of RNA^TAR chains (Fig. 1b). The tetramers feature the dimer-of-dimers architecture as has been observed in isolated lentiviral IN constructs[10,14] and intasome assemblies[15,16], yet deviates considerably from two-fold symmetry (Fig. 1c). The tetramers engage RNA chains solely through their CTDs. Within each tetramer, a pair of CTDs form the canonical clamshell dimer[17], each binding across the major groove of the RNA duplex, clasping the minor groove in between (Fig. 2a). This binding mode effectively gauges the geometric parameters of the A-form duplex and may therefore enable the viral protein to recognize RNA duplex structures. Each of the two paired CTDs use a triad of positively charged residues—Arg228, Arg264 and Lys265 (corresponding to HIV-1 IN Arg228, Arg263 and Lys264, respectively)—to interact with the phosphodiester backbone. One of the remaining CTDs in each SIVtal IN tetramer interacts with the RNA backbone engaged by the opposing tetramer through the Lys244, Arg263 and Lys270 side chains (equivalent to HIV-1 IN Lys244, Arg262 and Arg269, respectively), effectively bridging the two halves of the octamer (Fig. 2b). The stacking of octameric filament repeat units is

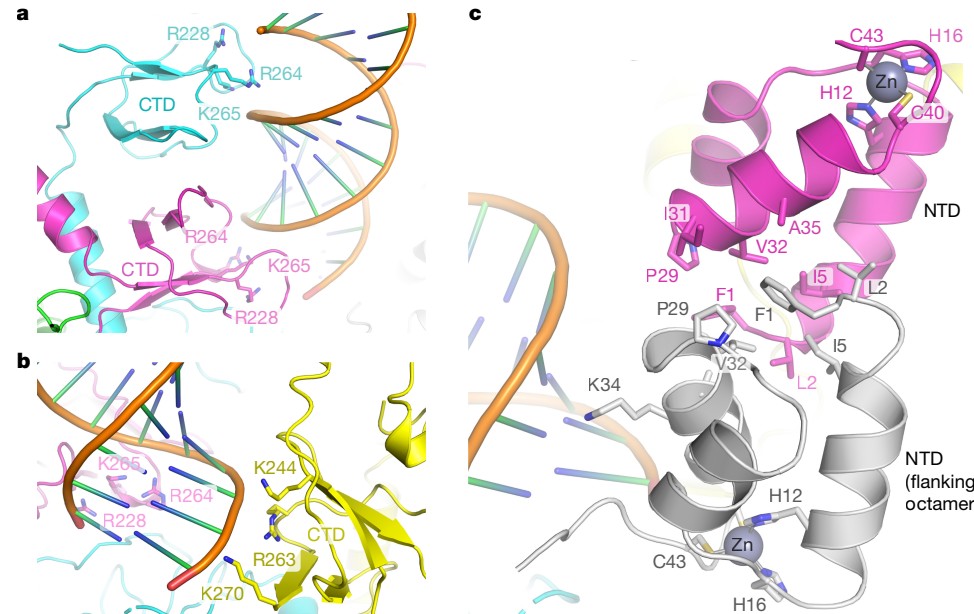

**Fig. 2 | IN–RNA and IN–IN interfaces within the SIVtal IN filament.**
**a**, Interaction of dimerized CTDs with an RNA duplex within the IN tetramer.
**b**, Interactions involving CTD from one IN tetramer with an RNA duplex engaged by the opposing tetramer. **c**, Interactions involving neighbouring IN octamers. Protein and RNA chains are shown as cartoons and are coloured as in Fig. 1b. The side chains of residues discussed in the main text are shown as sticks and indicated. His12, His16, Cys40 and Cys43 comprise the zinc-coordinating motif conserved among retroviral INs. The diagrams were created using PyMOL.

mediated by IN NTD–NTD interactions (Fig. 2c); here, residues Phe1, Val2, Ile5, Pro29 and Val32 (corresponding to HIV-1 IN Phe1, Leu2, Ile5, Pro29 and Val32, respectively) from neighbouring subunits form a compact hydrophobic interface. Moreover, the side chain of NTD residue Lys34 (Lys34 in HIV-1 IN) from one octamer interacts with the backbone of the RNA chain belonging to the neighbouring octamer (Fig. 2c).

The structure of the SIVtal IN–RNA[TAR] filament explains many previous observations made for HIV-1, including the importance of IN tetramerization for RNA binding[2], the protection of Lys264 from modification by *N*-hydroxysuccinimide in the presence of RNA[1], and the RNA-binding defects of virion-incorporated K34A, R262A/R263A and R269A/K273A IN tetramers[1,2]. Furthermore, in agreement with their involvement in RNA binding, HIV-1 IN mutant viruses K34A, R228A, K244A, R262A/R263A, R262A/K264A and R269A/K273A, as well as the disruption of the NTD structure by H12N, were shown to elicit the class II phenotype[1,2,18–20], supporting the involvement of the IN filament assembly in RNA retention.

## IN filaments inside native HIV-1 cores

Over 95% of the mature HIV-1 core shell is composed of CA hexamers that form a regular lattice with a centre-to-centre distance between neighbouring repeat units of around 91 Å (refs. 21–23). Notably, this parameter matches precisely the distance between consecutive IN octamer repeats within the SIVtal IN–RNA[TAR] structure. Furthermore, the IN octamer repeat units are inclined relative to the main filament axis by around 120°, which allows a near-perfect overlap with the mature CA hexamer lattice (Extended Data Fig. 4a). This unexpected congruency of two structurally unrelated supramolecular assemblies suggested that they could have coevolved to associate during viral maturation. To test this hypothesis, we produced virus-like particles by co-transfection of HEK293T cells with a near-full-length HIV-1 proviral construct and a plasmid overexpressing IN in the form of a fusion with HIV-1 viral protein R (Vpr) and mNeonGreen (NG). The latter was included to increase the IN copy number by taking advantage of the ability of Vpr to direct fusion partners into viral particles[24]. For safety considerations, the proviral

construct carried a deletion of *tat*[25] and IN active-site mutations[26], the latter of which were also included in the Vpr-NG-IN construct. The wild-type (WT) and mutant versions of Vpr-NG-IN importantly boosted the infectivity of class I and class II HIV-1 IN mutant viruses, as previously shown for other Vpr-IN chimeras[20,27], while Vpr-NG-IN only marginally impaired WT HIV-1 infectivity (Extended Data Fig. 4b). Mature cores were isolated by ultracentrifugation of viral particles through a layer of mild detergent. Semi-quantitative western blotting revealed approximate doubling of viral core IN content through Vpr-NG-IN co-expression (Extended Data Fig. 4c).

We vitrified purified cores on cryogenic electron microscopy (cryo-EM) grids and imaged them on a transmission electron microscope (Extended Data Fig. 5a). Using single-particle image processing approaches, we isolated around 1 million particles, each representing a fragment of the mature CA lattice. 3D classification focused on the signal underlying the hexagonal lattice enabled us to select a subset containing a well-defined linear polymer consistent with repeating IN octamer units (Extended Data Fig. 5b–d). Further image processing resulted in a 3D reconstruction spanning 14 complete CA hexamers and 3 IN octamers (47 × 47 × 47 nm³) at an overall resolution of 4.8 Å (Fig. 3a and Extended Data Fig. 6a). Moreover, to aid in refinement of an atomistic model, we reconstructed a smaller volume, 34 × 34 × 34 nm³, spanning four complete CA hexamers and one well-defined IN octamer, at 4.6 Å (Extended Data Fig. 6b and Extended Data Table 1). The cryo-EM maps enabled unambiguous docking and real-space refinement of the CA and IN subunits, consistent with the resolution metrics.

The overall architecture of the IN filament visualized in native HIV-1 cores is highly similar to the arrangement in the SIVtal IN–RNA[TAR] complex (compare Figs. 1b and 3c). In cores, each IN tetramer contributing to the octamer unit nestles in a CA hexamer, making four distinct points of contact, all involving the major homology region (MHR) of the corresponding CA subunits. Highly conserved among orthoretroviruses, the MHR, located within the CA CTD, has three invariant residues (Gln155, Glu159 and Arg167 in HIV-1) that span a loop (residues 153–160) and α-helix 8 (residues 161–172) (Extended Data Fig. 7a). The MHR has essential roles in HIV-1 assembly and virus maturation[28,29].

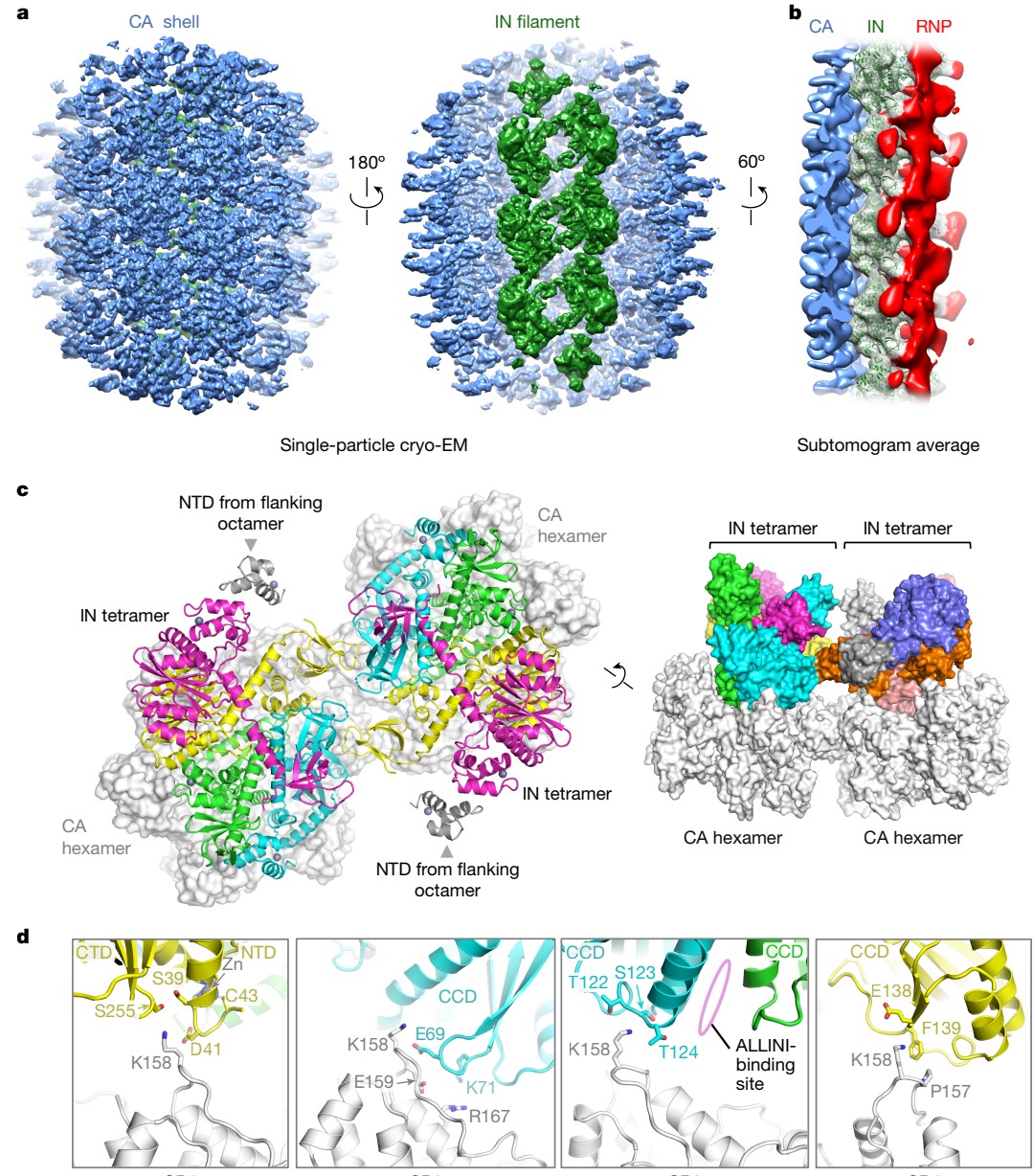

**a** CA shell · IN filament

180°

Single-particle cryo-EM

**b** CA · IN · RNP

60°

Subtomogram average

**c** NTD from flanking octamer · CA hexamer · IN tetramer · IN tetramer · CA hexamer · NTD from flanking octamer · IN tetramer · IN tetramer · CA hexamer · CA hexamer

**d** CR1 · CR2 · CR3 · CR4

CR1: CTD, NTD, Zn, S39, C43, S255, D41, K158

CR2: CCD, K158, E69, E159, K71, R167

CR3: CCD, T122, S123, CCD, T124, K158, ALLINI-binding site

CR4: CCD, E138, F139, K158, P157

**Fig. 3 | IN filament at the lumen of native HIV-1 cores. a**, Cryo-EM map reconstructed using a box size of 47 × 47 × 47 nm³, viewed from the outer (left) and luminal (middle) surfaces of the viral core. For clarity, the map was filtered using EMReady[43]. The map regions corresponding to the CA shell and IN octamers are shown in light blue and green, respectively. **b**, Reconstruction by subtomogram averaging of cores produced without Vpr-NG-IN co-expression. The map regions corresponding to IN are shown as a semi-transparent surface with fitted IN octamers as green cartoons, while those not explained by CA or IN are coloured red. Further details are provided in Extended Data Fig. 9f. **c**, Cartoon representation of the IN octamer (coloured as in Fig. 1b) seated on CA hexamers (light grey, shown as a surface representation) shown along its two-fold symmetry axis (left) and along the plane of the CA lattice (right). NTDs from flanking octamers are shown as grey cartoons. **d**, Magnified views of CR1–4. Protein chains are shown as cartoons, coloured as in **c**. Although conformations of amino acid side chains are poorly defined due to modest resolution of the cryo-EM map, they are shown as sticks to indicate their approximate locations as they appear in the refined model. The site of ALLINI binding is indicated. Images of cryo-EM maps in **a** and **b** were generated using UCSF Chimera; for **c** and **d**, the diagrams were created using PyMOL.

HIV-1 CA Lys158, located at the tip of the MHR loop, comes in close proximity to IN residues Ser39–Asp41, Glu69–Lys71, Thr122–Thr124 and Glu138–Phe139, which we refer to as CA contact regions 1, 2, 3 and 4 (CR1–4), respectively (Fig. 3d). Although the resolution of the cryo-EM map is insufficient to determine side chain orientations, the amino acid compositions of the CRs strongly suggests formation of multiple ionic, hydrogen bonding and hydrophobic contacts. Extensive cryo-EM density-guided molecular dynamics (MD) simulations supported the roles of CA residues Lys158, Glu159 and Pro157 in

interactions with IN residues across all four contact regions (Extended Data Fig. 8).

To validate our observations, we reconstructed cryo-electron tomograms of HIV-1 cores derived from virions produced without Vpr-NG-IN co-expression (Extended Data Fig. 9a). Template matching using a CA lattice from our cryo-EM structure, followed by 3D classification, identified a subset of subvolumes containing well-defined linear IN polymers (Extended Data Fig. 9b,c). Subsequent image processing and subtomogram averaging yielded a 3D reconstruction of the IN filament bound

to the luminal face of the CA lattice at an overall resolution of 12.6 Å (Fig. 3b, Extended Data Fig. 9d–f and Extended Data Table 1). In addition to confirming the IN–CA interactions, the cryo-ET map revealed a prominent density overlaying the IN filament, probably corresponding to the viral RNP. Notably, this density penetrated the IN filament at sites that correspond closely to the RNA duplex-binding positions observed in our SIVtal IN–RNA[TAR] structure (Extended Data Fig. 9f). The overlaying RNP density was noticeably weaker in our single-particle cryo-EM reconstruction, potentially due to the oversupply of IN in those cores (Extended Data Fig. 6b).

## Importance of the IN and CA interfaces

To test the importance of the observed IN–CA contacts, we introduced mutations into a single-round derivative of HIV-1[NL4-3] carrying the luciferase reporter gene (HIV-Luc). We assessed the ability of the mutant constructs to produce viral particles, infect target cells and, in some cases, support reverse transcription. Virus IN and CA content was assessed by semi-quantitative western blotting, and virion morphology by EM of fixed and stained specimens (Fig. 4 and Extended Data Fig. 7b).

IN mutants D41K, E69A/K and F139A/D were analysed to examine CR1, CR2 and CR4 functionalities, respectively; as the key IN CR3 residue Thr124 is highly polymorphic among circulating strains[30], it was not targeted. D41K- and F139A-mutant viruses had comparatively mild (33% and 55%, respectively) infection defects. By contrast, E69K-, E69A- and F139D-mutant viruses had infection defects of over 100-fold (Fig. 4b). While the IN-to-CA ratios in these defective viruses were similar to WT, each nevertheless revealed significant increases in the frequency of eccentric particles (Fig. 4d and Extended Data Fig. 7b). Consistent with this observation, the E69K-, E69A- and F139D-mutant viruses exhibited significant ~50% to 80% defects in viral DNA synthesis (Fig. 4c). To assess the importance of the IN NTD in mediating filament stacking, we targeted Val32 (Fig. 2c). V32A- and V32E-mutant viruses were highly defective (>1,000-fold infection defects). In addition to having significant >10-fold defects in reverse transcription and increased eccentric and immature particle counts, these mutants were defective for virion IN protein content, similar to the class II IN mutant control virus H12N[2,18] (Fig. 4 and Extended Data Fig. 7b).

To examine the importance of the CA MHR residues, we analysed multiple CA Lys158 substitutions (K158R/A/E/D). Owing to the proximities of CA Glu159 and Arg167 to IN in CR2 (Fig. 3d and Extended Data Fig. 8), CA mutants E159D/Q and R167A were also analysed. As controls, we assessed Q155A/N changes of the remaining invariant MHR residue and K170A, which altered a Lys residue within α8 distal from the IN-binding interface (Extended Data Fig. 7a). The CA mutants Q155A and E159D displayed severe >25-fold particle-release defects and were accordingly omitted from downstream biochemical analyses (Fig. 4a). The other mutants with invariant MHR residue changes, Q155N, E159Q and R167A, supported <0.1%, 3.1% and 5.7% of WT HIV-Luc infection, respectively (Fig. 4b). Q155N and E159Q particles were significantly enriched for enlarged irregular structures (Fig. 4d), as previously described for K158A and K158D[29]. Although these structures also predominated K158E virus preparations, both K158E and K158A were also enriched for eccentric particles (Fig. 4d). Notably, R167A virions were specifically enriched for eccentric particles (Fig. 4d).

CA Lys158, together with Lys227, incorporates the essential CA assembly cofactor inositol hexakisphosphate (IP6) into immature virions, and the second-site T8I substitution in Gag SP1 compensated for the particle-release and infectivity defects of the K158A mutant virus[31], which was recapitulated here (Fig. 4a–c). T8I similarly improved the particle-release, infection and reverse-transcription defects of the K158E virus by around 2.7-fold, 200-fold and 17-fold, respectively (Fig. 4a,b). Notably, T8I counteracted the formation of irregular K158A and K158E structures, yielding K158A/T8I and K158E/T8I viruses specifically enriched for eccentric particles (Fig. 4d). The CA(K170A)-mutant

virus was also severely defective[29] (Fig. 4b). However, these particles were largely morphologically similar to the WT (Fig. 4d), consistent with the distal positioning of Lys170 relative to IN in the structure. Although R167A-mutant virions were first imaged over 30 years ago[28], the importance and the molecular basis of the eccentric phenotype was unclear.

HIV-1 IN uses cellular LEDGF/p75 to gain selectivity for transcription units during viral DNA integration[32,33]. The LEDGF/p75 IN-binding domain (IBD) engages a cleft at the IN CCD dimer interface, which is also targeted by ALLINIs[3,34–36]. Notably, two out of the four IBD interaction sites are occluded within each HIV-1 IN tetramer on the CA lattice, and CR3 lies proximal to the ALLINI-binding site (Fig. 3d and Extended Data Fig. 6c), suggesting that IN–CA and IN–LEDGF/p75 interactions may be exclusive. To test this hypothesis, we produced viral particles by co-transfection of the HIV-Luc proviral construct with a vector expressing Vpr fused to WT or IN interaction-defective IBD mutant I365A/D366N[37]; a third construct with a stop codon to interrupt IBD translation was used as a control. Consistent with IN displacement from the CA lattice, the virus containing the WT IBD displayed three to fourfold reductions in reverse transcription and infectivity, respectively, as well as an elevated fraction of eccentric particles, although we cannot rule out potential contributions from the approximately 45% reduction in virion IN protein (Fig. 4a–d and Extended Data Fig. 7b).

## Discussion

By imaging native HIV-1 cores, we identified that IN forms a regular filament assembly intimately associated with the luminal side of the CA lattice (Fig. 3 and Extended Data Fig. 9f). Our structure of the in vitro assembled SIVtal IN–RNA[TAR] complex demonstrates that the IN filament represents a functional RNA binding module (Fig. 1). Notably, the cryo-ET reconstruction of the filament in native HIV-1 cores revealed additional density, not explained by CA or IN, that closely matches the position of the RNA[TAR] duplexes in a complex with SIVtal IN (Extended Data Fig. 9f), strongly indicating that these positions are occupied by RNA in vivo. Capture of variable stem loops in viral RNA by the IN filament would staple the entire RNP to the core lumen. However, more work is required to dissect IN interactions with the entire gamut of viral RNA structures, including single-stranded RNA (Extended Data Figs. 1 and 3a).

As HIV-1 particles contain an estimated 120 IN molecules[38,39], we expect around 15 IN octamers per 200–250 CA hexamers within a mature viral core, with less than 20% of CA hexamers occupied by IN tetramers. Notably, the filaments that were identified through our subtomogram classification invariably oriented along the longest dimensions of HIV-1 cores (Extended Data Fig. 9a). It will be important to determine the properties of the CA lattice configuration that is preferred or perhaps enforced by the IN filament. Although the MHR has long been recognized for its role in orthoretroviral particle assembly and maturation[28,40], our data clarify that this region of CA also helps to nucleate RNPs for IN-mediated incorporation of viral RNA into mature HIV-1 cores. Future work will aim to assess the roles of other retroviral MHRs in IN-mediated encapsulation of their RNA genomes.

The stoichiometry of retroviral IN complexes follows an empirical $2^N$ rule: the minimal IN complex is a dimer, mediated by a symmetric interaction of CCDs[41]. While further multimerization is genus specific, the studied examples of lentiviral INs from HIV-1, MVV and SIVtal exist as stable tetramers in the absence of nucleic acids[11,12] (Extended Data Fig. 2a). The lentiviral IN tetramer—a dimer of dimers[10,14]—serves as a building block for the lentiviral supramolecular complexes involved in RNA retention (Figs. 1b and 3c) and DNA integration[15,42]. Comparison of the MVV intasome, comprising four IN tetramers[13,15], with the HIV-1 IN–RNA filament propagating along the capsid lumen reveals marked similarities. Nestled onto CA hexamers, two consecutive pairs of HIV-1 IN tetramers are prepositioned for hexadecameric intasome assembly,

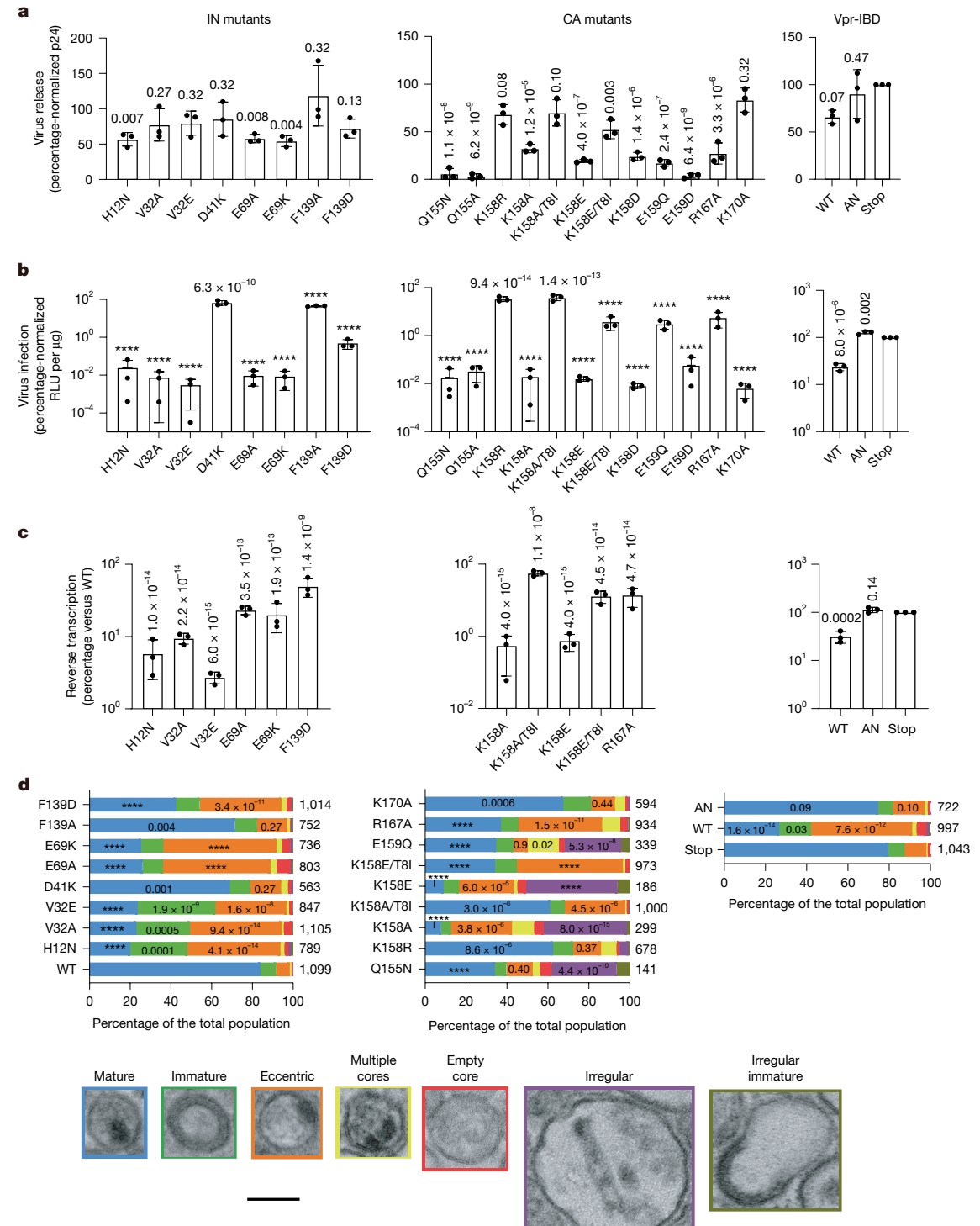

**Fig. 4 | Mutant viral phenotypes. a**, HIV-1 IN and CA mutant virus release percentage-normalized to WT HIV-Luc. Data are mean ± s.d. of *n* = 3 independent experiments, each comprising technical duplicates. Viruses with Vpr-IBD were percentage-normalized to Vpr-Stop. AN, I356A/D366N. **b**, Viral infectivities (RLU per μg) were percentage normalized as in **a**. Data are mean ± s.d. of *n* = 3 independent experiments, each with duplicate samples. H12N infectivity data are from ref. 44. **c**, Reverse transcription, percentage-normalized as described in **a**. Data are mean ± s.d. of *n* = 3 independent experiments, each with duplicate infection samples. PCR assays used technical triplicates. **d**, Particle morphologies (percentage of the total populations) were assessed using transmission electron microscopy (TEM). Categories were assigned as follows: (1) mature (blue): RNPs congruent with a discernible side view or end-on capsid; (2) immature (green): partial or full radially arranged toroidal electron density; (3) eccentric (orange):

RNPs situated outside a discernible capsid; (4) multiple cores (yellow): the presence of two or more cores; (5) empty core (red): particle lacking electron density and with a discernible capsid; (6) irregular (purple): comparatively large structures with indications of viral RNP and/or capsid structures; (7) irregular immature (greenish brown): comparatively large structures containing radially arranged toroidal features. Similar results were observed across two independent experiments. The numbers to the right of the stacked bars are the total particle counts. Statistical analyses (versus WT virus or Vpr-Stop) were performed using ordinary one-way (**a**–**c**) or two-way (**d**) analysis of variance (ANOVA) with Holm–Šídák correction for multiple comparisons. *P* values are indicated at the top of the vertical bars (**a**–**c**) and in **d** for all mature and eccentric populations. For other populations, *P* values of <0.05 are indicated (for unlabelled populations, *P* > 0.05); ****$P < 10^{-15}$.

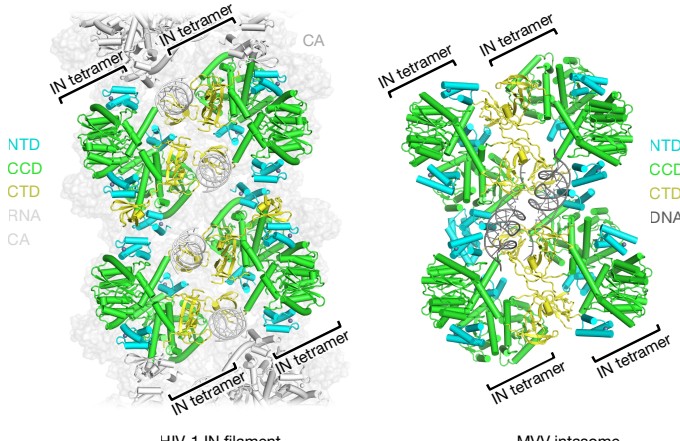

**Fig. 5 | The structural relationship of the proto-intasomal IN–RNA filament and the intasome.** Two consecutive HIV-1 IN octamer repeat units of the filament (left) and MVV intasome[13,15] (right; PDB: 7Z1Z) are shown as a cartoon representation with IN chains coloured according to the domain: NTDs (cyan), CCDs (green) and CTDs (yellow). CA hexamers and DNA are shown as a light grey surface representation and a grey cartoon, respectively. Neighbouring IN subunits within the CA/RNA-bound IN filament are shown in grey. IN tetramers are indicated by square brackets. Supplementary Video 2 demonstrates morphing of the two structures. The images were created using PyMOL.

requiring only minor reorientation and a swap of a pair of NTDs to bind to viral DNA ends (Fig. 5 and Supplementary Video 2). We accordingly propose 'proto-intasomal filament' as an apt moniker for the IN–RNA structure. It is tempting to speculate that the CA lattice contributes to intasome assembly as the RNA template is degraded during viral DNA synthesis by reverse transcriptase. Notably, the target-DNA-binding surface of the intasome formed in this manner would face the underlying CA lattice, potentially helping to prevent autointegration before chromosomal DNA can be engaged for intermolecular strand transfer. Furthermore, the observed incompatibility between IN–CA and IN–LEDGF/p75 interactions (Extended Data Fig. 6c) suggests that the host factor may facilitate the disengagement of IN from the core lumen to enable integration. Given the overall architectural similarities between the proto-intasomal filament and the intasome, it is not surprising that so many amino acid substitutions in IN elicit the class II mutant phenotype[2,5,6]. Notably, the spacing of CA hexamers along the immature lattice is incongruent with the dimensions of the proto-intasomal filament (Extended Data Fig. 4a), indicating that the IN–RNA complex engages the capsid during mature CA lattice formation. In conclusion, our results provide the structural bases for IN–RNA interactions and the formation of the proto-intasomal IN–RNA filament on the luminal side of the capsid, explaining the role of IN in RNP incorporation into morphologically functional HIV-1 cores and warranting further work to study retroviral CA–IN interactions at atomistic detail.

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

## Methods

### Recombinant proteins

The vector pCPH6P-HIV1-IN used for bacterial expression of HIV-1 IN with a cleavable hexahistidine (His$_6$) tag was previously described[45]. To obtain pCPH6P-SIVtal-IN, the DNA region encoding HIV-1 IN was replaced with a codon-optimized fragment encoding SIVtal IN (the corresponding amino acid sequence was derived from NCBI GenBank entry CAJ57812)[46]. HIV-1 and SIVtal INs were produced in endonuclease-A-deficient *Escherichia coli* PC2 cells (BL21(DE3), endA::Tet$^R$, T1$^R$, pLysS)[47] transformed with pCPH6P-HIV1-IN and pCPH6P-SIVtal-IN, respectively. The cells were grown in LB medium containing 120 µg ml$^{-1}$ ampicillin to an absorbance at 600 nm ($A_{600}$) of around 0.9 and supplemented with 50 µM ZnCl$_2$, and protein expression was induced by addition of 0.01% (w/v) isopropyl-β-D-1-thiogalactopyranoside for 4 h at 30 °C. Bacterial cells were lysed by sonication in core buffer containing 1 M NaCl and 20 mM Tris-HCl, pH 7.5 supplemented with 1 mM phenylmethylsulfonyl fluoride and cOmplete EDTA-free protease inhibitor cocktail (Roche). To prevent aggregation of HIV-1 IN, all buffers used during purification of this protein were supplemented with 7.5 mM 3-((3-cholamidopropyl) dimethylammonio)-1-propanesulfonate (CHAPS); the detergent was not required and was avoided during purification of SIVtal IN. The supernatant was precleared by centrifugation and incubated with NiNTA agarose (Qiagen) for 1 h °C at 4 °C in the presence of 15 mM imidazole to capture His$_6$-tagged proteins. The resin was washed extensively in core buffer supplemented with 15 mM imidazole, and the protein was eluted with 200 mM imidazole in core buffer. To remove the hexahistidine tag, the eluate was incubated with human rhinovirus 14 3C protease (1:50 (w/w) ratio) overnight at 4 °C in the presence of 5 mM dithiothreitol (DTT). Cleaved proteins were diluted with ice-cold 20 mM Tris-HCl, pH 7.5 to adjust NaCl concentration to around 150 mM and immediately injected into a precooled 5 ml HiTrap Heparin HP column (GE Healthcare). A linear 0.15–1 M NaCl gradient in 7.5 mM CHAPS, 20 mM Tris-HCl, pH 7.5 (for HIV-1 IN) or 30 mM HEPES-NaOH, pH 7.5 (SIVtal IN) was used to elute the proteins; peak fractions were combined, supplemented with 1 mM DTT and the NaCl concentration was adjusted to 1 M. HIV-1 and SIVtal INs were concentrated using a 10-kDa cut-off Vivaspin device (Generon) to 8–10 and 1 mg ml$^{-1}$, respectively, snap-frozen in liquid nitrogen and stored at −80 °C.

### BLI analysis

All of the experiments were conducted using the Octet R8 instrument (Sartorius) at 25 °C in base buffer containing 150 mM NaCl, 20 mM Tris-HCl, pH 7.5, 1 mM DTT, 0.05% (v/v) Tween-20 and 10 U ml$^{-1}$ SUPERaseIn RNase inhibitor (LifeTechnologies, AM2696); sensorgrams were recorded using Octet BLI Discovery Software (Sartorius). Biotinylated RNA oligonucleotides (10 or 20 nM; Integrated DNA Technologies) were immobilized on Octet Streptavidin biosensors to reach 0.2 nm wavelength shift threshold in base buffer. The sensors were moved to wells containing 0.5, 0.25 or 0.125 µM HIV-1 or SIVtal IN in base buffer, and IN binding was recorded for 300 s. Dissociation was recorded for 300 s in the same buffer without IN. In control experiments, sensors without immobilized RNA were exposed to varying IN concentrations to test for non-specific binding.

### HDX–MS

Individual proteins and protein–RNA complexes (3 µM, final) were incubated with 40 µl D$_2$O buffer for 3, 30 and 180 s at room temperature in triplicate. The labelling reaction was quenched by adding chilled 2.4% (v/v) formic acid in 2 M guanidinium hydrochloride and immediately frozen in liquid nitrogen. The samples were stored at −80 °C before analysis. The quenched protein samples were rapidly thawed and processed for proteolytic cleavage by pepsin followed by reversed-phase HPLC separation of the resulting peptides.

In brief, the protein was passed through an Enzymate BEH immobilized pepsin column (2.1 × 30 mm, 5 µm, Waters) at 200 µl min$^{-1}$ for 2 min and the peptic peptides were trapped and desalted on a 2.1 × 5 mm C18 trap column (Acquity BEH C18 Van-guard pre-column, 1.7 µm, Waters). Trapped peptides were subsequently eluted and separated over 11 min using a 5–43% gradient of acetonitrile in 0.1% (v/v) formic acid at 40 µl min$^{-1}$. Peptides were separated on a reverse-phase column (Acquity UPLC BEH C18 column 1.7 µm, 100 mm × 1 mm; Waters). Peptides were detected on the Cyclic mass spectrometer (Waters) acquiring over an $m/z$ of 300 to 2,000, with the standard electrospray ionization source and lock mass calibration using [Glu1]-fibrino peptide B (50 fmol µl$^{-1}$). The mass spectrometer was operated at a source temperature of 80 °C with a spray voltage of 3.0 kV. Spectra were collected in positive-ion mode.

Peptide identification was performed by MS$^e$ software[48] using an identical gradient of increasing acetonitrile in 0.1% (v/v) formic acid over 12 min. The resulting MS$^e$ data were analysed using Protein Lynx Global Server software (Waters) with an MS tolerance of 5 ppm. Mass analysis of the peptide centroids was performed using DynamX software (Waters). Only peptides with a score >6.4 were considered. The first round of analysis and identification was performed automatically by DynamX software; however, all peptides (deuterated and non-deuterated) were manually verified at every timepoint for the correct charge state, presence of overlapping peptides and correct retention time. Deuterium incorporation was not corrected for back-exchange and represents relative, rather than absolute changes in deuterium levels. Changes in hydrogen–deuterium amide exchange in any peptide may be due to a single amide or a number of amides within that peptide. All timepoints in this study were prepared at the same time and individual timepoints were acquired on the mass spectrometer on the same day. The MS data have been deposited to the ProteomeXchange Consortium through the PRIDE[49] partner repository under dataset identifier PXD070910.

### SEC–MALLS analysis

For size-exclusion chromatography coupled to multiangle laser light scattering (SEC–MALLS) analysis, SIVtal IN (100 µl) in 0.5 M NaCl, 3 mM NaN$_3$, 0.5 mM Tris-(2-carboxyethyl)phosphine (TCEP) and 25 mM Tris-HCl (pH 7.5) was injected onto the Superdex-200 Increase 10/300 column (Cytiva) equilibrated in the same buffer. Chromatography was performed at 25 °C, at a flow rate of 1 ml min$^{-1}$ using the JASCO-4000 semimicro HPLC system. Scattered light intensities and protein concentrations in the eluate were measured using the DAWN-HELEOS II laser photometer and an OPTILAB-TrEX differential refractometer (Wyatt Technology), respectively. The data were analysed using ASTRA software v.7.3.2 (Wyatt Technology) using recordings from both detectors, assuming a specific refractive index increment (d$n$/d$c$) of 0.186 ml g$^{-1}$. The weight-averaged molar mass of SIVtal IN in chromatography peaks was determined from the combined data of three independent experiments with IN diluted to 1, 0.5 and 0.25 mg ml$^{-1}$.

### Cryo-EM data collection on the SIVtal IN–RNA$^{TAR}$ complex

Graphene oxide grids were prepared as previously described[50]. In brief, UltrAuFoil R1.2/1.3 grids (Quantifoil) were pretreated in a glow discharge unit (GloQube Plus, Quorum) for 5 min at 25 mA, and then incubated with 4 µl 0.22 mg ml$^{-1}$ graphene oxide flake suspension (Sigma-Aldrich) for 2 min. Synthetic RNA (Integrated DNA Technologies) diluted in RNase-free water to 1.175 µM, was incubated at 95 °C for 5 min and cooled on ice. SIVtal IN–RNA complexes were assembled by combining IN (prepared in 0.5 M NaCl, HEPES-NaOH, pH 7.5) and RNA at final concentrations of 4 µM (0.13 mg ml$^{-1}$) and 1 µM, respectively, in 0.15 M NaCl, 25 mM HEPES-NaOH, pH 8.0. The mixture (4 µl) was immediately applied to a graphene-oxide-coated grid. The grids were blotted on both sides for 4 s in a Vitrobot Mark IV (FEI, Thermo Fisher Scientific) instrument at 4 °C and 95% humidity and vitrified by

plunging into liquid ethane-propane. Blotting paper was from Agar Scientific (47000-100), and the blot force was set to −1.

Data collection was performed on the Titan G2 transmission electron microscope (Thermo Fisher Scientific) operated at 300 kV and equipped with a Falcon 4i direct electron detector and a Selectris energy filter (Thermo Fisher Scientific; Extended Data Table 1). Micrographs were recorded using EPU software (Thermo Fisher Scientific) in the dose-fractionation mode, at a calibrated magnification corresponding to 0.95 Å per physical pixel and a total sample exposure dose of 40.3 e⁻ Å⁻². A total of 38,395 micrograph movies was collected (3 per foil hole) automatically using EPU software (Thermo Fisher Scientific) with an energy filter slit set to 10 eV and a defocus range −1 to −4 μm. In total, 1,674 EER frames, recorded per micrograph movie, were subsequently processed in 31 fractions, with an exposure dose of 1.3 e⁻ Å⁻² per fraction. The movie frames were aligned with dose weighting using Relion-4.0 (refs. 51,52), and contrast transfer function (CTF) parameters were estimated from frame sums using Gctf-v1.18 (ref. 53).

### Single-particle image processing and structure refinement of the SIVtal IN–RNA^TAR complex

An initial set of particles picked using crYOLO with the general model[54] were processed for reference-free 2D classification in cryoSPARC-4.6.2 (ref. 55). Particles belonging to well-defined 2D classes were used to train a model for particle picking in Topaz[56]. The entire dataset was picked with Topaz using the trained model and with Gautomatch-v0.56 (https://www.mrc-lmb.cam.ac.uk/kzhang/Gautomatch/), using 2D class averages (low-pass filtered to 18 Å) as templates. Particles, extracted with pixel size 3.8 Å, were processed for several rounds of 2D classification in cryoSPARC, and those contributing to well-defined 2D classes (Extended Data Fig. 3a) were re-extracted with a pixel size 1.9 Å for tandem ab initio reconstruction and heterogenous refinement in cryoSPARC into three classes. Particles from the Topaz- and Gautomatch-picked subsets belonging to a single well-defined 3D class were combined, pruned of duplicates using the relion_star_handler tool with a proximity cut-off of 35 Å and used for training an optimized Topaz model. The entire dataset was picked again using Topaz with the new model, and the particles underwent the same image processing, including 2D classification, ab initio reconstruction into three classes followed by heterogenous refinement in cryoSPARC. Particles from a single well-defined 3D reconstruction from each separately picked set were combined and filtered for duplicates with a proximity cut-off of 35 Å. The resulting 181,089 particles, re-extracted without binning, were processed for tandem ab initio reconstruction and heterogenous refinement in cryoSPARC into three classes, yielding a single well-defined class comprising 170,999 particles. Non-uniform refinement of these particles resulted in a 3D reconstruction at an overall resolution of 4.3 Å with $C_2$ symmetry imposed. To take advantage of the translational symmetry, the particle set was expanded by extracting potentially new particles offset by one repeat length in both directions along the IN polymer axis (98 micrograph pixels in either direction along the $y$ axis of the reconstruction). The expanded dataset, pruned for duplicates, was processed for two rounds of tandem cryoSPARC ab initio reconstruction and heterogenous refinement into three classes, yielding a subset of 182,209 particles. This translation symmetry expansion procedure was repeated one more time, resulting in 225,773 particles affording a 3D reconstruction at 4.1 Å resolution with $C_2$ symmetry imposed. An additional round of purification by tandem ab initio reconstruction and heterogenous refinement into two classes yielded a subset of 219,353 particles. Non-uniform refinement of these particles resulted in a 3D reconstruction at 4.0 Å resolution with $C_2$ symmetry imposed, which was used for per-particle CTF refinement in cryoSPARC and Bayesian polishing in Relion-4.0. At this stage, local refinement in cryoSPARC with a soft mask around a single IN octamer repeat unit and the associated RNA chains produced a 3D reconstruction at a resolution of 3.5 Å using $C_2$ symmetry. The refined particles

were sorted into six 3D classes without realignment in Relion while imposing a soft mask focusing on a single repeat unit plus associated RNA and the regularization parameter $T = 4$; 102,121 particles contributing to the high-resolution class allowed a local reconstruction at an overall resolution of 3.3 Å using $C_2$ symmetry. To select particles with best-ordered RNA chains, locally refined particles were processed for the particle subtraction procedure to remove signal contributed by IN subunits followed by 3D classification without realignment and without imposing symmetry into five classes with a soft mask focusing on the RNA subunits and the regularization parameter $T = 4$, as implemented in Relion. CryoSPARC global non-uniform and local refinement of the selected best 53,737 particles produced the final reconstructions at 3.7 and 3.3 Å resolution, respectively, using $C_2$ symmetry (Extended Data Fig. 3b,c and Extended Data Table 1).

The resolution metrics reported here are according to the gold-standard Fourier shell correlation (FSC) 0.143 criterion[57,58]. Local resolution of the 3D reconstruction was estimated in cryoSPARC (Extended Data Fig. 3c). For illustration purposes and to aid in model building, the cryo-EM map was processed with EMReady-v2 (http://huanglab.phys.hust.edu.cn/EMReady2/)[43]; for real-space refinement of the atomistic model, the reconstruction was sharpened and filtered as implemented in cryoSPARC based on local resolution metrics.

The model of the SIVtal IN–RNA complex was initially assembled by rigid body docking of individual HIV-1 IN domains (from PDB 1K6Y (ref. 14), 8A1P (ref. 36) and 5TC2) in the cryo-EM map using UCSF Chimera[59]. Because the orientation (and possibly register) of the HIV-1 TAR stem loop is undetermined, the RNA was modelled as an oligo-A/U duplex. The model was adjusted in Coot[60] to match the amino acid sequence of SIVtal IN and refined using phenix.real_space_refine (v.1.21.2-5419)[61]. The quality of the final model was assessed using MolProbity[62] (Extended Data Table 1). Structural images were generated using UCSF Chimera[59] and PyMOL (https://www.pymol.org/).

### Cryo-EM of SIVtal IN in the absence of RNA and in the presence of (GA)₁₈ ribooligonucleotide

SIVtal IN in the absence of RNA was prepared and vitrified on graphene-oxide-coated holey carbon grids exactly as described for the IN–RNA complex. In total, 3,220 micrograph movies were recorded on a 200 kV Glacios microscope using a Falcon 3 direct electron detector operated in linear mode at a calibrated magnification corresponding to a pixel size of 1.24 Å with accumulated total specimen exposure of 53 e⁻ Å⁻², spread over 10 fractions and a defocus range of −1.5 to −3.5 μm. Micrograph movie frames were aligned and combined using MotionCor2 (v.1.4.0)[63] with dose weighting, and CTF parameters were estimated from frame sums using Gctf (v.1.18)[53]. The micrographs revealed sparse patches of aggregated material without obvious regular supramolecular assemblies. Accordingly, no well-defined 2D class averages were obtained with particles picked using crYOLO v.1.9.6 with general model (Extended Data Fig. 3a).

To image SIVtal IN-(GA)₁₈ complexes, a sample containing 4 μM SIVtal IN and 1 μM ribooligonucleotide in 150 mM NaCl, 20 mM HEPES-NaOH (pH 8.0) was vitrified on an UltrAufoil 1.2/1.3 300-mesh grid, using the Vitrobot Mark IV (FEI, Thermo Fisher Scientific) instrument at 4 °C and 95% humidity. The grid was pretreated in the GloQube Plus glow discharger (Quorum) at 25 mA for 1 min. The sample was spotted onto the grid twice (10 μl each time) with blotting between the applications (3.5 s blot time, force −10, 10 s wait time, 4 °C). In total, 40,377 micrograph movies were recorded on the Titan Krios microscope equipped with a Falcon 4 camera and processed as described above for the SIVtal-RNA^TAR complex. The images revealed the presence of well-defined linear IN polymers (Extended Data Fig. 3a). Iterative rounds of particle picking and 2D classification resulted in well-defined 2D averages consistent with formation of single IN filaments (Extended Data Fig. 3a). Owing to strong preferential orientation with only top and side views present, further image processing did not yield a high-quality 3D reconstruction.

## Preparation of HIV-1 cores for cryo-EM, data collection and preprocessing

Lenti-X HEK293T cells (Takara Bio, 632180) were cultured in Dulbecco's modified minimal medium (DMEM, Life Technologies) supplemented with 10% (v/v) FBS (Life Technologies) at 37 °C in a humidified 5% $CO_2$ atmosphere. Cells were negative for mycoplasma, as evidenced via regular monthly testing with the MycoAlert mycoplasma detection kit (Lonza, LT07-218). The IN-coding region in pVpr-mNeonGreen-IN[64] was modified with D64N and D116N mutations to abrogate the active site in the protein product. To produce HIV-1 virions, Lenti-X HEK293T cells grown in two T175 flasks to around 70% confluence, were transfected with pNLC4-3 IN D64N/D116N tatΔ33-64bp[25] and pVpr-mNeonGreen-IN(D64N,D116N) (29 μg and 9.7 μg per flask, respectively) or with NLC4-3 IN D64N/D116N *tat*Δ33-64bp alone (39 μg per flask) using polyethylenimine (64 μg per flask) in OptiMEM medium (Life Technologies). The next day, the medium was changed to DMEM supplemented with 10% FBS. Virus-containing supernatant collected 48 h after transfection was precleared by centrifugation at 500*g* for 5 min and passed through a 0.45-μm filter. Viral particles were concentrated by ultracentrifugation through a cushion of 20% sucrose in ST buffer (25 mM Tris-HCl pH 7.5, 100 mM NaCl) supplemented with 1 mM inositol hexaphosphate (ST/IP6) in a SW32Ti rotor at 30,000 rpm for 3 h. The glassy green pellet containing viral particles was resuspended in ST/IP6. To isolate cores, concentrated viral particles were processed for ultracentrifugation through a layer of 1% (v/v) Triton X-100 into a linear 30–85% (w/v) sucrose gradient (made in ST/IP6) as previously described[65]. Fractions containing cores, identified by NG fluorescence or by the presence of reverse transcriptase activity (measured using a quantitative PCR assay[66] adapted for TaqMan technology[67]), were dialysed against ST/IP6 to remove sucrose. The fractions were analysed by western blotting (Extended Data Fig. 4c) with mouse monoclonal anti-HIV-1 CA antibody ARP-6458 (obtained through the NIH HIV Reagent Program, Division of AIDS, NIAID, NIH, contributed by M. H. Malim) and rabbit polyclonal anti-HIV-1 IN antibody[68], diluted at 1:5,000 and 1:10,000, respectively. Signals were developed using horseradish-peroxidase-conjugated goat anti-mouse IgG (Dako, P0447, 1:10,000) and swine anti-rabbit antibodies (Dako, P0399, 1:10,000) in conjunction with ECL Select detection reagent (Cytiva, RPN2235).

R2/2 300-mesh holey carbon grids (Quantifoil) were glow discharged using a PDC-002-CE plasma cleaner (Harrick Plasma) instrument at 100 mA for 45 s in air. Dialysed cores (3.5 μl) were applied to a pretreated grid under 95% humidity at 20 °C, before blotting and plunge-freezing in liquid ethane-propane using a GP2 Automatic plunge freezer (Leica Microsystems). Vitrified HIV-1 cores were imaged on a 300 kV Titan Krios G2 cryo-electron microscope equipped with a Falcon 4i direct electron detector and a Selectris energy filter (Thermo Fisher Scientific; Extended Data Table 1).

For structural analysis of the cores with supplemental IN, we used single-particle cryo-EM approaches that were previously used for 3D reconstruction of the CA lattice within in vitro assembled capsid-like particles[69–71]. Data were recorded using EPU software in dose-fractionation mode, at a calibrated magnification corresponding to 0.95 Å per physical pixel. 1,674 EER frames collected per micrograph movie were subsequently processed in 31 fractions, with an exposure dose of 1.6 e⁻ Å⁻² per fraction. A total of 47,025 micrograph movies was recorded (9 per foil hole) using EPU software (Thermo Fisher Scientific) with an energy filter slit width of 10 eV and a defocus range set at −1.5 to −3.5 μm. The micrograph movie frames were aligned with dose weighting as implemented in Relion (v.4.0)[51,52], and CTF parameters were estimated from motion corrected images using Gctf (v.1.18)[53].

Native cores isolated from HIV-1 particles without supplemental Vpr-NG-IN were studied by cryo-ET. The tilt series, each from −60° to +60° with 3° increments, were acquired using Tomography software (Thermo Fisher Scientific) with the dose-symmetric scheme[72], a total

exposure of 104 e⁻ Å⁻² (2.5 e⁻ Å⁻² per tilt image), a calibrated magnification resulting in a pixel size of 1.56 Å and a defocus range of −2.5 to −5.5 μm. Each tilt micrograph movie stack containing six frames in TIFF format was motion-corrected and summed in WarpTools[73]. Tilt images were aligned using patch tracking in IMOD[74], and CTF-corrected tomograms were reconstructed with WarpTools. To create illustrations, the tomograms were denoised using DeepDeWedge[75]. Reconstructed tomograms were visualized using ChimeraX[76,77] with ArtiaX plug-in[78].

## Single-particle cryo-EM image processing of HIV-1 cores with excess IN

YOLOv11, a deep-learning tool developed for general computer vision tasks (https://docs.ultralytics.com/models/yolo11/), was used to identify HIV-1 cores on micrographs. The small YOLOv11 model (yolo11s.pt, with 9.44 million parameters) was trained on a carefully curated subset of 595 micrographs containing 2,578 manually boxed cores. The training set additionally included 137 null examples, that is, micrographs lacking recognizable cores but containing full range of contaminating signal (crystalline ice, carbon edges and cellular debris) present. Manual annotation was done using Roboflow (https://app.roboflow.com), while model training and core detection used resources provided by GoogleColab (https://colab.research.google.com). JPEG copies of the micrographs (512 × 512 pixels), produced by the EPU software, were suitable for model training and core detection. The following parameters were used during training: model=yolo11s.pt, epochs=600, imgsz=640, patience=200, and the model reached mean average precision values mAP50 and mAP50−95 of 0.856 and 0.528, respectively. The model, applied to the entire dataset with confidence parameter set to 0.25, predicted a total of 120,747 cores, each defined by a rectangular bounding box (Extended Data Fig. 5a). All downstream image processing was done on 20,214 micrographs comprising 43% of the dataset, containing at least one core identified by YOLOv11.

The micrographs were picked exhaustively using a hierarchical four-step approach. Initially, particles were picked within each core bounding box with regular grid spacing of 120, 150 and 180 micrograph pixels. As some cores overlapped on the micrographs, each resulting particle set was filtered to remove duplications, allowing not less than 100 micrograph pixels (95 Å) between picked centres. The resulting 4,218,210 (spaced by 120 pixels), 2,837,521 (150 pixels) and 2,094,749 (180 pixels) particles extracted with a box size of 90 px (binned fourfold to 3.8 Å per pixel) were processed for 4–6 rounds of 2D classification in cryoSPARC, using 200 classes, a batch size of 400 particles per class and 50 on-line EM iterations. Particles belonging to well-defined 2D classes (Extended Data Fig. 5b) were used for ab initio reconstruction followed by heterogenous refinement in cryoSPARC using five classes (Extended Data Fig. 5c). 3D classes refining as a hexameric CA lattice from each subset were combined and pruned of duplicates enforcing a minimal distance cut-off of 60 Å, resulting in a subset of 495,638 particles. These were re-extracted with a box size of 120 pixels (binned fourfold to a pixel size of 3.8 Å) and processed for purification by heterogenous refinement in cryoSPARC (v.4.6.2) using two initial volumes: a refined reconstruction of the hexameric CA lattice obtained at the previous stage and a junk trap prepared by phase randomization of Fourier components with spatial frequences exceeding 1/40 Å⁻¹. The initial resolution was set to 15 Å and refinement box size to 120 pixels, with the rest of the parameters left at default values. The 235,835 particles remaining after three consecutive rounds of heterogenous refinement, re-extracted with pixel size of 1.9 Å, were processed for the heterogenous refinement procedure with a phase randomized junk trap. This time, the initial resolution was set to 12 Å and the refinement box size was set to 160 pixels. After three rounds of heterogenous refinement, the purified subset comprised 179,033 particles (crude set 1); non-uniform refinement of these particles without and with $C_2$ symmetry imposed resulted in 3D reconstruction of the CA lattice at 5.8 and 4.7 Å, respectively.

The particles were used to train a custom Topaz[56] model. The micrographs were repicked with Topaz using the trained model and with Gautomatch v.0.56 (https://www.mrc-lmb.cam.ac.uk/kzhang/Gautomatch/) using 2D class averages showing a well-defined CA lattice, low-pass-filtered to 18 Å resolution, as templates to repick the entire dataset. In total, 10,542,672 Topaz-picked and 3,689,441 Gautomatch-picked particles (found within YOLOv11 identified core bounding boxes) were extracted, binned fourfold and processed using the procedure detailed above, including consecutive rounds of 2D classification, heterogenous and non-uniform refinement (imposing $C_2$ symmetry at the final stage), yielding subsets of 477,807 and 443,013 aligned particles (crude sets 2 and 3, respectively). At this point, particles picked by the three methods (crude sets 1–3) were combined and filtered for duplications, imposing a minimal distance of 30 Å, resulting in a subset of 808,938 particles.

Global refinement of the combined particle set generated a reconstruction at a resolution of around 4.5 Å, and the alignment parameters were used to erase signal contributed by the CA lattice using particle subtraction procedure in Relion (Extended Data Fig. 5d). Modified particles were processed for 45 iterations of 3D classification into 7 classes without realignment in Relion with a soft semi-cylindrical mask focusing on the volume underlying the CA lattice and $T$ value set to 2 (Extended Data Fig. 5d); the starting model was the original 3D reconstruction low-pass filtered to 40 Å. The procedure isolated a class comprising 50,827 particles revealing a well-defined linear polymer of IN octamers (Extended Data Fig. 5d). These particles were used to train Topaz to repick the micrographs one more time. The resulting 5,795,765 particles were processed for 2D classification and heterogenous refinement yielding 219,985 aligned particles (crude set 4).

Crude sets 1–4 were combined and pruned for duplicates allowing a minimum of 30 Å between particle centres on micrographs, resulting in a collection of 975,484 particles. The particles, extracted with twofold binning, were processed for 3D autorefinement in Relion (imposing $C_2$ symmetry), followed by subtraction of CA signal and 3D classification without realignment (with a semi-cylindrical mask focusing on the volume underlying CA lattice and using eight classes) as described above. The procedure yielded a single 3D class corresponding to a chain of IN octamers and comprising 59,749 particles. To take advantage of the translational symmetry of the IN filament, the particle set was expanded by adding potentially new particles off-set by one and two repeat lengths in either direction along the IN polymer. The translational symmetry-expanded particle set was pruned for duplicates allowing a minimal distance between particle centres of 10 Å. The resulting 280,358 particles were used in non-uniform refinement followed by 3D classification (without realignment) into eight classes in cryoSPARC with a soft semi-cylindrical mask encompassing the space underlying the CA lattice. The best-defined 3D class, comprising 43,320 particles, was processed for non-uniform refinement and one more round of translational symmetry expansion resulting in 138,637 non-overlapping particles. Following 3D classification in cryoSPARC, several rounds of non-uniform refinement, duplicate pruning and CTF refinement in cryoSPARC, the final reconstructions were obtained through non-uniform refinement in cryoSPARC (with $C_2$ symmetry applied) using subsets comprising 58,103 and 46,116 particles to an overall resolution of 4.63 Å (with a box size of 240 × 1.44 Å, encompassing 1 IN octamer repeat unit and 4 complete CA hexamers) and 4.83 Å (332 × 1.44 Å, containing 3 complete IN octamers and 14 CA hexamers), respectively (Extended Data Fig. 6 and Extended Data Table 1). For illustration purposes and to aid in model building, the cryo-EM map was processed with EMReady v.2 (http://huanglab.phys.hust.edu.cn/EMReady2/)[43]. For real-space refinement of the atomistic model, the reconstruction was sharpened and filtered as implemented in cryoSPARC based on local resolution metrics.

The atomistic model was assembled by rigid-body docking CA hexamers (derived from PDB 6SKK)[79] and individual IN domains (from PDB 1K6Y (ref. 14), 9C9M (ref. 44) and 8A1P (ref. 36)) into the cryo-EM map using UCSF Chimera[59]. The model was extended and locally refined in Coot[60] followed by global refinement in Namdinator[80]. Two IP6 ligands were added per CA hexamer, based on the features of the cryo-EM map. The final model, derived by iterative cycles of improvement using Coot and phenix.real_space_refine (v.1.21.2-5419)[61], had a good fit to the cryo-EM map and reasonable geometry, as assessed by MolProbity[62] (Extended Data Table 1).

## Subtomogram averaging of IN filaments inside native cores produced in the absence of Vpr-NG-IN

Fourteen tilt series that were sufficiently well aligned to produce tomograms with clearly discernible hexagonal CA lattice features (Extended Data Fig. 9a) were selected for further processing. An initial subset of particles was picked by template matching in CTF-corrected tomograms (reconstructed at 10 Å per pixel) using pytom-match-pick[81]. The template represented the CA lattice isolated from the single-particle cryo-EM reconstruction of cores produced in the presence of Vpr-NG-IN ($47 × 47 × 47$ nm$^3$ box; see above). To minimize bias, features corresponding to IN were removed by segmentation in UCSF Chimera, and the resulting template was low-pass filtered to a resolution of 20 Å. A total of 2,375 subtomograms was extracted using Warp (3.02 Å per pixel; 192 pixel box size) and processed for 3D classification into six classes in Relion v.4.0. The initial reference, generated using relion_reconstruct from the extracted subtomograms (using the --ctf --3d_rot settings), was low-pass filtered to 60 Å. Classification was run for 45 iterations with a $T$ value of 0.2, without symmetry imposed. This procedure yielded a single class comprising 164 particles showing a well-defined IN filament (Extended Data Fig. 9b). These filaments could also be visualized in denoised tomograms when oriented approximately parallel to the $xy$ plane (Extended Data Fig. 9c). We found that the shape of CA lattice used for template matching was critical for identification of the subtomograms containing IN. This may be because the filaments prefer and/or induce specific local curvature of the core wall, which may explain orientation of the filaments along the sides of the cores (Extended Data Fig. 9a). Refinement of the subtomogram subset in Relion with a soft mask and $C_2$ symmetry resulted in a reconstruction at an overall resolution of 19 Å. To exploit the translational symmetry of the IN filaments, additional subtomograms were extracted by shifting the original particles by ±1 and ±2 IN octamer repeat units along the filament axis. Duplicate subtomograms were removed using relion_star_handler with a distance cut-off of 60 Å, resulting in a total of 773 unique particles. These were re-extracted in Warp and processed for 3D classification in Relion into six classes ($T = 1$, no symmetry imposed), yielding a subset of 494 particles contributing to three well-defined classes. This dataset was expanded again using the same translational symmetry strategy, resulting in 1,065 subtomograms after duplicate removal. These were classified into seven classes ($T = 1$, no symmetry imposed), yielding a subset of 617 particles, which was further refined to 594 particles through an additional round of 3D classification into six classes ($T = 1$, no symmetry imposed). This final set of subtomograms was used for 3D refinement in Relion, with $C_2$ symmetry and a soft mask, resulting in a reconstruction at an overall resolution of 12.6 Å, and local resolution of around 10 Å throughout most of the CA and IN regions (Extended Data Fig. 9d,e). A locally filtered map used to create illustrations was generated in cryoSPARC (Fig. 3b and Extended Data Fig. 9f).

## IN–CA complex atomic model building for MD simulations

Before performing MD simulations, we prepared a model of the IN octamer–CA hexameric lattice complex based on the cryo-EM map and refined atomistic model using the following procedure. First, missing residues not resolved in the cryo-EM density, corresponding to the IN NTD-CCD linker (residues 42–54) and short segments in the CCD (residues 139–143) and CTD (residues 267–272), were modelled

using the comparative modelling tool MODELLER (v.10.6)[82,83] in ChimeraX[76,77]. For each IN chain in the octamer, five independent full-length models were generated using the cryo-EM derived structure and the IN sequence as templates. The model with the lowest DOPE score was selected for each chain, and the resulting models were combined into a complete IN octamer structure. $Zn^{2+}$ cations coordinated by residues His12, His16, Cys40 and Cys43 in the IN NTDs were retained from the cryo-EM-derived structure. IN chain structures were consistent with the IN sequence used in the HIV-1 core samples. Thus, the IN model has the D64N and D116N amino acid substitutions, which precluded divalent metal ion coordination at the catalytic sites. The complete IN octamer structure was then combined with the CA hexameric lattice to generate a IN–CA complex model.

Hydrogen atoms were added on the basis of the predicted protonation state of the amino acids at pH 7.0, as determined using propKa3 (refs. 84,85). The protein complex was then solvated in a periodic box of TIP3P water molecules[86,87] using the solvate plug-in in Visual Molecular Dynamics (VMD) v.1.9.4a57, compiled using Python (v.3.9)[88]. $Na^+$ and $Cl^-$ ions were added to a concentration of 150 mM using the cionize and autoionize plug-ins in VMD[88], while ensuring the charge neutrality of the solvated system. To minimize the simulation box volume, the orientation of the IN–CA complex was optimized using rigid-body rotations. A minimum distance of 10 Å was maintained between the protein and the edge of the solvent box, resulting in a system size of 284 Å × 245 Å × 162 Å. The total atom count of the solvated IN–CA system was 1,054,000 atoms.

## MD simulation setup
As a further refinement step, we performed MD flexible fitting (MDFF)[89] to refine the position of the protein backbone in accordance with the cryo-EM density. In MDFF, the cryo-EM density map is used as a grid-based potential applied to selected coupled atoms, biasing their motion to fit into the density. For MDFF, we first performed energy minimization, coupling the protein backbone and heavy atoms of IP6 to the cryo-EM density using the gridForces module in NAMD (v.3.0.1)[90] with a gridScaling factor of 0.3 kcal mol$^{-1}$ amu$^{-1}$. The system was minimized for 35,000 steps using a conjugate gradient descent algorithm, ensuring that the gradient converged below 10 kcal mol$^{-1}$ Å$^{-2}$.

After minimization, we conducted a 10 ns equilibration of the system in an NPT ensemble (constant number of particles, pressure and temperature), maintaining the coupling to the density with gridScaling factor (0.3 kcal mol$^{-1}$ amu$^{-1}$) to allow the protein backbone to dynamically adjust its fitting to the density. Temperature and pressure were maintained constant during equilibration at 310 K and 1 atm, respectively. Finally, to alleviate any internal strain introduced by the biasing potential, we performed a scheme of five sequential 35,000 step minimization runs, using progressively reduced gridScaling factors (5.0, 1.0, 0.5, 0.1, 0 kcal mol$^{-1}$ amu$^{-1}$). The resulting structure from the procedure previously described, was then used as the starting structure for subsequent simulations of the IN–CA complex.

## MD production simulations
The resulting structure from the MDFF procedure was used as the starting model for four independent production simulations. Each replica simulated the IN–CA complex for 1 µs in an NPT ensemble, with grid force coupling applied to the protein backbone (for residues well resolved in the cryo-EM density) using a grid scaling factor of 0.3 kcal mol$^{-1}$ amu$^{-1}$. This coupling preserved the agreement of the IN–CA protein backbone with the observed density while allowing for side chains to interact freely throughout the simulation. Pressure was maintained at 1 atm using the Nose–Hoover Langevin piston, with a period of 200 fs and a decay time of 100 fs. Pressure control was configured to maintain the $xy$ plane area of the system constant while allowing fluctuations in the $z$ axis. The temperature was kept at 310 K using a Langevin thermostat with a damping constant of 1 ps$^{-1}$.

All simulations used the CHARMM36m forcefield for proteins[91] and the TIP3P-charmm model for water molecules[86]. Bonds between hydrogen and heavy atoms were constrained using the SHAKE[92,93] and SETTLE[94] algorithms. The simulation time step was set at 4.0 fs, enabled by the application of the hydrogen mass repartition scheme[95,96], which redistributed mass from heavy atoms in the solute to their bonded hydrogen atoms. Non-bonded interactions were calculated with a 12 Å cut-off for short-range electrostatic interactions, while long-range electrostatic interactions were calculated every 8.0 fs using the particle mesh Ewald[97] algorithm with a 1 Å grid spacing. All energy minimization runs were conducted using a CPU-only version of NAMD v.3.0.1, while NPT production simulations were performed using the GPU-resident NAMD v.3.1 alpha 2 (ref. 90).

## IN–CA interaction contact analysis
Residue-level contacts between IN and CA were analysed across all simulation trajectories using custom Tool Command Language scripts in VMD[88]. For each frame of a simulation trajectory, the coordinates were evaluated to assign contacts between IN and CA residues. Contacts were evaluated using a 3.5 Å distance threshold. Contact occupancies were calculated as the fraction of frames in which a given pair of residues remained within this distance threshold, relative to the total number of frames in the trajectory. A contact with 100% occupancy indicates that two residues remained within 3.5 Å of each other throughout the entire trajectory, while an occupancy of 0% indicates that no contacts were identified at any point. Due to the symmetry of the IN octamer, the four IN–CA contact regions were present in duplicate (one in each IN tetramer). Thus, contact occupancies reported in Extended Data Fig. 8 represent the average across symmetry-equivalent IN–CA contact regions in the IN octamer and across all simulation replicas. In total, over 200,000 frames were analysed to calculate contact occupancies, with each frame representing 20 ps intervals in the simulation.

## Virology
Synthetic fragments (IDT) carrying CA and SP1 changes were incorporated into SpeI/ApaI-digested pNLX.Luc.R(-)ΔAvrII DNA[98] using the NEBuilder HiFi DNA Assembly kit. IN changes were similarly made using AgeI/PflMI-digested plasmid. Plasmid pLR2P-vprIN[99] was digested with BamHI and XhoI to incorporate synthetic IBD fragments (LEDGF/p75 residues 347–471) downstream of Vpr. All newly made plasmids were verified by restriction enzyme digestion and whole-plasmid sequencing (Plasmidsaurus). Plasmid DNAs expressing single-round IN mutant luciferase viruses H12N, V165A and D64N/D116N were as previously described[20].

HEK293T cells (ATCC, CRL-3216), which were used to produce viruses by transfection and also for infection assays, were cultured as described above for Lenti-X HEK293T cells. Cells were confirmed negative for mycoplasma contamination by PCR using PHOENIXDX MYCOPLASMA MIX (Procomcure Biotech, PCCSKU15209). For reverse transcription and infectivity assays, around 10$^6$ cells plated the previous day into six-well plates were co-transfected with around 2 µg total DNA consisting of pNLX.Luc(R-)ΔAvrII and vesicular stomatitis virus G envelope (VSV-G) expressor at a 6:1 mass ratio using PolyJet DNA transfection reagent. For Vpr-IBD and Vpr-NG-IN complementation experiments, the mass ratios were 5.1:0.9:1 (pNLX.Luc(R-)ΔAvrII:Vpr-IBD:VSV-G) and 3:1:1 (pNLX.Luc(R-)ΔAvrII:Vpr-mNeonGreen-IN:VSV-G). After 2 days, the supernatants, precleared at 500$g$ for 5 min, were filtered through 0.45-µm filters and treated with 2 U µg$^{-1}$ Turbo DNase for 1 h at 37 °C to degrade residual plasmid DNA. The concentration of p24 was assessed using an ELISA kit from Advanced Bioscience Laboratories. For immunoblotting and TEM, transfections were scaled up (around 10$^7$ cells plated the previous day in 15 cm dishes) to 30 µg total plasmid DNA. The resulting 0.45 µm filtered supernatants were pelleted by ultracentrifugation using a Beckman SW32-Ti rotor at 26,000 rpm for 2 h at 4 °C.

Infections, normalized by p24 to 0.25 pg per cell, were performed with around $10^5$ HEK293T cells per 24-well plate well. At 6 h after infection, the medium was replaced with fresh DMEM. At 48 h after infection, cells were collected, washed twice with PBS and lysed using passive lysis buffer as recommended by the manufacturer (Promega). The luciferase activity, assessed as relative light units per µg of total protein in the cell extracts (RLU per µg), was determined as previously described[100]. Infections for reverse transcription measurements were washed after 1.5 h to remove virus and included dimethyl sulfoxide (DMSO) or 20 µM efavirenz to control for potential plasmid carryover from transfection. Genomic DNA was isolated at 8 h after infection and quantitative PCR was performed as previously described[101]. DNA quantities were normalized by spectrophotometry and reverse transcript levels were normalized to the WT after subtracting $C_t$ values for efavirenz-treated cultures from matched DMSO-treated samples.

For immunoblotting, volume-normalized virus pellets resuspended in protein sample buffer were separated by electrophoresis on Bolt 4–12% Bis-Tris Plus gels and transferred to polyvinylidene difluoride membranes. Membranes blocked for 1 h at room temperature were probed overnight at 4 °C in 12.5% non-fat dry milk containing a 1:1,000 dilution of mouse anti-CA monoclonal AG3.0 (obtained from the NIH HIV Reagent Program, Division of AIDS, NIAID, NIH and contributed by M.-C. Gauduin) or 1:2,000 dilution of inhouse rabbit anti-IN polyclonal antibodies[102]. The next day, the membranes were probed with secondary antibodies conjugated to horseradish peroxidase (goat anti-rabbit, Agilent, P0448, 1:2,000; rabbit anti-mouse, Dako, P0161, 1:4,000), treated with enhanced chemiluminescence detection reagents and imaged on the ChemiDoc MP imaging system. The relative intensity of IN signals was normalized to CA signals using ImageJ[103].

For TEM, virus pellets were resuspended in 1 ml fixative solution (2.5% glutaraldehyde, 1.25% paraformaldehyde, 0.03% picric acid, 0.1 M sodium cacodylate, pH 7.4) and incubated at 4 °C overnight. The preparation and sectioning of fixed virus pellets were performed at the Harvard Medical School Electron Microscopy core facility as previously described[104]. Sections (50 nm) were imaged at ×20,000 to ×30,000 magnification using the JEOL 1200EX or Tecnai G2 Spirit BioTWIN transmission electron microscope operated at 80 kV. Approximately 40 micrographs were recorded per sample, and viruses were manually counted and assigned on the basis of one of seven different phenotypes (Fig. 4d).

### Statistics
Individual virus-based experiments (Fig. 4 and Extended Data Fig. 4b) had technical duplicate samples and the results are presented as the mean ± s.d. of at least $n = 3$ independent experiments. Statistical analyses, conducted in GraphPad Prism v.10.6.0, compared mutant viral responses with the WT (Fig. 4) or matched sets of Vpr-complemented viruses (Extended Data Fig. 4b) using one-way or two-way ANOVA with Holm–Šídák correction for multiple comparisons. Statistical analyses were omitted for samples that lacked $n = 3$ independent experiments (Extended Data Fig. 7b). Owing to practical constraints, investigators were not blinded to sample identity and outcome assessment.

### Reporting summary
Further information on research design is available in the Nature Portfolio Reporting Summary linked to this article.

### Data availability
The cryo-EM and cryo-ET reconstructions were deposited at the EMDB under accession codes EMD-54067, EMD-54068, EMD-54070, EMD-54071 and EMD-55409, and refined coordinates at the PDB under accession codes 9RMU and 9RMX (Extended Data Table 1). Raw tilt-series data are available in EMPIAR under accession code EMPIAR-13078. HDX–MS data are available at the ProteomeXchange under identifier PXD070910.

All input parameters, structures and configuration parameters necessary for performing the MD simulations presented in this work are freely available at Zenodo[105] (https://doi.org/10.5281/zenodo.15866690). Structures used as starting models or for comparative analyses in this work are freely available at the PDB: 1K6Y, 2B4J, 5TC2, 6ES8, 6SKK, 7ASH, 7Z1Z, 8A1P and 9C9M. Source data are provided with this paper.

### Code availability
Scripts used for analysis of the MD simulation trajectories can be found at Zenodo[105] (https://doi.org/10.5281/zenodo.15866690).

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

**Acknowledgements** We thank H.-G. Kräusslich for the gift of pNLC4-3 IND64N/D116N tatΔ33–64bp construct and Z. Ambrose for pVpr-mNeonGreen-IN. This work was funded by US National Institutes of Health grants U54AI170791 (P.C., A.N.E., J.R.P. and P.Z.), R01AI178846 (J.R.P.) and R21AI184080 (P.Z.). The laboratories of P.C. and I.A.T. are supported by the Francis Crick Institute, which receives its core funding from Cancer Research UK (CC2058 and CC2029), the UK Medical Research Council (CC2058 and CC2029) and the Wellcome Trust (CC2058 and CC2029). This work was also supported by the Wellcome Discovery Award 311427/Z/24/Z (to P.Z.) and MRC grant MR/Z504312/1 (to G.Z.). This work used the Delta system at the National Center for Supercomputing Applications and the Stampede3 supercomputer at the Texas Advanced Computing Center through allocation MCB-170096 from the Advanced Cyberinfrastructure Coordination Ecosystem: Services & Support (ACCESS) program, which is supported by National Science Foundation grants 2138259, 2138286, 2138307, 2137603 and 2138296.

**Author contributions** M.R.S. and P.C. refined the cryo-EM structures. F.C. prepared samples and analysed IN-(GA)₁₈ assemblies. P.C., A.N., F.C. and G.Z. analysed cryo-ET data. P.C. performed subtomogram averaging with help from A.N. and G.Z.; Z.L. and A.N.E. carried out the virology studies. J.S.R. and J.R.P. conducted MD. J.H., F.C. and P.Z. prepared and vitrified native HIV-1 cores. M.R.S., N.J.C., L.M., J.S. and Z.Z. carried out in vitro studies, including all BLI experiments. M.R.S., E.P., A.N. and P.C. acquired cryo-EM data. M.R.S. and I.A.T. conducted SEC–MALLS experiments. S.M. and M.S. acquired and analysed HDX–MS data. J.S.R. and J.R.P. conducted MD calculations and data analysis. P.C. and A.N.E. wrote the manuscript with input from all of the authors.

**Funding** Open Access funding provided by The Francis Crick Institute.

**Competing interests** The authors declare no competing interests.

**Additional information**
**Correspondence and requests for materials** should be addressed to Juan R. Perilla, Alan N. Engelman or Peter Cherepanov.

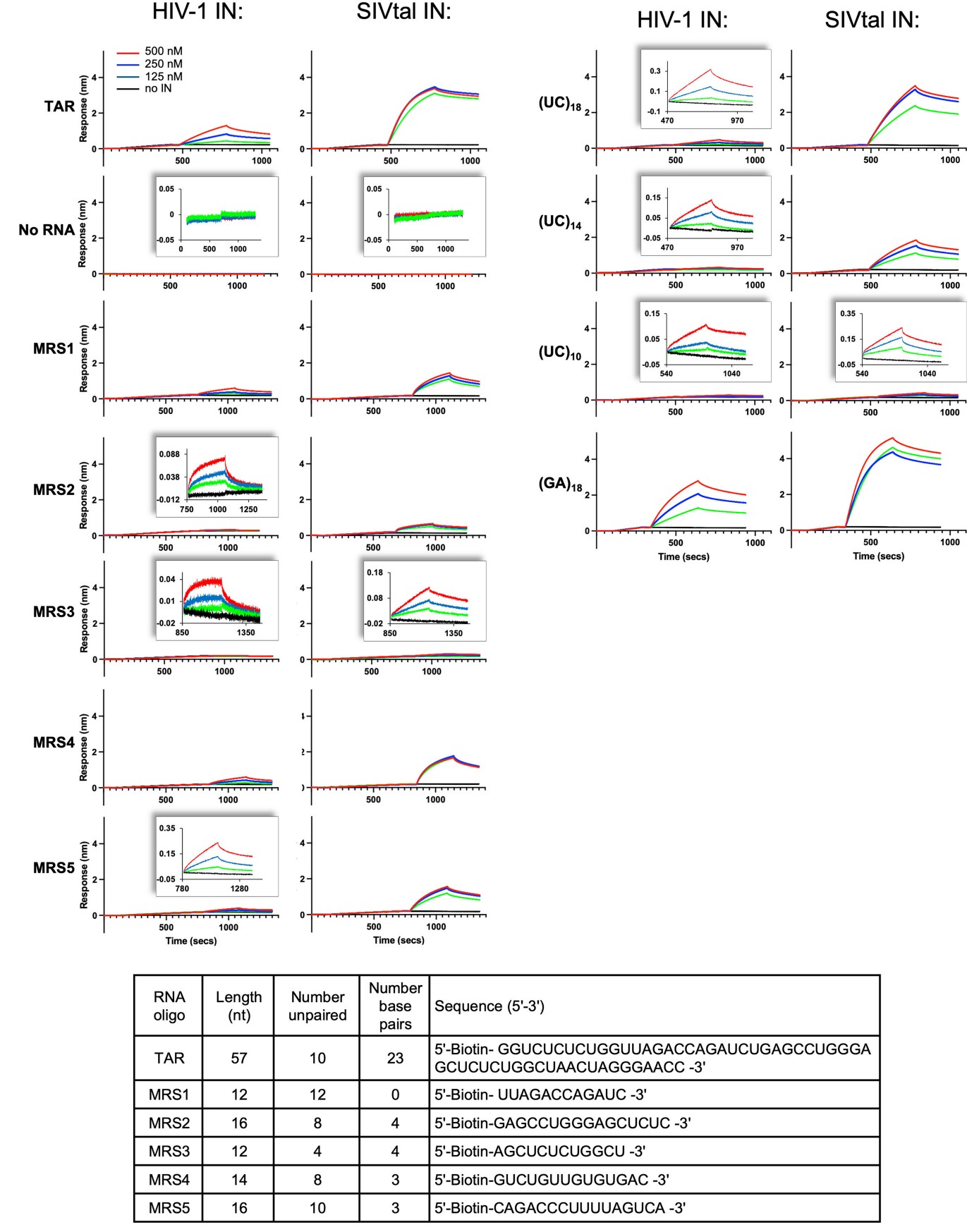

| RNA oligo | Length (nt) | Number unpaired | Number base pairs | Sequence (5'-3') |
|---|---|---|---|---|
| TAR | 57 | 10 | 23 | 5'-Biotin- GGUCUCUCUGGUUAGACCAGAUCUGAGCCUGGGAGCUCUCUGGCUAACUAGGGAACC -3' |
| MRS1 | 12 | 12 | 0 | 5'-Biotin- UUAGACCAGAUC -3' |
| MRS2 | 16 | 8 | 4 | 5'-Biotin-GAGCCUGGGAGCUCUC -3' |
| MRS3 | 12 | 4 | 4 | 5'-Biotin-AGCUCUCUGGCU -3' |
| MRS4 | 14 | 8 | 3 | 5'-Biotin-GUCUGUUGUGUGAC -3' |
| MRS5 | 16 | 10 | 3 | 5'-Biotin-CAGACCCUUUUAGUCA -3' |

**Extended Data Fig. 1** | See next page for caption.

**Extended Data Fig. 1 | Interaction of HIV-1 and SIVtal INs with RNA oligonucleotides in vitro.** The graphs represent biolayer interferometry (BLI) sensorgrams of HIV-1 and SIVtal IN binding to biotinylated ribooligonucleotides immobilized on streptavidin sensors. IN was assayed at 500, 250, 125 nM: red, blue, and green traces, respectively; black traces represent reference curves recorded in assay buffer lacking IN. Comparatively low responses are enlarged as insets. The identity of ribooligonucleotides (HIV-1 TAR, MRS1-5, specified oligo-UC, or (GA)$_{18}$) used in each pair of measurements is indicated. The table to the bottom shows the length, the number of unpaired nucleotides, the number of predicted base-pairs, based on structure prediction using RNAfold web server (http://rna.tbi.univie.ac.at/cgi-bin/RNAWebSuite/RNAfold.cgi)[106]. TAR and MRS1-5 ribooligonucleotides were biotinylated at 5′ ends, while oligo-UC and (GA)$_{18}$ carried biotin at 3′ ends.

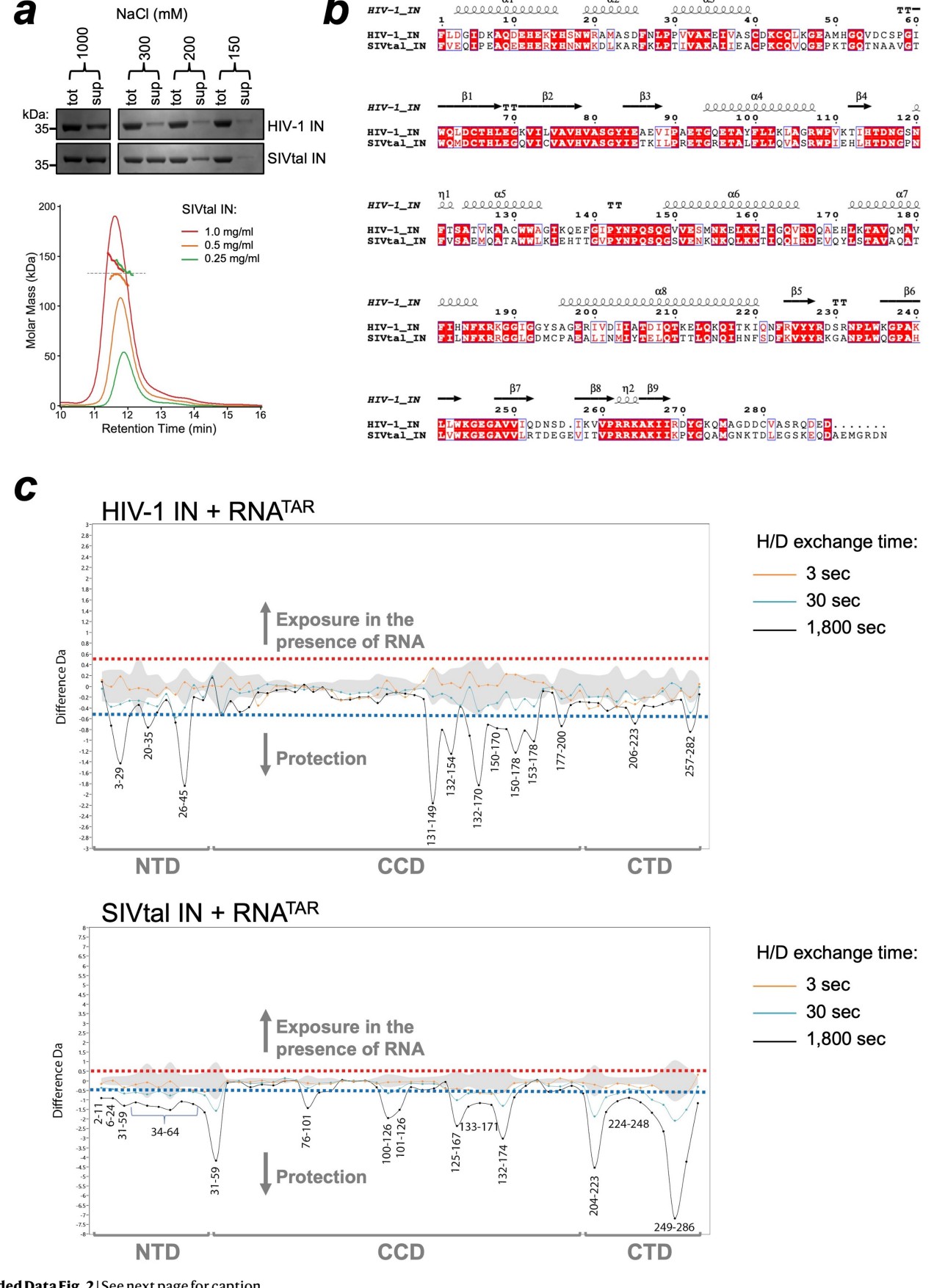

**Extended Data Fig. 2** | See next page for caption.

**Extended Data Fig. 2 | Biochemical properties and hydrogen-deuterium exchange of SIVtal and HIV-1 INs the presence of RNA. a**, *Top:* Solubility comparison of SIVtal and HIV-1 INs in the absence of detergent at varying NaCl concentrations. Proteins were diluted to 0.15 mg/mL in 20 mM HEPES-NaOH (pH 8.0), 1 mM DTT, and 1000, 300, 200, or 150 mM NaCl, as indicated, and incubated on ice for 10 min. Total (tot) and supernatant (sup) fractions after centrifugation at $15,000 \times g$ were analysed by SDS–PAGE. Gels were stained with Coomassie Brilliant Blue G-250. Increased solubility of SIVtal IN relative to HIV-1 IN was consistently observed in multiple (n > 3) independent repeats. Concordantly, unlike HIV-1 IN, SIVtal IN can be purified in the absence of CHAPS (see Methods). *Bottom:* Size-exclusion chromatography coupled with multiangle laser light scattering (SEC–MALLS) analysis of SIVtal IN. Protein was injected at 1.0, 0.5, and 0.25 mg/mL, with corresponding elution profiles shown in red, orange, and green, respectively. Molecular mass estimates from individual peaks (coloured dots) and the global average across all conditions (grey dashed line; 136 ± 9 kDa) are shown. The theoretical mass of the SIVtal IN tetramer is 133.6 kDa. **b**, Amino acid sequence alignment of HIV-1 and SIVtal INs. Residue identities are highlighted in bold white on a red background; residues with similar properties are shown in red text. Secondary structure elements of HIV-1 IN - α-helices (α), β-strands (β), $3_{10}$ helices (η), and turns (T) - are indicated above the alignment and numbered sequentially according to their position in the primary sequence. **c**, Differential deuterium uptake of peptic peptides from HIV-1 (top) and SIVtal (bottom) IN during incubation with $D_2O$, in the absence or presence of RNA$^{TAR}$; data points and trend lines are coloured red, blue, and black for the 3 s, 30 s, and 1,800 s time points, respectively. Peptides measured by mass spectrometry are identified by their residue spans within the IN sequence. Red and blue dashed lines represent thresholds for statistically significant increases (enhanced exchange) or decreases (protection) in deuterium uptake in the presence of RNA, respectively. Domain boundaries for the N-terminal domain (NTD), catalytic core domain (CCD), and C-terminal domain (CTD) are indicated below each profile. The data indicate extensive regions of both HIV-1 and SIVtal INs are protected from exchange in the presence of RNA, consistent with widespread interactions. The grey area between the dotted red and blue lines represents the standard deviation of the deuterium exchange data, indicating the variability or uncertainty in the measurements across replicates and charge states for each peptide. For gel source data, see Supplementary Fig. 1.

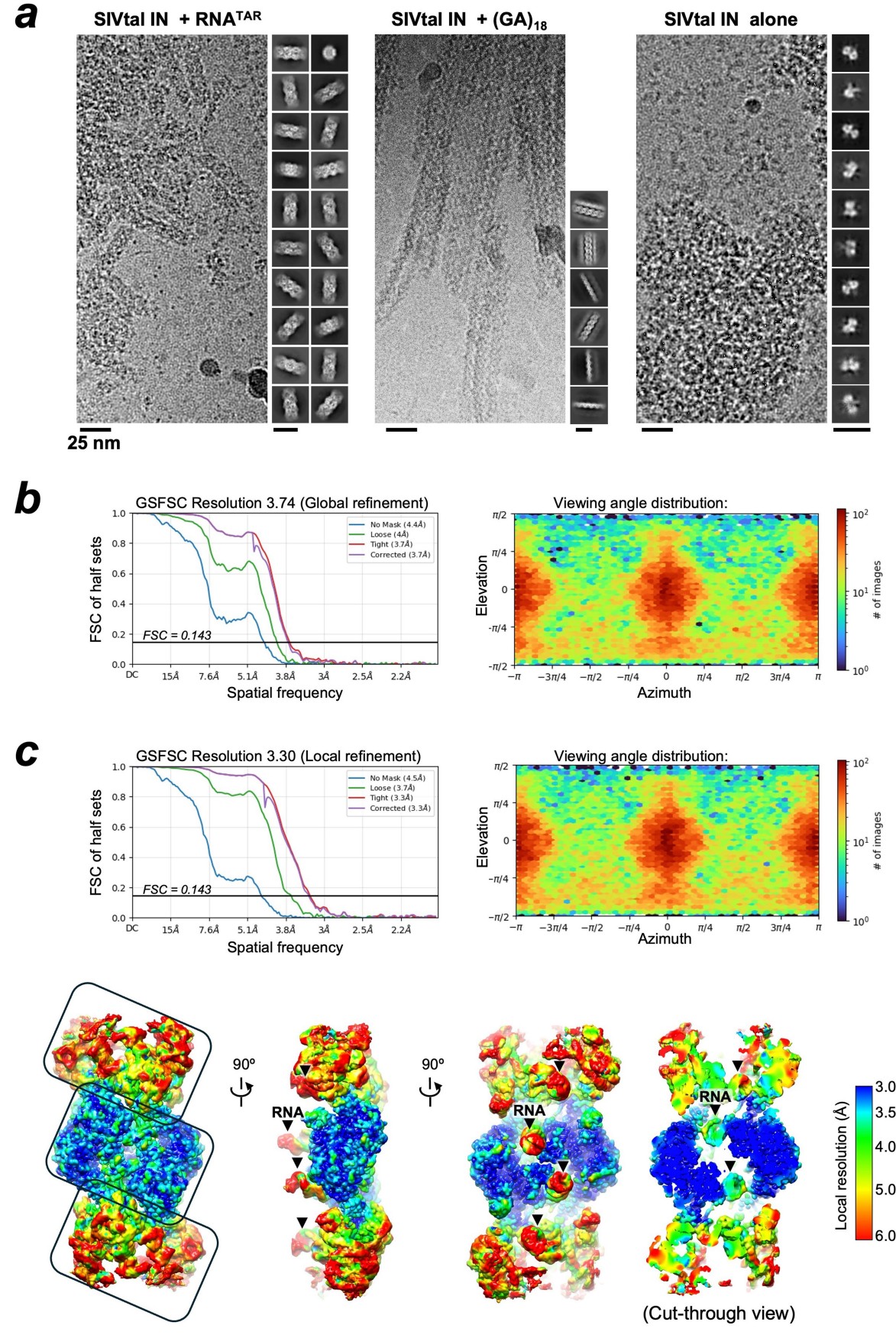

**Extended Data Fig. 3** | See next page for caption.

**Extended Data Fig. 3 | Single-particle cryo-EM analysis of SIVtal IN in the presence and absence of RNA. a**, Representative electron micrographs and 2D class averages of the SIVtal IN in the presence of RNA[TAR] (*Left*), oligo-GA ribooligonucleotide (*Middle*), and in the absence of RNA (*Right*). Assembly of SIVtal IN complexes with TAR and GA18 RNA complexes were repeated 3 times (on different days), with filaments observed each time. Similarly, SIVtal IN in the absence of RNA was imaged 3 independent times, and no filaments were observed by cryo-EM. Scale bars are 25 nm. **b**, Cryo-EM reconstruction of the SIVtal IN-RNA[TAR] complex by global refinement of both sister filaments. *Left*: half-map Fourier shell correlation (FSC) without mask (blue line), with loose mask (green line), tight mask (red line), and corrected optimized mask (purple line). The FCS threshold of 0.143 (horizontal black line) corresponds to a global resolution of 3.7 Å. *Right*: viewing angle distribution of assigned to individual particle images that contributed to the final 3D reconstruction. **c**, Local refinement focused on a single octamer unit within one filament. *Left*: half-map FSC without mask (blue line), with loose mask (green line), tight mask (red line), and corrected optimized mask (purple line). The FCS threshold of 0.143 (horizontal black line) corresponds to a global resolution of 3.3 Å. *Right*: viewing angle distribution of assigned to individual particle images that contributed to the final 3D reconstruction. *Bottom*: orthogonal views of the cryo-EM map in surface representation, coloured by local resolution. Rounded boxes indicate locations of IN octamer repeat units.

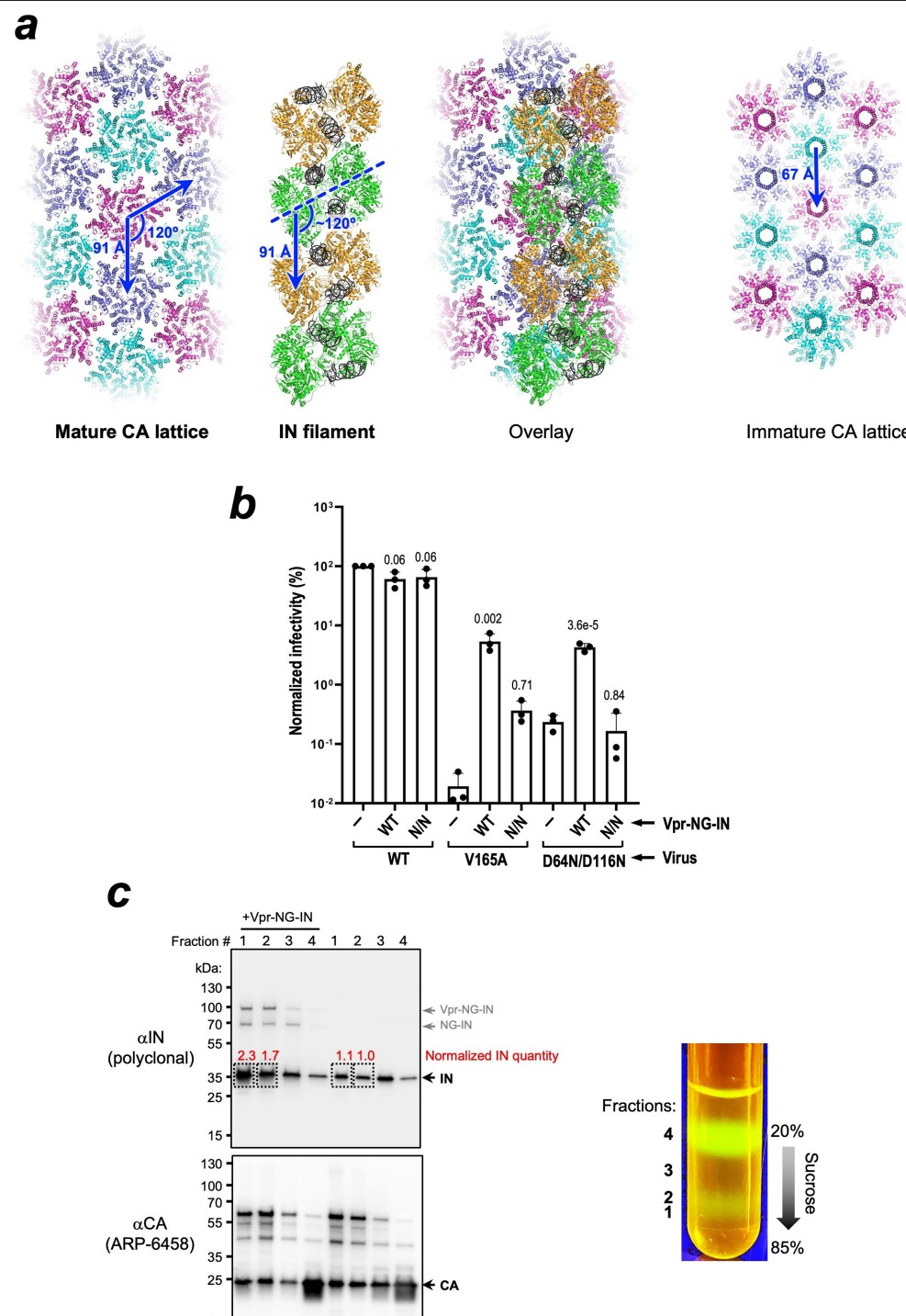

**Extended Data Fig. 4** | See next page for caption.

**Extended Data Fig. 4 | Viral polymer geometries, Vpr-NG-IN validation, and HIV-1 core isolation. a**, From *left* to *right*, panels show cartoon representations of the mature CA lattice (PDB entry 6ES8[107]), the SIVtal IN filament observed in the cryo-EM structure with RNA[TAR], the overlay of the two assemblies, and the immature Gag lattice (assembled from PDB entry 7ASH[108]). Neighbouring CA hexamers and IN octamers are shown in alternating colours for clarity. The spacing between unit cells in the mature CA lattice (hexamers) and in the IN filament (octamers) is identical - 91 Å, as indicated by arrows. This same distance separates adjacent IN tetramer pairs within each octameric repeat. Furthermore, the IN octamers are inclined by ~120° relative to the filament axis, making the IN filament geometrically congruent with the mature CA lattice, as can be seen when the two structures are overlayed. By contrast, the immature HIV-1 CA lattice features a considerably shorter repeat length (67 Å) that is incongruent with the geometry of the IN-RNA filament. **b**, Vpr-NG-IN functionality. Activities of indicated virus constructs following infection of cells with equivalent levels of viral p24. Results (mean ± SD for n = 3 independent experiments, with each experiment harbouring technical duplicate samples) were percent-normalized to WT HIV-Luc co-transfected with Vpr-Stop (–) in place of Vpr-NG-IN. V165A and D64N/D116N (N/N) are class II and class I IN mutant viruses, respectively. Statistical analysis (P values versus matched Vpr-Stop virus indicated above bars) was performed by ordinary one-way ANOVA with Holm-Šídák's correction for multiple comparisons. **c**, Isolation of HIV-1 cores. HIV-1 particles were produced with or without co-transfection of the N/N Vpr-mNeonGreen (NG)-IN construct. Viral cores were isolated by ultracentrifugation through a layer of non-ionic detergent and a 20–85% sucrose gradient. The photograph (right) shows the gradient under blue light illumination, highlighting two major fluorescent fractions: a light fraction at the top (low molecular weight material) and a heavy fraction at the bottom (containing viral cores). The gradient was fractionated (fractions 1–4, as indicated at left), and each fraction was analysed by Western blotting using polyclonal anti-IN (top left panel) and anti-CA (bottom left panel) antibodies. Migration positions of CA, IN, Vpr-NG-IN, and NG-IN proteins are marked to the right; molecular weight markers are indicated at left. Intensities of IN bands, normalized to the corresponding CA signal in each fraction, are shown in red. In addition to the canonical HIV-1 protease site engineered between Vpr and NG (Ile-Arg-Lys-Val-Leu-Phe-Leu-Asp-Gly-Glu), the Vpr-NG-IN fusion construct used in this work includes a putative cleavage site between NG and IN (Ser-Gly-Glu-Phe-<u>Phe-Leu-Asp-Gly-Glu</u>, with the underlined residues marking the native N-terminus of HIV-1 IN). Both PROSPERousPlus (http://prosperousplus.unimelb-biotools.cloud.edu.au/)[109] and HIVcleave (http://www.csbio.sjtu.edu.cn/bioinf/HIV/)[110] tools predict this sequence to be a high-confidence HIV-1 protease substrate, consistent with the accumulation of free mature IN, rather than the larger Vpr-NG-IN or NG-IN fusions, in viral cores. Comparative immunoblotting of cores and without Vpr-NG-IN was done twice with similar results. For gel source data, see Supplementary Fig. 1.

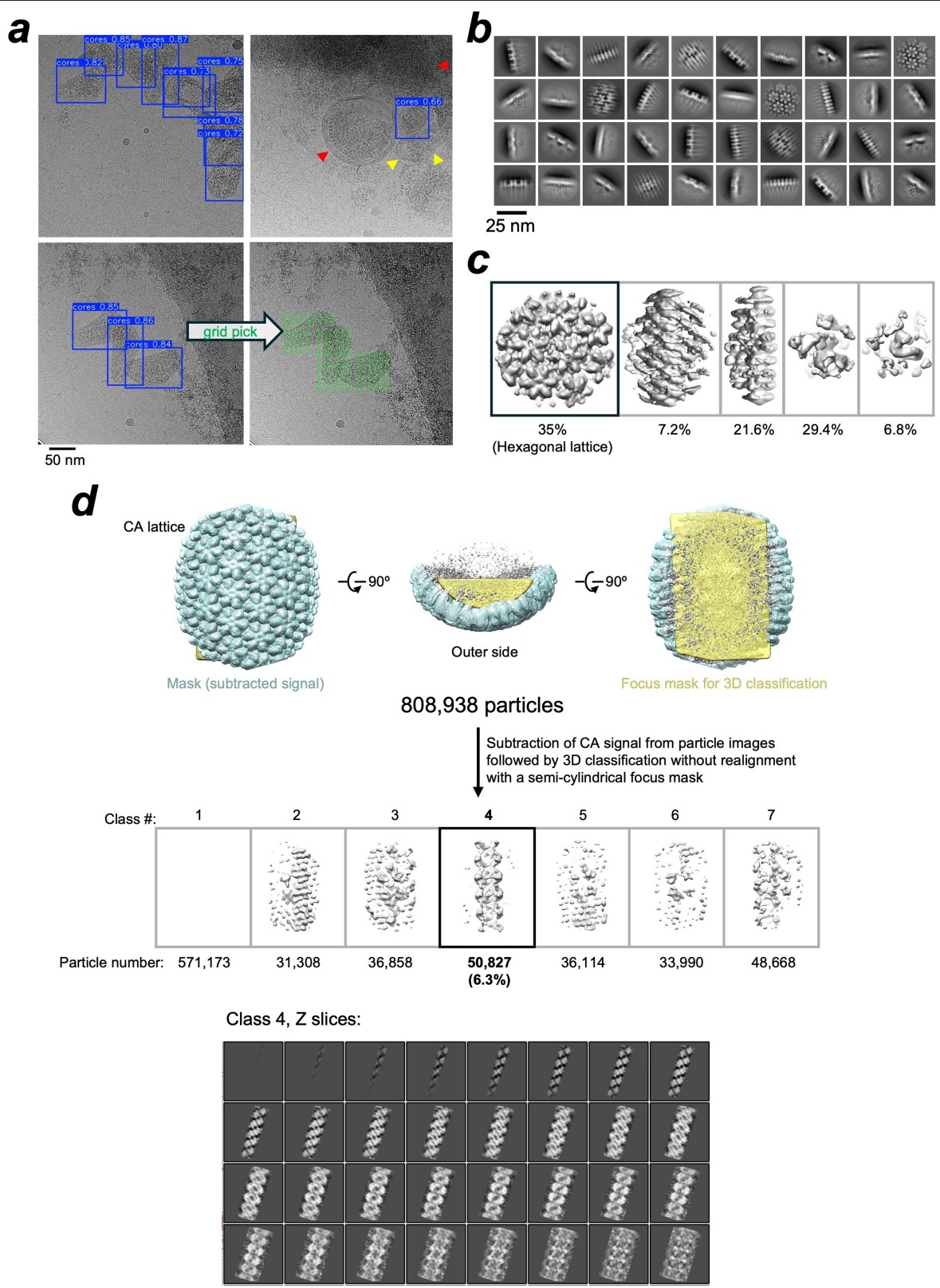

**Extended Data Fig. 5 |** See next page for caption.

**Extended Data Fig. 5 | Imaging HIV-1 cores produced with co-expressed Vpr-NG-IN, particle picking, and classification. a**, Example micrographs with cores identified by YOLOv11 (blue boxes; the confidence value for each identified core is shown above the corresponding box). The right image in the top row includes immature particles (red arrowheads) as well as mature cores not detected by the initial screen (yellow arrowheads). The presence of immature cores is consistent with western blot analysis of sucrose gradient fractions (Extended Data Fig. 4c). Initially, particles within each YOLOv11 box were picked using an evenly spaced grid; an example of 12-nm grid picking is shown in the bottom right panel (green circles). Note that YOLOv11 boxes were used both to generate grids for initial picking and to prune Gautomatch results; however, full micrograph areas were picked using Topaz. Scale bar is 50 nm. **b**, 2D class averages from areas picked within native cores. Scale bar is 25 nm. **c**, Particles purified by 2D classification were subjected to 3D classification via ab initio reconstruction (in 5 classes) followed by heterogenous refinement in cryoSPARC. The class on the left represents a fragment of mature hexagonal CA lattice. The fraction of input particles contributing to each 2D class is shown below the corresponding average. **d**, 3D classification with CA signal subtraction to resolve luminal structures in HIV-1 cores. Initial 3D reconstruction of a CA lattice fragment containing 16 complete CA hexamers was generated from 808,938 particles. The resulting map (shown in grey in three orthogonal views, with luminal and outer surfaces indicated) was used to define a focused mask on the CA lattice (semi-transparent cyan surface). Using Relion, the CA signal within this mask was subtracted from each particle image. The remaining signal was then subjected to 3D classification without particle realignment, using a semi-cylindrical mask (semi-transparent yellow surface) focused on the luminal space beneath the CA lattice. A low-pass filtered version of the initial map (filtered to 40 Å) served as the starting model. Among the seven resulting classes, one (class 4, comprising 6.3% of input particles) revealed a distinct linear assembly. The deck of Z-axis slices through this class average are shown at the bottom.

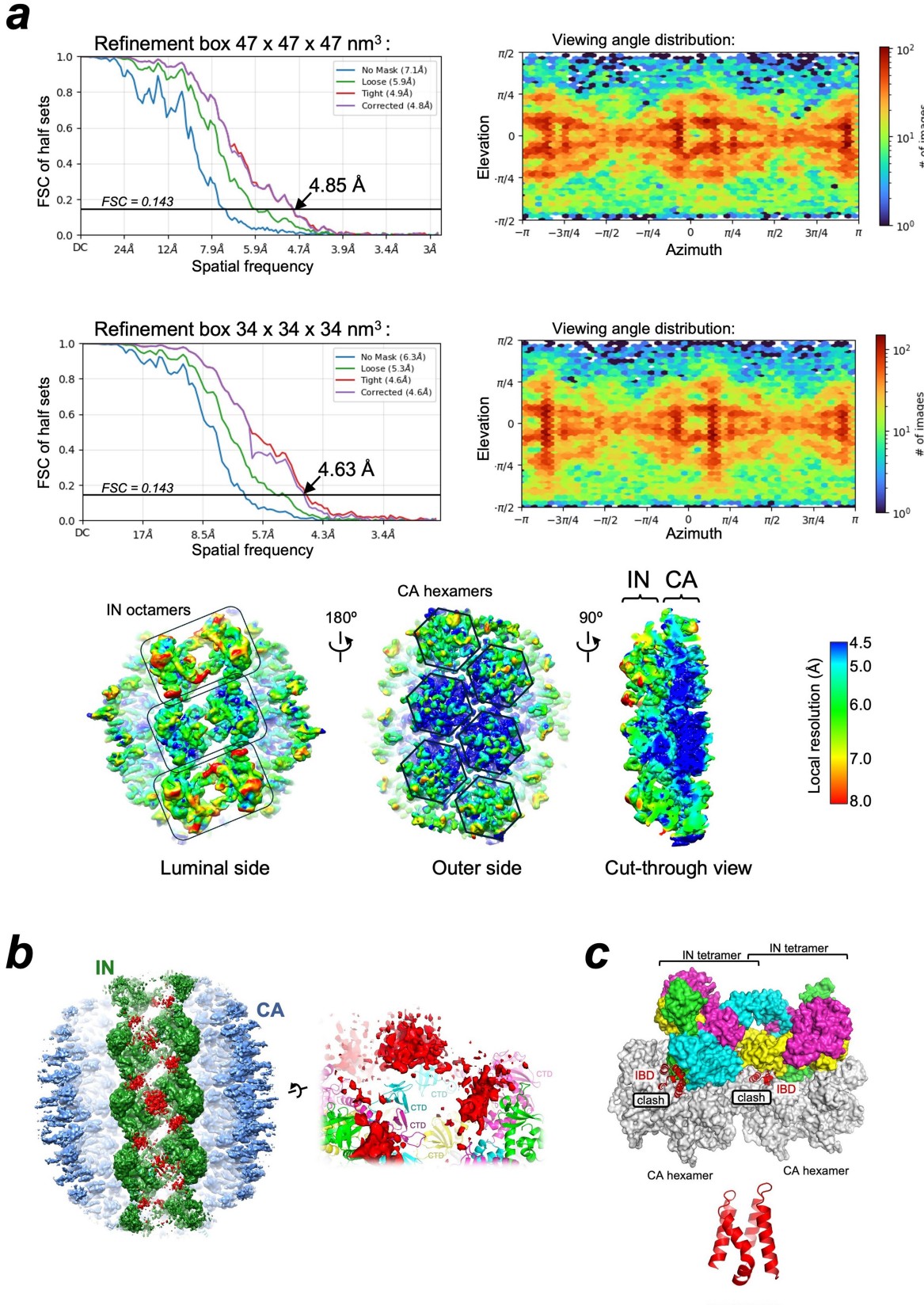

**Extended Data Fig. 6** | See next page for caption.

**Extended Data Fig. 6 | Final cryo-EM reconstructions of HIV-1 CA lattice with associated IN filaments. a**, Two reconstructions were generated, enclosing volumes of 47 × 47 × 47 nm³ (top panels) and 34 × 34 × 34 nm³ (bottom panels). The smaller reconstruction was used for atomistic model building and real-space refinement, while the larger reconstruction, which captures more CA and IN repeat units, is suited for visualizing the overall assembly. Half-map FSC curves and particle orientation distributions obtained from non-uniform refinement in cryoSPARC are shown for both assemblies. The lower panels show the smaller cryo-EM map from three views: outer surface, luminal side, and an orthogonal cut-through view. The map is coloured by local resolution. For reference, individual CA hexamers and IN octamers are marked by hexagons and rounded rectangles, respectively. **b**, *Left:* Sharpened cryo-EM map of the CA–IN assembly, globally filtered to 4.7 Å resolution. CA and IN densities are coloured blue and green, respectively (as in Fig. 3a); residual density not accounted for by CA or IN is shown in red. *Right:* Cartoon representation of the atomistic model overlaid with unassigned density, which is concentrated around the C-terminal domains (CTDs) of IN, consistent with the presence of RNA. **c**, *Top:* Model of an IN octamer bound to CA hexamers (surface representation; coloured as in Fig. 3c) with LEDGF/p75 IN-binding domains (IBDs, red cartoon) docked onto IN. The model reveals steric clashes (highlighted) between the IBDs and the CA lattice, suggesting that IN binding to LEDGF/p75 and to the CA lattice are mutually exclusive. *Bottom:* Cartoon representation of the LEDGF/p75 IBD (PDB ID 2B4J[34]).

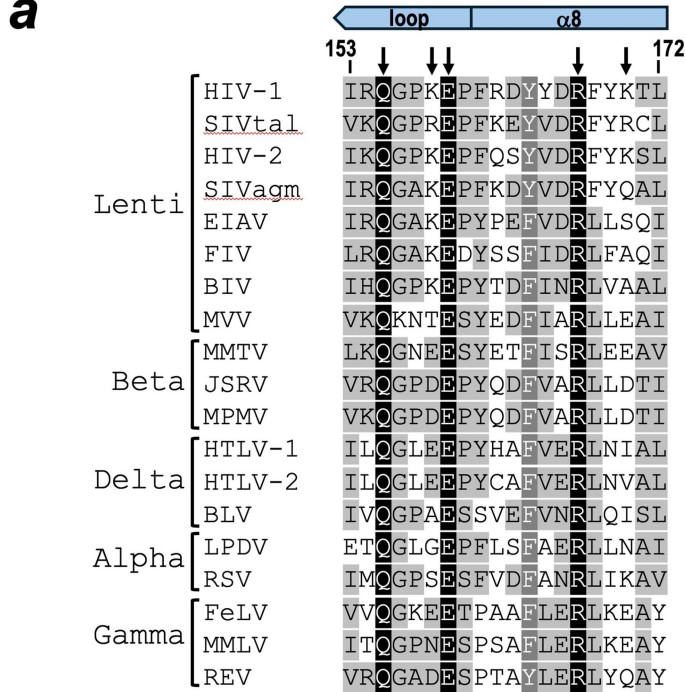

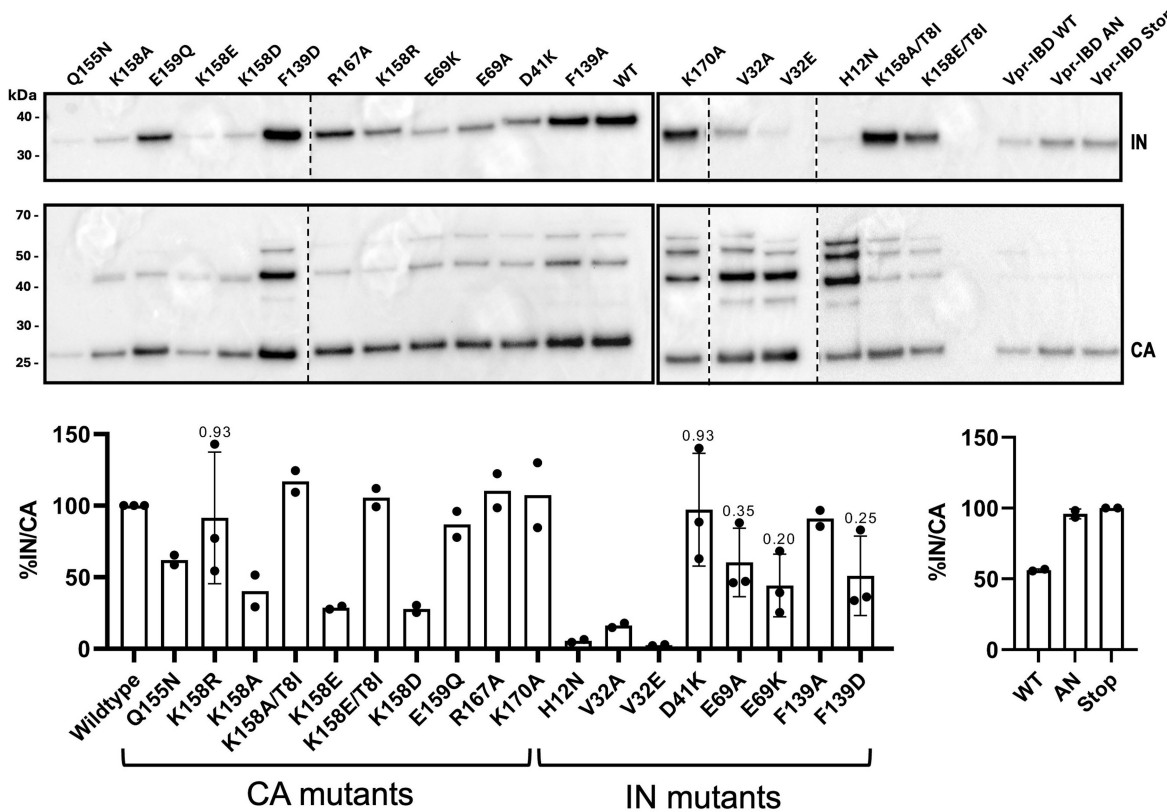

**Extended Data Fig. 7** | See next page for caption.

**Extended Data Fig. 7 | MHR alignment and virion protein content.**
**a**, Alignment of 19 retroviral MHRs, with residues comprising HIV-1 loop and α8 residues indicated above. Three invariant residues (these sequences are not conserved among epsilonretroviral or spumaretroviral proteins) are highlighted by black background, whereas dark grey highlights a physiochemically conserved position (Tyr/Phe). Light grey highlights amino acids that are physiochemically conserved in at least 8 of the sequences. Viral genera are indicated to the left; numbers refer to positions in HIV-1 CA. Downward arrows demarcate CA residues targeted for mutational analysis. Sequences were derived from the following GenBank accession codes/ viruses (strains indicated, when known): HIV-1 strain NL4-3, U26942.1; SIVtal strain 01CM8023, AM182197.1; HIV-2 ROD isolate, X05291.1; SIV from African green monkey (agm) strain tan-1, U58991.1; equine infectious anaemia virus (EIAV), M16575.1; feline immunodeficiency virus (FIV), M25381.1; bovine immunodeficiency virus (BIV), NC_001413.1; maedi-visna virus (MVV) strain kv1772, NC_001452.1; mouse mammary tumour virus (MMTV), NC_001503.1; Jaagsiekte sheep retrovirus (JSRV), NC001494.1; Mason-Pfizer monkey virus (MPMV), NC_001550.1; human T-lymphotropic virus 1 (HTLV-I), NC001436.1; HTLV-2, NC_001488.1; bovine leukaemia virus (BLV), K02120.1; lymphoproliferative disease virus (LPDV) strain 12/AR/2009, KC802224.1; Rous sarcoma virus (RSV) strain Prague C, J02342.1; feline leukaemia virus (FeLV) strain Rickard subgroup A, AF052723; Moloney murine leukaemia virus, J02255.1; reticuloendotheliosis virus (REV) strain HLJR0901, GQ415646.2. **b**, Representative immunoblots for CA and IN mutant viruses are shown; numbers to the left are mass standard positions in kDa and dashed lines demarcate separate immunoblot panels. IN-to-CA ratios for each virus are quantified below (mean for samples with n = 2 independent experiments, mean ± SD for n = 3 independent experiments). Statistics for n = 3 samples (P values versus the WT indicated above bars) were performed by ordinary one-way ANOVA with Holm-Šídák's correction for multiple comparisons. For gel source data, see Supplementary Fig. 1.

| Contact Region | IN residue | CA residue | Interaction occupancy (%) |
|---|---|---|---|
| CR1 | Ser255 | Lys158 | 46.0 |
| | Asp41 | Lys158 | 23.6 |
| | Serr39 | Lys158 | 23.2 |
| CR2 | Glu69 | Glu159 | 84.9 |
| | Glu69 | Lys158 | 59.7 |
| | Arg166 | Pro160 | 53.2 |
| | Arg166 | Lys158 | 52.9 |
| | Glu69 | Gly156 | 52.8 |
| | Glu69 | Pro157 | 48.0 |
| | Arg166 | Glu159 | 41.2 |
| | Gly70 | Arg167 | 30.0 |
| CR3 | Thr124 | Lys158 | 97.5 |
| | Thr124 | Pro157 | 49.7 |
| | Ser123 | Lys158 | 38.0 |
| | Thr122 | Lys158 | 35.3 |
| CR4 | Phe139 | Pro157 | 98.1 |
| | Thr122 | Pro157 | 90.2 |
| | Asn117 | Pro157 | 77.2 |
| | Glu138 | Lys158 | 47.0 |
| | Phe139 | Lys158 | 43.3 |

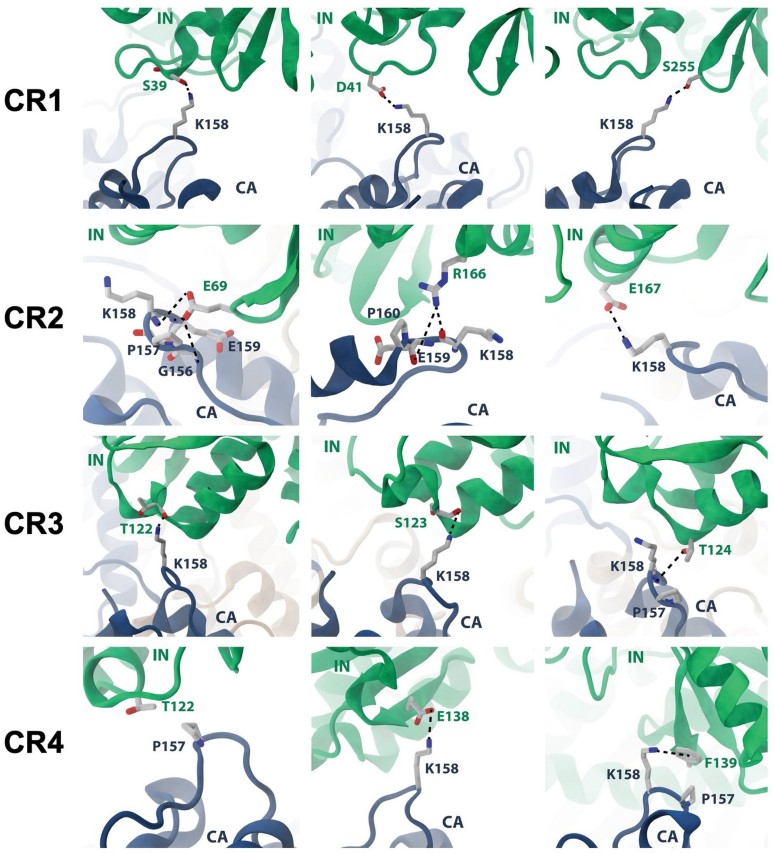

**Extended Data Fig. 8** | See next page for caption.

**Extended Data Fig. 8 | IN-CA contact region (CR) interactions observed in molecular dynamics simulations.** *Top:* Most frequent IN-CA contacts from 4 independent 1-µs MD simulations. *Bottom:* Representative frames from MD simulations illustrating IN-CA contacts. In CR1, CA Lys158 frequently formed hydrogen bonds with IN Ser255 (~46% occupancy) and engaged in more transient hydrogen bonds and salt bridge interactions with IN Ser39 and Asp41, respectively (~23% occupancy). Due to the low contact occupancy measured for these interactions, CR1 does not appear to be stabilized by a single dominant interaction. Instead, CA Lys158 transiently interacts with multiple nearby polar and charged IN residues. CR2 contains several charged amino acids in both CA and IN, resulting in higher occupancy interactions. Here, CA Lys158 formed salt bridge interactions with IN Glu69 (~60% occupancy) and Asp167 (~20% occupancy). While CA Glu159 typically forms an intramolecular salt-bridge with CA Arg167 in the absence of IN, it assumed novel interactions in the context of CR2. CA Glu159 is found in close contact with the backbone of IN Glu69 (~85% occupancy) and formed salt bridges with IN Arg166 (~41% occupancy). CA Arg167 also transiently interacted with the backbone of IN Gly70 (~30% occupancy). Additionally, the backbone of CA MHR residues (Gly156, Pro157, Pro160) frequently approached IN Glu69 and Arg166 sidechains. CR3 is stabilized via hydrogen bond interactions between CA Lys158 with IN Thr124 (~97% occupancy) and transiently with IN Ser123 (~38% occupancy) and Thr122 (~35% occupancy). CR4 features a stable hydrophobic interface between CA Pro157 and IN Phe139 (~98% occupancy), surrounded by IN Thr122 and Asp117. At this contact region, although CA Lys158 was in close contact with IN Phe139 throughout the simulations (~43% occupancy), the sidechain was not oriented to form cation-π interactions. Instead, it was primarily involved in salt-bridge formation with IN Glu138 (~47% occupancy). IN is coloured in green, the CA CTD in dark blue, and the CA NTD in gold. Interacting residues are shown in liquorice representation.

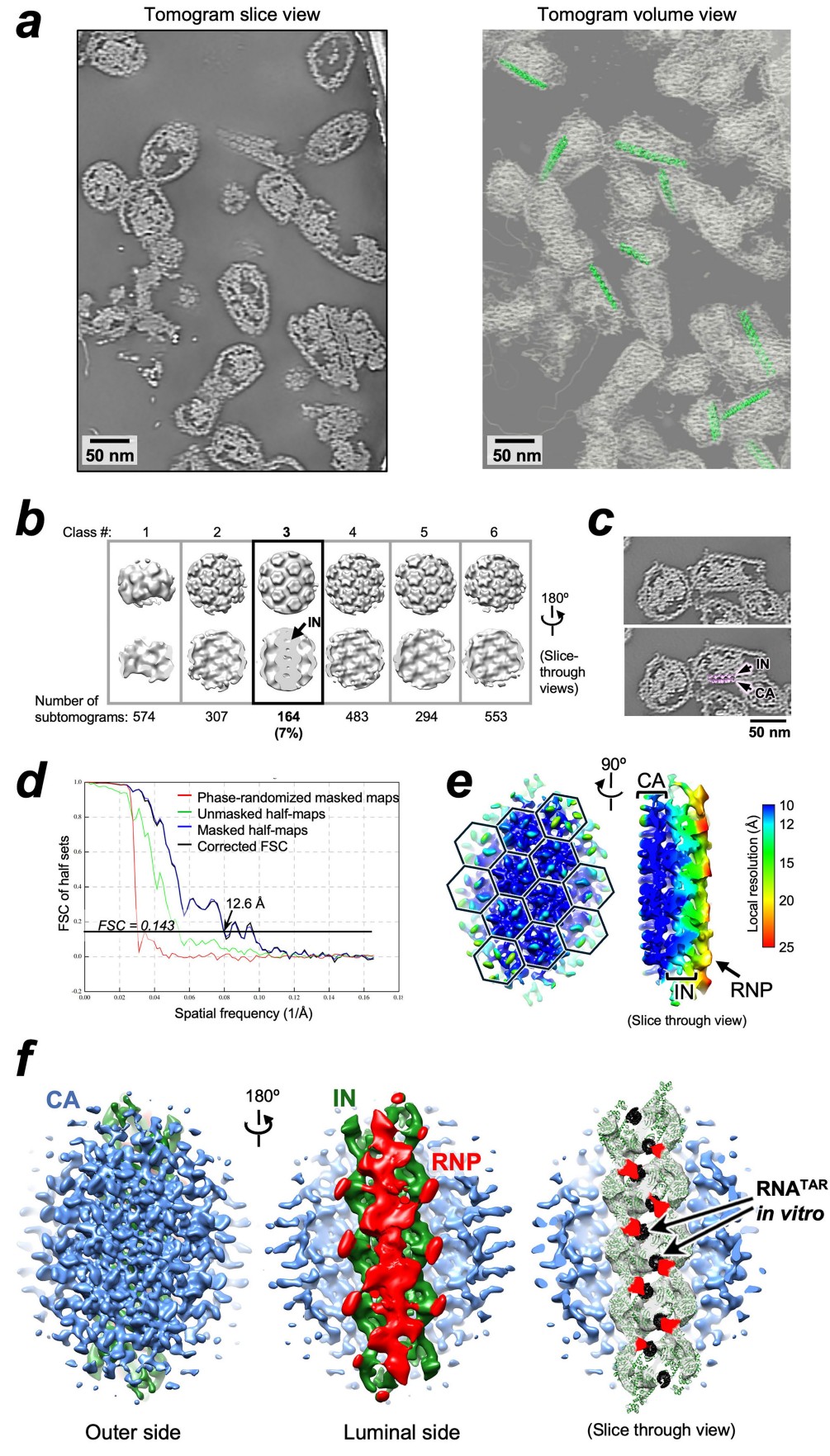

**Extended Data Fig. 9** | See next page for caption.

**Extended Data Fig. 9 | Cryo-ET analysis of native HIV-1 cores produced without oversupply of IN. a**, *Left*: a Z slice of a denoised tomogram. *Right*: the same tomogram displayed in volume mode (white) with IN octamers (green) using ChimeraX and ArtiaX. Note that identified IN filaments propagate along the length of cores. Scale bars are 50 nm. **b**, Results of 3D classification of subtomograms picked by template matching using CA lattice, as described in Methods. Class 3 comprising 7% of input particles features a well-defined linear polymer consistent with a chain of IN octamers. **c**, An example of a core particle with IN filament opportunely oriented along the XY plane of a tomogram. The bottom image shows overlay of the tomogram slice with the CA-IN assembly in pink; CA and IN indicated with arrowheads. **d**, Half-map Fourier shell correlation (FSC) without mask (green curve), mask (blue curve), and corrected optimized mask (black curve). FSC correlation between phase-randomized half-maps is shown by the red line. The FCS threshold of 0.143 (horizontal black line) corresponds to a global resolution of 12.6 Å. **e**, Cryo-ET map from two views: outer surface and an orthogonal cut-through view. The map is coloured by local resolution. For reference, individual CA hexamers are marked by hexagons on the left image. **f**, The cryo-ET map viewed from outer (*Left*) and luminal (*Middle and Right*) sides. Map features corresponding to CA, IN and RNP are coloured blue, green and red, respectively. The image on the right shows a slice-through view removing the overlaying RNP density, with map features corresponding to IN shown semi-transparent and fitted IN octamers as cartoons. Locations of RNA stem loops consistent with the in vitro reconstruction of the SIVtal IN-RNA$^{TAR}$ complex are indicated with black arrowheads on the middle IN octamer. Note that RNP penetrates the IN filament at the locations corresponding to RNA$^{TAR}$ (shown with atoms as black spheres) in the in vitro structure.

**Extended Data Table 1 | Cryo-EM/ET data collection, image processing and model refinement**

| Data collection | *In vitro* assembled SIVtal IN-RNA$^{TAR}$ complex (single-particle analysis) | | IN filament in HIV-1 cores with Vpr-NG-IN overexpression (single-particle analysis) | | IN filament in cores without Vpr-NG-IN (subtomogram averaging) |
|---|---|---|---|---|---|
| Microscope, operating voltage | Titan Krios G2, 300 keV | | Titan Krios G2, 300 keV | | Titan Krios G2, 300 keV |
| Detector | Falcon 4i | | Falcon 4i | | Falcon 4i |
| Magnification (nominal) | 130,000 | | 130,000 | | 81,000 |
| Micrograph pixel size (Å) | 0.95 | | 0.95 | | 1.56 |
| Underfocus range (nominal, μm) | 1.0 - 4.0 | | 1.5 - 3.5 | | 2.5 - 5.5 |
| Number of frames per movie | 1,674 (EER) | | 1,674 (EER) | | 6 (TIFF) |
| Total electron fluence (e/Å$^2$) | 41.0 | | 49.6 | | 104 |
| Automation software | FEI EPU | | FEI EPU | | FEI Tomography |
| Number of micrograph movies/tilt series used | 38,395 | | 20,214 | | 14 |
| **Reconstruction** | Double filament (global refinement) | Single filament (local refinement) | 47 x 47 x 47 nm$^3$ box | 34 x 34 x 34 nm$^3$ box | 58 x 58 x 58 nm$^3$ box |
| **EMDB ID** | **EMD-54067** | **EMD-54068** | **EMD-54070** | **EMD-54071** | **EMD-55409** |
| Software for tilt alignment and tomogram reconstruction | - | - | - | - | IMOD, WarpTools |
| Software for 2D classification | cryoSPARC | cryoSPARC | cryoSPARC | cryoSPARC | - |
| Software for 3D classification | cryoSPARC, Relion | cryoSPARC, Relion | cryoSPARC, Relion | cryoSPARC, Relion | Relion |
| Software for final reconstruction | cryoSPARC | cryoSPARC | cryoSPARC | cryoSPARC | Relion |
| Number of refined particles | 53,737 | 53,737 | 46,116 | 58,103 | 559 |
| Symmetry | *C2* | *C2* | *C2* | *C2* | *C2* |
| Pixel size in final reconstruction (Å) | 0.95 | 0.95 | 1.425 | 1.425 | 3.02 |
| Global resolution (FSC 0.143, Å) | 3.74 | 3.30 | 4.85 | 4.63 | 12.6 |
| Map resolution range (Å) | 3.5 - 7.0 | 3.0 - 6.0 | 4.5 - 8.0 | 4.5 - 8.0 | 10 - 20 |
| Map sharpening B factor | -86 | -103 | -137 | -185 | -500 |
| **Model refinement** | | | | | |
| **PDB ID** | | **9RMU** | | **9RMX** | |
| Software for real-space refinement | | Phenix | | Phenix | |
| Model composition | | | | | |
| Number of non-H atoms | | 18,876 | | 72,790 | |
| Number of protein residues | | 2,190 | | 9,257 | |
| Number of nucleotides | | 64 | | 0 | |
| Number of ligands (IP6, Zn) | | 0, 10 | | 8, 12 | |
| *B* factors (Å$^2$) | | | | | |
| Protein | | 99.9 | | 255 | |
| RNA | | 110.8 | | - | |
| Ligands | | 125.7 | | 305 | |
| Real-space correlation (CC$_{mask}$, CC$_{box}$, CC$_{peaks}$, CC$_{volume}$) | | 0.80, 0.65, 0.59, 0.79 | | 0.81, 0.70, 0.49, 0.81 | |
| R.m.s. deviations | | | | | |
| Bond lengths (Å) | | 0.002 | | 0.002 | |
| Bond angles (°) | | 0.443 | | 0.516 | |
| Model validation | | | | | |
| MolProbity score | | 1.53 | | 1.50 | |
| Clash score | | 4.53 | | 6.02 | |
| Poor rotamers (%) | | 1.23 | | 1.39 | |
| CaBLAM outliers (%) | | 2.70 | | 1.41 | |
| Ramachandran plot quality (%) | | | | | |
| Favored, Disallowed | | 96.47, 0 | | 97.73, 0 | |

Alan Engelman
Juan Perilla

# Reporting Summary

## Statistics

For all statistical analyses, confirm that the following items are present in the figure legend, table legend, main text, or Methods section.

| n/a | Confirmed | |
|-----|-----------|---|
| ☐ | ☒ | The exact sample size (*n*) for each experimental group/condition, given as a discrete number and unit of measurement |
| ☐ | ☒ | A statement on whether measurements were taken from distinct samples or whether the same sample was measured repeatedly |
| ☐ | ☒ | The statistical test(s) used AND whether they are one- or two-sided *Only common tests should be described solely by name; describe more complex techniques in the Methods section.* |
| ☒ | ☐ | A description of all covariates tested |
| ☐ | ☒ | A description of any assumptions or corrections, such as tests of normality and adjustment for multiple comparisons |
| ☐ | ☒ | A full description of the statistical parameters including central tendency (e.g. means) or other basic estimates (e.g. regression coefficient) AND variation (e.g. standard deviation) or associated estimates of uncertainty (e.g. confidence intervals) |
| ☐ | ☒ | For null hypothesis testing, the test statistic (e.g. $F$, $t$, $r$) with confidence intervals, effect sizes, degrees of freedom and *P* value noted *Give P values as exact values whenever suitable.* |
| ☒ | ☐ | For Bayesian analysis, information on the choice of priors and Markov chain Monte Carlo settings |
| ☒ | ☐ | For hierarchical and complex designs, identification of the appropriate level for tests and full reporting of outcomes |
| ☒ | ☐ | Estimates of effect sizes (e.g. Cohen's *d*, Pearson's *r*), indicating how they were calculated |

*Our web collection on statistics for biologists contains articles on many of the points above.*

## Software and code

Policy information about availability of computer code

| Data collection | Cryo-EM images were recorded using EPU version 3.10 and Tomography version 5.21, available from Thermo Fisher. Biolayer interferometry data were recorded using Octet BLI Discovery software version 12.2 (Sartorius). NAMD versions 3.0.1 and 3.1 alpha were used for MD simulations, as described. |
|---|---|
| Data analysis | Data analysis and validation was conducted using commercially or publicly available software (all references are given in the manuscript). MSe data were analyzed using Protein Lynx Global Server software (Waters). HDX data were analyzed using DynamX software version 3.0 (Waters) Cryo-EM image processing and 3D volume reconstruction: Relion version 4 and cryoSPARC version 4.6.2. Motion correction was done using Relion-4.0, MotionCor2 version 1.4.0, or WarpTools version 2.0.0 as described. CTF estimation was done using Gctf-v1.18 or WarpTools version 2.0.0 as described. Particles were picked using crYOLO version 1.9.6, Gautomatch version 0.56, and Topaz version 0.2.5a, as described. YOLOv11 (Ultralytics) was used to identify cores in micrographs; Roboflow web application (https://roboflow.com/) was used to build training dataset for YOLOv11. Cryo-ET data processing: WarpTools version 2.0.0 and IMOD version 5.1.1. UCSF Chimera version 1.17.3 was used for rigid body docking of atomistic models into cryo-EM maps and to create images. Namdinator web application (https://namdinator.au.dk/) was used for preliminary refinement of docked structural models. ChimeraX version 1.7 with AritiaX plugin was used to visualize tomograms. PyMOL version 2.4.1 was used to create illustrations with structural models. Tomogram denoising was done using DeepDeWedge software. Particle picking was done using Cryolo version 1.9.9, Gautomatch version 0.56, Topaz 0.2.5a, or Pytom version 0.11.1. Cryo-EM map post-processing: CryoSPARC version 4.6.2 and EMReady version 2. |

Cryo-EM structure real-space refinement: Coot version 0.9.8 and Phenix version 1.21.2.
Final model validation: MolProbity version 4.5 and Phenix version 1.21.2.
MALLS data were analyzed using ASTRA version 7.3.2 software (Wyatt Technology).
MODELLER version 10.6 was used to build missing residues to prepare models for molecular dynamics.
Visual Molecular Dynamics (VMD) version 1.9.4a57, compiled to work with python 3.9.
Scripts used for analysis of the molecular dynamics simulation trajectories can be found at https://doi.org/10.5281/zenodo.15866690.
ImageJ version 1.54g was used for semi-quantitative analysis of CA and IN signals on Western blots.
Virology data statistics was done with GraphPad Prism version 10.6.0
RNA structures were predicted using RNAfold web server (http://rna.tbi.univie.ac.at/cgi-bin/RNAWebSuite/RNAfold.cgi).
Protease cleavage sites were predicted using PROSPERousPlus (http://prosperousplus.unimelb-biotools.cloud.edu.au/) and HIVcleave (http://www.csbio. sjtu.edu.cn/bioinf/HIV/) web servers.

For manuscripts utilizing custom algorithms or software that are central to the research but not yet described in published literature, software must be made available to editors and reviewers. We strongly encourage code deposition in a community repository (e.g. GitHub). See the Nature Portfolio guidelines for submitting code & software for further information.

## Data

Policy information about availability of data

All manuscripts must include a data availability statement. This statement should provide the following information, where applicable:
- Accession codes, unique identifiers, or web links for publicly available datasets
- A description of any restrictions on data availability
- For clinical datasets or third party data, please ensure that the statement adheres to our policy

The cryo-EM and cryo-ET reconstructions were deposited with the EMDB under accession codes EMD-54067, EMD-54068, EMD-54070, EMD-54071, EMD-55409 and refined coordinates with the PDB under accession codes 9RMU and 9RMX (Extended Data Table 1). Raw tilt series data are available in EMPIAR under accession code EMPIAR-13078. HDX-MS data are available via ProteomeXchange with identifier PXD070910. All input parameters, structures and configuration parameters necessary for performing the molecular dynamics simulations presented in this work are freely available in Zenodo (https://doi.org/10.5281/zenodo.15866690). Structures used as starting models or for comparative analyses in this work are freely available from the PDB: 1K6Y, 2B4J, 5TC2, 6ES8, 6SKK, 7ASH, 7Z1Z, 8A1P, 9C9M.

## Research involving human participants, their data, or biological material

Policy information about studies with human participants or human data. See also policy information about sex, gender (identity/presentation), and sexual orientation and race, ethnicity and racism.

| | |
|---|---|
| Reporting on sex and gender | N/A |
| Reporting on race, ethnicity, or other socially relevant groupings | N/A |
| Population characteristics | N/A |
| Recruitment | N/A |
| Ethics oversight | N/A |

Note that full information on the approval of the study protocol must also be provided in the manuscript.

# Field-specific reporting

Please select the one below that is the best fit for your research. If you are not sure, read the appropriate sections before making your selection.

☒ Life sciences    ☐ Behavioural & social sciences    ☐ Ecological, evolutionary & environmental sciences

For a reference copy of the document with all sections, see nature.com/documents/nr-reporting-summary-flat.pdf

# Life sciences study design

All studies must disclose on these points even when the disclosure is negative.

| | |
|---|---|
| Sample size | Viruses were derived from transfected HEK293T cells in vitro. Final sample sizes were defined by the number of mutants under study. Sample sizes for individual experiments were in large part determined empirically. For example, to determine virus morphology by TEM, individual experiments contained 6 samples to match the capacity of the ultracentrifuge rotor.<br>For the molecular dynamics simulations, n=4 independent 1 microsecond trajectories were utilized for the IN-CA contact analysis. Using n>3 simulation replicas and microsecond length are standard for all-atom molecular dynamics studies. |
| Data exclusions | During cryo-EM image analysis, particle images belonging to 2D or 3D classes representing noise or lacking identifiable features were excluded, as explained in the Methods section. |

No data were excluded from virology experiments or from MD simulations.

| Replication | Fig. 4d: The experiments were repeated twice with similar results, aggregated results are shown in the figure, and results of separate trials are given in Source Data.<br>Extended Data Fig. 2a: Increased solubility of SIVtal IN relative to HIV-1 IN was consistently observed in multiple (n>3) independent repeats. Concordantly, unlike HIV-1 IN, SIVtal IN can be purified in the absence of CHAPS (see Methods).<br>Extended Data Fig 3a: Assembly of SIVtal IN complexes with TAR and GA18 RNA complexes were repeated 3 times (on different days), with filaments observed each time. Similarly, SIVtal IN in the absence of RNA was imaged 3 independent times, and no filaments were observed by cryo-EM.<br>Extended Data Fig. 4c: Purification of the HIV-1 cores in sucrose gradients is a routine procedure in our laboratories. Comparative immunoblotting of cores and without Vpr-NeonGreen-IN was done twice with similar results. Importantly, the fusion construct does not significantly affect HIV-1 infectivity and moreover rescues infectivity of HIV-1 IN mutants (ED Fig. 4b).<br>Cryo-EM/ET datasets for image processing and structural characterization were acquired from the best sample (one vitrified grid) identified by screening on lower power microscopes. Cryo-EM and cryo-ET reconstructions were refined using pairs of independent half-sets.<br>Final virus release, infection, and reverse transcription datasets were based on minimally three independent experiments, with each experiment containing technical duplicate samples. All attempts of experimental repeats were successful.<br>Some immunoblotting samples were analyzed two independent times, and both repeats were successful. Such samples were omitted from statistical evaluations.<br>MD simulations were collected from four independent replicas. |
|---|---|
| Randomization | Individual experiments were defined by the number of physically acceptable experimental samples. As discussed above, this sample size could be affected by the number of centrifuge buckets. Samples were semi-randomized based on the fact that the identity of precise samples was not always the same between experiments. Different random seeds were utilized to initialize each independent MD simulation replica.<br><br>Randomization was employed during cryo-EM 3D refinement in Relion and CryoSPARC, which use independent random subsets of particle images to avoid over-refinement. |
| Blinding | Due to practical constraints, investigators were not blinded to sample identity and outcome assessment. |

# Reporting for specific materials, systems and methods

We require information from authors about some types of materials, experimental systems and methods used in many studies. Here, indicate whether each material, system or method listed is relevant to your study. If you are not sure if a list item applies to your research, read the appropriate section before selecting a response.

## Materials & experimental systems

| n/a | Involved in the study |
|---|---|
| ☐ | ☒ Antibodies |
| ☐ | ☒ Eukaryotic cell lines |
| ☒ | ☐ Palaeontology and archaeology |
| ☒ | ☐ Animals and other organisms |
| ☒ | ☐ Clinical data |
| ☒ | ☐ Dual use research of concern |
| ☒ | ☐ Plants |

## Methods

| n/a | Involved in the study |
|---|---|
| ☒ | ☐ ChIP-seq |
| ☒ | ☐ Flow cytometry |
| ☒ | ☐ MRI-based neuroimaging |

## Antibodies

| Antibodies used | HIV-1 capsid, monoclonal AG3.0, NIH HIV Reagent Program, Division of AIDS, NIAID, cat. no. ARP-4121, lot. no. 70052504<br>HIV-1 capsid, monoclonal ARP-6458, NIH HIV Reagent Program, Division of AIDS, NIAID, cat. no. #24-3, lot. no. 130055<br>HIV-1 integrase, polyclonal antibodies generated in-house<br>Polyclonal Goat Anti-Rabbit Immunoglobulins/HRP, Agilent, cat. no. P0448, lot. no. 41723082<br>Polyclonal Swine Anti-Rabbit Immunoglobilins/HRP, Dako, cat. no. P0399, lot no. 20028547<br>Polyclonal Rabbit Anti-Mouse Immunoglobulins/HRP, Dako, cat. no. P0161, lot. no. 20044780<br>Polyclonal Goat Anti-Mouse Immunoglobulins/HRP, Dako, cat. no. P0447, lot. no. 20058518 |
|---|---|
| Validation | AG3.0 and ARP-6458 are widely used in the field (e.g., PMID: 22153299; 26586435) and revealed the appropriately-sized band via immunoblotting.<br>The HIV-1 integrase antibodies were validated at the time of production via the ability to detect purified recombinant integrase as a single band via immunoblotting and has since been used by others in their research (PMIDs: 33351861 and 10877832).<br><br>Species reactivity and suitability for Western blotting of conjugated secondary antibodies was validated by the commercial suppliers (Agilent/Dako) and by many publications, which can be found on CiteAb resource.<br>P0448: https://www.citeab.com/antibodies/3288347-p0448-goat-anti-rabbit-immunoglobulins-hrp-affinity?des=e8124065ecc857ce<br>P0399: https://www.citeab.com/antibodies/3288354-p0399-swine-anti-rabbit-immunoglobulins-hrp-affinit?des=5d02b793f06e63a5<br>P0161: https://www.citeab.com/antibodies/3288338-p0161-rabbit-anti-mouse-immunoglobulins-hrp-ig-frac?des=d4cc0efd2934bd69<br>P0447: https://www.citeab.com/antibodies/3288336-p0447-goat-anti-mouse-immunoglobulins-hrp-affinity?des=b716b61c45a8fd36 |

# Eukaryotic cell lines

Policy information about cell lines and Sex and Gender in Research

| | |
|---|---|
| Cell line source(s) | HEK293T cells were obtained from ATCC (cat. no. CRL-3216).<br>Lenti-X HEK293T cells were obtained from Takara Bio (cat. no. 632180). |
| Authentication | Cells were not independently authenticated. |
| Mycoplasma contamination | HEK293T cells were negative for mycoplasma as evidenced via regular monthly testing with MycoAlert mycoplasma detection kit (Lonza, cat. no. LT07-218).<br><br>Lenti-X HEK293T cells were negatuve by PCR using PHOENIXDX® MYCOPLASMA MIX (Procomcure Biotech, cat. no. PCCSKU15209). |
| Commonly misidentified lines<br>(See ICLAC register) | No commonly misidentified cell lines were used in this study. |

# Plants

| | |
|---|---|
| Seed stocks | N/A |
| Novel plant genotypes | N/A |
| Authentication | N/A |

