## [Peer Review file · Nature]

Integrase anchors viral RNA to the HIV-1 capsid interior

Corresponding Author: Dr Peter Cherepanov

Version 0:

Reviewer comments:

Referee #1

(Remarks to the Author)

This manuscript describes the observation that the HIV-1 integrase forms octamers that interact with the inner surface of the mature HIV-1 capsid. This observation is complemented with a structure of an SIV integrase complexed with RNA that adopts an oligomeric structure matching that observed for the “in situ” HIV-1 integrase, allowing description of IN-RNA interactions. A series of mutagenesis and molecular dynamics experiments further characterise the interactions between integrase and capsid.

In my opinion, the core observation of this manuscript – that integrase octamers directly interact with the inner surface of the HIV-1 capsid – has profound implications for understanding HIV-1. It is relevant to how we think about the function of integrase inhibitors, how we think about the mechanism for genome encapsidation in the mature capsid, how we think about formation of the intasome. The experiments are in general well performed, and the core observation is well supported. I am strongly in favour of publication of a revised manuscript.

Suggestions for revision:

1. There are clear differences between the HIV and SIV RNA protection patterns in Supplementary figure 4. There are also clear differences between the positions of the RNA densities (however badly resolved), between the in vitro SIV and the in situ HIV structures. The authors should comment on this – is it possible that the RNA binding differs in some way?

2a. The structure does not provide sufficient resolution to interpret side chain positions in protein-protein interactions directly so a set of MD experiments are used to describe these. I am not sure what to conclude from the three paragraphs on MD which talk about amino-acid interactions. Do the authors believe that the MD allows them to reliably describe the amino acid interactions mediating assembly and only awaits a high-resolution structure to be confirmed (in which case is this what is shown in 3c?) Or do the authors believe that the MD is informative to provide a best-guess structure that they then use to design mutagenesis experiments? In short, to what extent should the reader consider this description of amino acid interactions as reliable? Depending on how they answer this question they could substantially shorten this description.

2b. Figure 3c shows sidechains involved in the interaction, although the corresponding text states that side chain positions cannot be refined. Please be clear in the legend whether these are MD sidechain positions, or add a clear caveat.

2c. How does 3c relate to Fig. S16?

2d. The authors could be clearer in conclusion of the manuscript that understanding the amino acid interactions that mediate IN-CA binding will require further work and probably higher resolution structures.

3. It is hard for the reader to work through the text and figures describing mutagenesis data while relating them to the observed structure. The full set of assays (particle production, infectivity, RT, IN incorporation, morphology), unsurprisingly reveal that many of the mutations have effects at other (or multiple) stages of the lifecycle. (For a number of these mutations these phenotypes are already described). The only mutants which can be interpreted as likely providing a read out of CA-IN interactions are those where IN incorporation is normal but where the morphology assay shows increased eccentric particles. As far as I can tell that is limited to E69A/K for IN and K158E/A/T8I for CA, possibly also R167A. The authors could consider providing another more direct measurement by measuring the IN:CA ratio in purified capsid core preparations from mutant viruses. My suggestion is to move most of the results and figures for the mutagenesis to the supplement and to

provide a short text and figure that states why interpretation of most mutations in the context of CA-IN interactions is not possible, and then concentrates on this small subset of mutations where a relevant phenotype can be interpreted.

4. Can the authors be sure that the reduced infectivity and eccentric core of particles containing WT LEDGF IBD is not just a result of the observed reduction in IN incorporation?

5. The couple of sentences in the paper describing the MHR were unhelpful and I suggest to cut them and simply talk about the region around Lys158. Otherwise the text triggers the idea that the MHR is conserved as an IN binding site which is not the case and not borne out by the subsequent sections.

6. The 120° angle observed in the in vitro system is in my understanding irrelevant to whether it matches the CA lattice or not. Any angle (eg 100 or 140 degrees) could match to the lattice because the matching angle depends on what residues on the tetramer form the interactions with CA?

7. How are statistics calculated for the n=2 experiments?

8. Why plot infection and RT normalised to 10²? Would it not be more intuitive to normalise to 1 if using a log scale? Perhaps add a dotted line for the normalised value.

Referee #2

(Remarks to the Author)

In addition to catalyzing integration of viral DNA into host DNA, HIV integrase (IN) plays an important role in viral morphogenesis. Its function in this stage of HIV replication is not well understood, although it has been shown to interact with viral RNA in the virion. The structural basis of this interaction has remained unknown. Here, the authors have determined cryo-EM structures that establish the structural basis for the HIV-1 IN-RNA interaction and reveal that integrase forms an RNA-binding module on the luminal side of the mature CA lattice. Furthermore, amino acid substitutions of residues involved in integrase-capsid contacts resulted in eccentric virions with RNA nucleoids located outside of the cores, supporting the biological importance of these interactions. This is important novel work that will be of interest to the general readers of Nature, as well as those in the virology field. However, there are a few questions the authors need to address before publication:

1. The observed IN-RNA co-polymers are composed of two parallel linear filaments consisting of IN octamer repeat units. However, the IN-RNA atomic model derived from the cryo-EM map presents a significant clash issue between RNAs in adjacent repeat units of the two filaments. The TAR RNA (57 nucleotides) used for IN-RNA complex assembly includes 23 paired and 10 unpaired nucleotides. Yet, the atomic model only shows 16 base pairs of RNA, while the remainder is not visible. Moreover, the modeled 16 bp RNA clashes with the opposing IN-RNA complex that forms the linear filament. This raises the question of how the full-length TAR RNA fits into the refined map.

2. In the in vitro IN-RNA complex, the two RNA duplexes exit the IN octamer in parallel but appear nearly perpendicular to the axis of the IN filament. This geometry seems incompatible with how the viral RNA genome binds in native viral core particles (Supplementary Figure 13) ?

3. In the data processing workflow for viral cores, over 10 million particles were initially picked, yet only ~50,000 particles were used for the final reconstruction of the IN-CA complex. This suggests that the IN filaments formed in purified native cores are highly heterogeneous. Could it be that the well-ordered IN octamer observed in the core represents just one of several possible modes of IN oligomerization?

4. The ratio of CA to IN is approximately 2:1, which is unusually low compared to wild-type viral cores. Does this imply that IN overexpression is necessary for the formation of the well-ordered IN tetramer-CA hexamer structures observed in the viral core?

Minor comments and typos:

5. Page 15, lines 350-351. "HIV-1 and SIV_{tal} INs were concentrated using a 10-kDa cutoff Vivaspin device (Generon) to 8-10 and 1 mg/mL, respectively ...". More soluble SIV IN was concentrated to 1 mg/mL. Is this a typo as HIV IN is less soluble?

6. Page 20, line 490. "HIV-1 IN domains (from PDB IDs 1K6Y 14, XXX, YYY, and ZZZ -Matthew) in the cryo-EM map using UCSF Chimera". The PDB IDs of XXX, YYY, and ZZZ need to be specified.

7. Page 21, lines 500-502. "The data were recorded on a 200-kV Talos Arctica microscope using a Falcon 3 direct electron detector operated in linear mode. In total, 3,220 micrograph movies were recorded on a 200-kV Glacios microscope ...". Were the data recorded on a Talos Arctica or a Glacios microscope? It needs clarification.

8. Page 21, lines 504- 505. "... and a defocus range of -1.5 to -3.5 Å." The unit of defocus should be μm rather than Å.

9. Page 24, line 589. "non-uniform refinement of these particles with without and with C2 symmetry imposed ...". The word

“with” needs to be removed.

10. Page 24, line 595. “and 3,689,44 Gautomatch-picked particles ...”. The number “3,689,44” needs to be corrected (missing digit?).

11. Page 26, line 627. “... one more round of translational symmetry expansion”. To maintain consistency in the style of method, the number of particles after expansion should be stated.

12. Page 41, line 1094. “Biolayer interferometry 1094 (BLI) sensorgrams of SIVtal (left) and HIV-1 (right) IN binding to biotinylated RNA oligonucleotides...”. It should be HIV-1 (left) and SIVtal (right) .

13. Supplementary Figure 1, the lower panel/table. The length (nt) of TAR should be 57 rather than 56.

14. Page 41, line 1105. “The panels show SIVtal (left) and HIV-1 (right) IN...”. It should be “The panels show HIV-1 SIVtal (left) and SIVtal (right) IN”.

Referee #3

(Remarks to the Author)

I co-reviewed this manuscript with one of the reviewers who provided the listed reports.

Referee #4

(Remarks to the Author)

Summary and evaluation:

The authors defined how HIV Integrase (IN) interacts with RNA, first by using biochemical approaches to screen different RNA constructs and lentiviral IN constructs. These studies identified the HIV-1 TAR RNA stem-loop and SIVtal IN as attractive targets for structural studies. A cryoEM reconstruction (overall resolution of 3.7 Å resolution) of SIVtal IN-TAR RNA revealed that the complex formed regular filaments comprising two identical parallel linear filaments joined through RNA bridges. The basic repeating unit of each filament is an IN octamer formed by two identical homotetramers that shared pairs of RNA chains. Significantly, the authors noticed that the pairwise IN tetramer spacing and orientation matched those of CA hexamers in the viral capsid lattice. To test whether this matching interaction is responsible for packaging IN and RNA within the maturing capsid, they performed cryoEM imaging, sorting, and focused refinements, which convincingly identified the same linear polymer of IN octamers within the interiors of purified, IN-enriched HIV capsids. Each IN tetramer makes close contacts with exposed residues of the CA major homology region (MHR) element, which has long been known to play essential roles in HIV-1 assembly and viral maturation. Impressively thorough structural and mutational analyses demonstrated the biological relevance of the different structural observations, and revealed the functional significance of interfaces between IN dimers, tetramers, RNA and the capsid lattice.

Evaluation:

This beautiful study reveals how IN binds RNA and helps drive viral RNA packaging into the viral core particle, thereby nicely expanding on the more traditional role of IN in catalyzing integration of the viral DNA integration into host chromosomes. Individual components of the study are generally very well performed and reliable, and they collectively represent a major advance that will be of general interest to structural biologists, HIV biologists, and virologists. As described below, the study could be elevated even further with a clearer explanation of the relationship between the double-stranded IN octamer-RNA filaments formed in vitro and the single-stranded IN octamer-capsid lattice formed within purified core particles, and by a few additional clarifications and analyses.

Specific comments:

1) Supplemental Figs. 1 and 2. Can the authors analyze the BLI data to extract kinetic rate constants and KD values (or explain clearly why they can't do that – it should at least be possible to report IN concentrations required for half maximal binding to the different RNAs).

2) To the authors' credit they have very nicely explained the functions of the conserved, exposed residues in the Major Homology Region (MHR). This has been a long-standing question that hasn't ever been adequately resolved (until now) and I suggest that the authors point this out explicitly in the Discussion. The MHR likely does multiple things, especially given its immature assembly phenotypes, but binding IN filaments in the maturing capsid is clearly one of them, and it's an important one. The authors clearly understand this point – they rigorously tested all of the contacts – but I suggest saying it explicitly in the Discussion.

3) Similarly, this study presents the first structure of a subcomplex within the HIV-1 capsid and I suggest noting that explicitly in the Discussion.

4) The purified cores appear to have a lot of p55Gag in them (Supplemental Fig. 9). I think this merits comment. It would also be useful to see negative stained EM images of the core preparations because it's hard to get a sense of core homogeneity from the cryoEM images (although the quality is clearly sufficient to generate high quality reconstructions).

5) The authors could do a better job of explaining the relationship between the double-stranded SIVtal IN protein/HIV TAR RNA complex and the single-stranded HIV-1 IN-capsid lattice complex. Specifically, it would be good to be more explicit about: A) The relationship between the two strands of IN octamers in the reconstructed double filaments from the SIVtal IN protein/HIV TAR RNA complex and the single strand of IN octamers within the HIV cores. Which surface from the contributing strands in the SIVtal IN structure is used to contact the HIV capsid hexamer lattice? The CA-IN contacts are nicely explained, but the relationship between the double stranded SIVtal IN/RNA filament and the single stranded HIV-1 IN filament within the capsid is hard to visualize - perhaps just one additional supplemental figure explaining this could be added. B) The relationship between the RNA binding sites in the double-stranded SIVtal IN/RNA filament and the extra "RNA" density in the single stranded HIV-1 IN filament. Perhaps an additional figure that overlaid the RNA positions in one of the two strands from the SIVtal IN/RNA complex (e.g., Fig. 1b) onto the extra "RNA" density shown in Supplemental Fig. 13a. I understand that there is more to learn about how the viral RNA binds the IN filament within the capsid, but I found it hard to even relate the RNA density in the two contexts. It looked to me as though they might not overlap well, but in any event this should be explained better.

6) The authors could also do a better job of testing and explaining whether the IN filament binds to preferred site(s) on the conical capsid. It should be possible to map the individual filaments that contribute to the reconstruction back onto a model capsid and thereby reveal whether the filaments bind preferentially to the body of the cone, whether there is a preferred set of positions, etc. It might also be useful to discuss a bit more whether the authors think that the IN filaments promote the shape or assembly of the capsid.

Minor points:

Main Text:

- 1) Line 66. "Supplementary Fig. 3b" should be changed to "Supplementary Fig. 3c" to match the figure legend and updated figure (see below).
- 2) Line 69. Similarly, "Supplementary Fig 3a" should be changed to "Supplementary Fig 3b" to match the figure legend.
- 3) Line 72. The word "of" is missing ("in the absence or presence OF RNATar")
- 4) Line 207. F139D doesn't seem to exhibit the >100 fold infection defect that is described (although the other mutants E69K and E69A do). F139D looks more like 5-fold.

Figures and Captions:

- 1) Figure 2 caption: Should mention that this structure is SIVtal IN
- 2) Figure 2c: Residue 2 is labeled "L2", but line 105 refers to Val2. The alignment in Supplementary Figure 3 shows HIV IN with Leu2 and SIV IN with Val2. Please correct the figure to "V2".
- 3) Labels for SIVtal and HIV-1 are reversed in the Supplementary Figure 1 caption, line 1094 (vs. the figure itself).
- 4) Supplementary Figure 1: use the same bright green color for 125 nM IN in the figure caption as is used in the sensorgrams.
- 5) Supplementary Figure 2 caption, Line 1105. Same reversal Supplementary Figure 1 caption; match the order of HIV IN and SIV IN in the figure and the figure caption (left vs right).
- 6) Supplementary Figure 3 caption: line 1113 legend uses terms "total" and "soluble" whereas the figure itself uses "tot" and "sup". Using "sol" in place of "sup" would make more sense.
- 7) Supplementary Figure 3: the Size-exclusion chromatograph has no label but is referred to as "b" in the legend. Then, the IN alignment panel is labeled "b" but is referred to as "c" in the figure caption.
- 8) Supplementary Figure 4 and caption: Please define what the shaded area between the red and blue dotted lines represents (perhaps the red/blue average?).
- 9) Supplementary Figure 6 caption, line 1147: "optimized" is used in the caption, but "Corrected" is used in figure. Should they match?
- 10) Supplementary Figure 6 caption: line 1147 refers to a "horizontal blue line" that appears black in the figure.
- 11) Supplementary Figure 6 caption: lines 1153 and 1154 have the same issues as line 1147.
- 12) Supplementary Figure 9 legend: line 1189 refers to "underlined residues marking the native N-terminus" but underlined residues aren't evident in the figure.
- 13) Supplementary Figure 10c: it would be helpful to add the numbers of particles (or percentages) that went into each 3D class (in addition to the capsid shell class).

Referee #5

(Remarks to the Author)

I co-reviewed this manuscript with one of the reviewers who provided the listed reports.

Version 1:

Reviewer comments:

Referee #1

(Remarks to the Author)

The authors have done a very good job addressing the reviewer comments. The additional data on viruses without overexpressed IN is valuable. I congratulate the authors on this impressive, landmark study.

Very minor comments:

1. Lines 187-189 comment that overlapping density weaker in single particle reconstruction: Given the excess of RNA over IN, is the oversupply of IN a reasonable explanation for this difference?
2. The images in extended data 9a appear shifted relative to one another.
3. Lines 232-233 - K170A infectivity and morphology were described by von Schwedler (ref 29), which could be recited here.
4. If the application of single-particle image processing to the capsid core was inspired by Highland et al. 2023, and/or Stacey et al., 2023, a citation could be added.

Referee #4

(Remarks to the Author)

The authors have addressed all of my concerns and in my view this manuscript is ready for publication.

Referee #5

(Remarks to the Author)

I co-reviewed this manuscript with one of the reviewers who provided the listed reports.

We thank all Reviewers for their positive assessment of our work and constructive criticisms, which helped us to improve the manuscript. Follows is our point-by-point responses to their comments. Please note that to comply with journal standards, the original 15 Supplementary Figures were compressed into 10 Extended Data items; Extended Data Figure 9 is moreover new cryo-ET data with native HIV-1 cores.

Referee #1

This manuscript describes the observation that the HIV-1 integrase forms octamers that interact with the inner surface of the mature HIV-1 capsid. This observation is complemented with a structure of an SIV integrase complexed with RNA that adopts an oligomeric structure matching that observed for the “in situ” HIV-1 integrase, allowing description of IN-RNA interactions. A series of mutagenesis and molecular dynamics experiments further characterise the interactions between integrase and capsid.

In my opinion, the core observation of this manuscript – that integrase octamers directly interact with the inner surface of the HIV-1 capsid – has profound implications for understanding HIV-1. It is relevant to how we think about the function of integrase inhibitors, how we think about the mechanism for genome encapsidation in the mature capsid, how we think about formation of the intasome. The experiments are in general well performed, and the core observation is well supported. I am strongly in favour of publication of a revised manuscript.

Suggestions for revision:

1. There are clear differences between the HIV and SIV RNA protection patterns in Supplementary figure 4. There are also clear differences between the positions of the RNA densities (however badly resolved), between the in vitro SIV and the in situ HIV structures. The authors should comment on this – is it possible that the RNA binding differs in some way?

Response:

As the HIV-1 and SIVtal integrases share ~50% sequence identity, they yield different peptide fragments upon pepsin digestion. This makes direct, residue-by-residue comparison between their HDX-MS profiles inherently difficult. Furthermore, hydrogen/deuterium exchange is influenced by both the local amino acid sequence and local hydrogen bonding, which further complicates direct comparisons. Nonetheless, grossly, the protection seen due to interactions with RNA seem to be similar in the sense that they are spread throughout all three IN domains.

Most importantly, HDX indicated specific and wide-spread structural changes that were unlikely to arise from non-specific electrostatic interactions alone, which motivated our initial efforts to study the IN-RNA complexes by cryo-EM.

In terms of the RNA densities between the SIVtal and in situ HIV-1 structures, we have performed new cryo-ET experiments with native cores. Although somewhat limited by overall resolution (12.6 Å), the results nevertheless importantly confirmed the presence of the IN filament structure at the luminal capsid surface (at the native integrase level), as well as additional density overlaying the IN filament, which we tentatively interpret as the viral RNP. Pleasingly, aspects of this density insert through the IN filament at places that are nearly congruent to the dsRNA region of RNA^{TAR} in the SIV IN structure (page 8): “Remarkably, this density penetrated the IN filament at sites that correspond closely to the RNA duplex-binding positions observed in our SIVtal IN-RNA^{TAR} structure (Extended Data Fig. 9f)”.

2a. The structure does not provide sufficient resolution to interpret side chain positions in protein-protein interactions directly, so a set of MD experiments are used to describe these. I am not sure what to conclude from the three paragraphs on MD which talk about amino-acid interactions. Do the authors believe that the MD allows them to reliably describe the amino acid interactions mediating assembly and only awaits a high-resolution structure to be confirmed (in which case is this what is shown in 3c?) Or do the authors believe that the MD is informative to provide a best-guess structure that they then use to design mutagenesis experiments? In short, to what extent should the reader consider this description of amino acid interactions as reliable? Depending on how they answer this question they could substantially shorten this description.

Response:

MD was used to confirm that amino acid compositions and geometry of the contract regions are conducive to formation of IN-CA interactions. We have significantly shortened the discussion of the MD results in the main text (from 3 paragraphs to a single sentence at the bottom of page 7) and moved the original discussion to Extended Data Fig. 8 legend.

2b. Figure 3c shows sidechains involved in the interaction, although the corresponding text states that side chain positions cannot be refined. Please be clear in the legend whether these are MD sidechain positions, or add a clear caveat.

Response:

The figures are consistent with the refined models. We explained this caveat in the legend to Figure 3c: “Although conformations of amino acid side chains are poorly defined due to modest resolution of the cryo-EM map, they are shown, as sticks, to indicate their approximate locations as they appear in the refined model.”

2c. How does 3c relate to Fig. S16?

Response:

The original figure was showing energy minimized model. We decided to revise the figure to show snapshots of interactions within CR1-4 observed during MD simulations (Extended Data Fig. 8 in the revised manuscript).

2d. The authors could be clearer in conclusion of the manuscript that understanding the amino acid interactions that mediate IN-CA binding will require further work and probably higher resolution structures.

Response:

OK, we modified our conclusion statement as suggested by the reviewer: “In conclusion, our results provide the structural bases for IN-RNA interactions and the formation of the proto-intasomal IN-RNA filament on the luminal side of the capsid, explaining the role of IN in RNP complex incorporation into morphologically functional HIV-1 cores and warranting further work to study retroviral CA-IN interactions at atomistic detail.”

3. It is hard for the reader to work through the text and figures describing mutagenesis data while relating them to the observed structure. The full set of assays (particle production, infectivity, RT, IN incorporation, morphology), unsurprisingly reveal that many of the mutations have effects at other (or multiple) stages of the lifecycle. (For a number of these mutations these phenotypes are already described). The only mutants which can be interpreted as likely providing a read out of CA-IN interactions are those where IN incorporation is normal but where the morphology assay shows increased eccentric particles. As far as I can tell that is limited to E69A/K for IN and K158E/A/T8I for CA, possibly also R167A. The authors could consider providing another more direct measurement by measuring the IN:CA ratio in purified capsid core preparations from mutant viruses. My suggestion is to move most of the results and figures for the mutagenesis to the supplement and to provide a short text and figure that states why interpretation of most mutations in the context of CA-IN interactions is not possible, and then concentrates on this small subset of mutations where a relevant phenotype can be interpreted.

Response:

We feel the reviewer is overly focused on one aspect of the mutagenesis work and, because of this, is restricting full appreciation of the approach and the results. First off, only 5 of the reported 20 IN/CA mutant viruses, CA K158E/D, IN H12N (historical control) and IN V32A/E, had statistically low IN-to-CA content. In terms of V32A/E, the cryo-EM structure of the SIVtal IN-RNA complex highlighted NTD-NTD contacts that mediated IN octamer-octamer interactions, the importance of which we wanted to test in the context of HIV-1 infection. Several of the participating residues, including F1, L2 and I5, were essentially unmutable, as they compose part of the RT/IN protease cleavage site. We accordingly targeted V32, which, to the best of our knowledge, had not previously been studied. The fact that both V32A and V32E were completely dead is accordingly significant. E.g., finding that these viruses infected cells at the WT level would downplay the significance of the noted structural biology result. Thus, the V32 mutagenesis results are noteworthy, independent of the nuanced finding that IN protein content was low for these viruses. For clarity, we slightly amended the wording of the short 2 sentence description of these findings and opt to retain this as part of page 9 main text: “V32A and V32E were highly defective viruses (>1,000-fold infection defects). In addition to harbouring significant >10-fold reverse transcription defects and increased eccentric and immature particle counts, these mutants were defective for virion IN protein content, similar to the class II IN mutant control virus H12N (Fig. 4 and Extended Data Fig. 7b).”

Although others had previously noted highly altered CA K158A/D virus morphologies (ref 29), the IN content of these viruses was unknown. Maintaining these data in main text importantly frames the impact of the ensuing compensatory T8I change on CA K158A/E biology (particle release, infectivity, reverse transcription, IN content and eccentric viral phenotypes).

Another important example is K170A, which altered a basic MHR residue distal from the purported IN interaction site. Though this virus was dead, it morphologically mirrored the WT (and retained normal IN-to-CA content ratio). In this way, K170A served an important specificity control for other targeted (proximal to IN interaction site) MHR basic residues (K158/T8I and R167) that yielded the eccentric mutant virus phenotype; page 10: “CA K170A was also a severely defective virus (Fig. 4b). These particles, however, were largely morphologically similar to WT (Fig. 4d), consistent with the distal positioning of Lys170 relative to IN in the structure.”

In deference to the request to downplay the significance of the MHR (upcoming comment 5), we remind the reviewer that R167 is one of the three invariant residues. Moreover, as referenced in Extended Data Fig. 8 legend, R167 in prior structures is usually seen to engage invariant residue E159; as predicted via MD simulations, both residues become available for IN interactions in CR2. These observations underscore our approach to alter all three invariant MHR residues (Q/E/R); as highlighted in text, because Q155 is comparatively distal from the IN interaction sites, the Q155 mutants served as invariant MHR residue controls (page 9) “As controls, we assessed Q155A/N changes of the remaining invariant MHR residue...”

Although we can appreciate the request to study the protein content of isolated HIV-1 cores, we feel that the content of pelleted viruses sufficiently controls for this aspect of the mutagenesis work and, accordingly, that work with isolated cores is not essential for the publication. The study of isolated cores is a good idea for follow-up work.

4. Can the authors be sure that the reduced infectivity and eccentric core of particles containing WT LEDGF IBD is not just a result of the observed reduction in IN incorporation?

Response:

Thank you for this suggestion. The modified text reads “Consistent with IN displacement from the CA lattice, the virus containing the WT IBD displayed 3 to 4-fold reductions in reverse transcription and infectivity, respectively, as well as an elevated fraction of eccentric particles, although we cannot rule out potential contributions from the ~45% reduction in virion IN protein (Fig. 4a-d and Extended Data Fig. 7b).”

5. The couple of sentences in the paper describing the MHR were unhelpful and I suggest to cut them and simply talk about the region around Lys158. Otherwise the text triggers the idea that the MHR is conserved as an IN binding site which is not the case and not borne out by the subsequent sections.

Response:

We would respectfully disagree. The “region around Lys158” is by definition the MHR (Wills JW, Craven RC, 1991 AIDS 5:639). It accordingly would be unscholastic to discuss Lys158 outside of this most conserved region of orthoretroviral capsid proteins. We do though agree that calling the MHR an IN binding site is unwarranted and is not supported by the data. As stated in response to comment 3 above, R167, one of the invariant residues, is clearly relevant (R167A virions were characterised as eccentric by Göttlinger and colleagues in 1994 J Virol!). Thus, this extends structurally and genetically beyond K158. Also as just mentioned, R167 oftentimes interacts intermolecularly with a second invariant residue, E159. Given these arguments, we feel that our curt description of the MHR and the targeting of all 3 invariant residues is warranted (see comment 3 discussion above). We would also direct the reviewer to Reviewer 3/4 comment 2, who very much appreciated our MHR descriptions.

6. The 120° angle observed in the in vitro system is in my understanding irrelevant to whether it matches the CA lattice or not. Any angle (eg 100 or 140 degrees) could match to the lattice because the matching angle depends on what residues on the tetramer form the interactions with CA?

Response:

The reviewer is right. Yet, it is clear that the IN octamer repeat units are inclined by ~120°. Noting this, along with the coincidental repeat lengths, was a memorable Aha moment, which we want to share. To avoid complex math in defining vectors, we opted to show the overlay of the CA lattice and IN filament in revised Extended Data Fig. 4a, which illustrates the most striking overlap.

7. How are statistics calculated for the n=2 experiments?

Response:

Thank you for this comment. In response, we have added a separate Statistics section at the end of the Methods section. For the virus infection data, we have moreover performed additional experimental repeats to ensure all Fig. 4a-c and Extended Data Fig. 4b data are the result of n=3 cumulative experiments. Please note that these amended results have not in any way affected our previous interpretations or conclusions, although some average values that are cited in text and p value ranges noted in legends have marginally changed. We also now graph these data with transparent bars to readily reveal the independent experimental datapoints. While some of the immunoblot quantifications were thrice repeated, other samples were analysed only twice. We have adopted transparent bars such that readers can tell this difference and have removed the Extended Data Fig. 7b statistical analysis to account for the fact that not all samples were n=3.

Preparation of electron micrographs for virion morphology determinations was independently done twice. Due to the large number of particles counted across experiments, which ranged from n=141 to n=1,105 across virus samples, we argue that these data nevertheless warrant statistical analyses.

8. Why plot infection and RT normalised to 10²? Would it not be more intuitive to normalise to 1 if using a log scale? Perhaps add a dotted line for the normalised value.

Response:

Thanks for bringing this to our attention and we apologise for the confusion. In all cases, mutant virus responses were percent-normalized to the WT (hence the 10² or 100% value). To fix, we have clearly labelled all y-axes and amended legend texts.

Referee #2:

In addition to catalyzing integration of viral DNA into host DNA, HIV integrase (IN) plays an important role in viral morphogenesis. Its function in this stage of HIV replication is not well understood, although it has been shown to interact with viral RNA in the virion. The structural basis of this interaction has remained unknown. Here, the authors have determined cryo-EM structures that establish the structural basis for the HIV-1 IN-RNA interaction and reveal that integrase forms an RNA-binding module on the luminal side of the mature CA lattice. Furthermore, amino acid substitutions of residues involved in integrase-capsid contacts resulted in eccentric virions with RNA nucleoids located outside of the cores, supporting the biological importance of these interactions. This is important novel work that will be of interest to the general readers of Nature, as well as those in the virology field. However, there are a few questions the authors need to address before publication:

1. The observed IN–RNA co-polymers are composed of two parallel linear filaments consisting of IN octamer repeat units. However, the IN–RNA atomic model derived from the cryo-EM map presents a significant clash issue between RNAs in adjacent repeat units of the two filaments. The TAR RNA (57 nucleotides) used for IN–RNA complex assembly includes 23 paired and 10 unpaired nucleotides. Yet, the atomic model only shows 16 base pairs of RNA, while the remainder is not visible. Moreover, the modeled 16 bp RNA clashes with the opposing IN–RNA complex that forms the linear filament. This raises the question of how the full-length TAR RNA fits into the refined map.

Response:

In the *in vitro* structure, each TAR RNA is shared between two structurally equivalent opposing runs of IN octamers. Using our locally refined cryo-EM map, we modelled more than half of the RNA ligand (as far as the map allowed). This is why when the model is fitted in both sides of the globally refined cryo-EM map, there is a 6-bp overlap at the outer tips of the RNA duplexes from opposing IN octamers. To generate a starting model for the double filament, one has to trim the outer tip of the RNA by 3 bp in the model.

2. In the *in vitro* IN–RNA complex, the two RNA duplexes exit the IN octamer in parallel but appear nearly perpendicular to the axis of the IN filament. This geometry seems incompatible with how the viral RNA genome binds in native viral core particles (Supplementary Figure 13)?

Response:

There was a disconnect between our *in vitro* and *in situ* structures primarily because the signal from the RNP was very weak and fragmented in the latter. Therefore, we are excited to include a cryo-ET structure of the filament in native HIV-1 cores produced without overexpression of IN. The subtomogram average reconstructed at 12.6 Å resolution revealed substantial features not explained by CA or IN, which likely correspond to the RNP. These include density penetrating the IN filament at positions close to those of the TAR RNA observed in our *in vitro* SIVtal IN filament (Extended Data Fig. 9f). With this new data, we can be more explicit in explaining how IN filament staples vRNA to core lumen (see page 11): “Remarkably, the cryo-ET reconstruction of the filament in native HIV-1 cores revealed additional density, not explained by CA or IN, that closely matches the position of the RNATAR duplexes in complex with SIVtal IN (Extended Data Fig. 9f), strongly indicating that these positions are occupied by RNA *in vivo*. Capture of variable stem loops in viral RNA by the IN filament would staple the entire RNP to the core lumen. However, more work is required to dissect IN interactions with the entire gamut of viral RNA structures, including single stranded RNA (Extended Data Figs 1 and 3a).”

3. In the data processing workflow for viral cores, over 10 million particles were initially picked, yet only ~50,000 particles were used for the final reconstruction of the IN–CA complex. This suggests that the IN filaments formed in purified native cores are highly heterogeneous. Could it be that the well-ordered IN octamer observed in the core represents just one of several possible modes of IN oligomerization?

Response:

Initially picked particles contain abundant junk, which does not lend itself to averaging; this is a common observation in single particle cryo-EM. And it cannot be otherwise in our case: we started by picking CA lattice flakes using a grid (essentially random picking within each core bounding box). But even template matching in 2D is prone to pick junk, especially with gautomatch (which is why we restricted gautomatch picks to the core bounding boxes). However, importantly, none of these techniques were biased to the presence of IN filament. After the particles were distilled through rounds of 2D and 3D classifications, we arrived at ~800k CA lattice flakes that collectively contributed to a reasonable 3D reconstruction of the CA lattice. From this set, we were able to isolate ~50k particles that contained the IN filament (class 4, Extended Data Fig. 5c). We could argue that the ~50k particles in class 7 also contained the IN filament, shifted by one CA hexamer row off from the CA flake center. Therefore, we have ~100k filaments in 800k CA flakes in a dataset picked using methods unbiased to the presence of the IN filament. Given ~10-fold excess of CA over IN in these virions, we would not expect every identified CA lattice flake to contain an IN filament.

The biology is likely more nuanced, as the IN filament may prefer (or induce) specific shape of the core wall. Follow up cryo-ET studies will address these questions.

4. The ratio of CA to IN is approximately 2:1, which is unusually low compared to wild-type viral cores. Does this imply that IN overexpression is necessary for the formation of the well-ordered IN tetramer–CA hexamer structures observed in the viral core?

Response:

We packed ~2x more IN inside cores as compared to what is usually present. Therefore, while the expected ratio of CA monomer to IN monomer is 20:1, we pushed this ratio to ~10:1 (1.7-2.3 fold more IN per mature CA, compared to normal cores; Extended Data Fig. 4c).

To ensure that co-expression of Vpr-mNeonGreen-IN did not significantly compromise the infectivity of the single-cycle HIV-Luc vector, we performed new trans-complementation experiments, testing either the WT or D64N/D116N mutant of Vpr-mNeonGreen-IN alongside WT, class I mutant D64N/D116N, or class II mutant V165A Luc viruses that were constructed using the same 3:1 viral:Vpr DNA mass ratio used for our structural work. This new data is shown as Extended Data Fig. 4b; associated page 6 text: "Wildtype (WT) and mutant version versions of Vpr-NG-IN significantly boosted the infectivity of class I and class II HIV-1 IN mutant viruses, as previously shown for other Vpr-IN chimeras (20,27), while Vpr-NG-IN marginally impaired WT HIV-1 infectivity (Extended Data Fig. 4b)."

Finally, our new cryo-ET structure refined from cores produced without Vpr-mNeonGreen-IN confirms that filament formation does not require oversupply of IN (Extended Data Fig. 9).

Minor comments and typos:

5. Page 15, lines 350-351. "HIV-1 and SIVtal INs were concentrated using a 10-kDa cutoff Vivaspin device (Generon) to 8-10 and 1 mg/mL, respectively ...". More soluble SIV IN was concentrated to 1 mg/mL. Is this a typo as HIV IN is less soluble?

Response:

HIV-1 was purified and concentrated in the presence of CHAPS. The detergent could be avoided altogether with SIVtal IN, although the protein did not concentrate well far beyond 1 mg/ml. SIVtal IN is more soluble at moderate salt concentrations than HIV-1 IN, as we show in Extended Data Fig. 2a, albeit the abysmal solubility of HIV-1 IN is easy to beat.

6. Page 20, line 490. "HIV-1 IN domains (from PDB IDs 1K6Y 14, XXX, YYY, and ZZZ –Matthew) in the cryo-EM map using UCSF Chimera". The PDB IDs of XXX, YYY, and ZZZ need to be specified.

Response:

Thank you! The relevant PDB IDs have been specified in the revised version: "The model of the SIVtal IN-RNA complex was initially assembled by rigid body docking of individual HIV-1 IN domains (from PDB IDs 1K6Y, 8A1P), and 5TC2) in the cryo-EM map using UCSF Chimera."

7. Page 21, lines 500-502. "The data were recorded on a 200-kV Talos Arctica microscope using a Falcon 3 direct electron detector operated in linear mode. In total, 3,220 micrograph movies were recorded on a 200-kV Glacios microscope ..." Were the data recorded on a Talos Actitica or a Glacios microscope? It needs clarification.

Response:

The dataset from which images are shown was acquired on Glacios. The duplicated sentence with Talos was removed in the revised version. Thank you for highlighting this ambiguity.

8. Page 21, lines 504- 505. "... and a defocus range of -1.5 to -3.5 Å." The unit of defocus should be µm rather than Å.

Response:

OK. Thank you!

9. Page 24, line 589. "non-uniform refinement of these particles with without and with C2 symmetry imposed ...". The word "with" needs to be removed.

Response:

OK. Thank you!

10. Page 24, line 595. "and 3,689,44 Gautomatch-picked particles ...". The number "3,689,44" needs to be corrected (missing digit?).

Response:

OK. Thank you! The correct number is 3,689,441.

11. Page 26, line 627. "... one more round of translational symmetry expansion". To maintain consistency in the style of method, the number of particles after expansion should be stated.

Response:

The dataset contained 138,637 particles after expansion. We added the requested number in the revised manuscript "The best-defined 3D class, comprising 43,320 particles, was subjected to non-uniform refinement and one more round of translational symmetry expansion resulting in 138,637 non-overlapping particles."

12. Page 41, line 1094. "Biolayer interferometry (BLI) sensorgrams of SIVtal (left) and HIV-1 (right) IN binding to biotinylated RNA oligonucleotides...". It should be HIV-1 (left) and SIVtal (right).

Response:

OK. Thank you.

13. Supplementary Figure 1, the lower panel/table. The length (nt) of TAR should be 57 rather than 56.

Response:

OK. Thank you.

14. Page 41, line 1105. "The panels show SIVtal (left) and HIV-1 (right) IN...". It should be "The panels show HIV-1 SIVtal (left) and SIVtal (right) IN".

Response:

OK. Thank you.

Referee #3,4:

Summary and evaluation:

The authors defined how HIV Integrase (IN) interacts with RNA, first by using biochemical approaches to screen different RNA constructs and lentiviral IN constructs. These studies identified the HIV-1 TAR RNA stem-loop and SIVtal IN as attractive targets for structural studies. A cryoEM reconstruction (overall resolution of 3.7 Å resolution) of SIVtal IN-TAR RNA revealed that the complex formed regular filaments comprising two identical parallel linear filaments joined through RNA bridges. The basic repeating unit of each filament is an IN octamer formed by two identical homotetramers that shared pairs of RNA chains. Significantly, the authors noticed that the pairwise IN tetramer spacing and orientation matched those of CA hexamers in the viral capsid lattice. To test whether this matching interaction is responsible for packaging IN and RNA within the maturing capsid, they performed cryoEM imaging, sorting, and focused refinements, which convincingly identified the same linear polymer of IN octamers within the interiors of purified, IN-enriched HIV capsids. Each IN tetramer makes close contacts with exposed residues of the CA major homology region (MHR) element, which has long been known to play essential roles in HIV-1 assembly and viral maturation. Impressively thorough structural and mutational analyses demonstrated the biological relevance of the different structural observations, and revealed the functional significance of interfaces between IN dimers, tetramers, RNA and the capsid lattice.

Evaluation:

This beautiful study reveals how IN binds RNA and helps drive viral RNA packaging into the viral core particle, thereby nicely expanding on the more traditional role of IN in catalyzing integration of the viral DNA integration into host chromosomes. Individual components of the study are generally very well performed and reliable, and they collectively represent a major advance that will be of general interest to structural biologists, HIV biologists, and virologists. As described below, the study could be elevated even further with a clearer explanation of the relationship between the double-stranded IN octamer-RNA filaments formed in vitro and the single-stranded IN octamer-capsid lattice formed within purified core particles, and by a few additional clarifications and analyses.

Specific comments:

1) Supplemental Figs. 1 and 2. Can the authors analyze the BLI data to extract kinetic rate constants and KD values (or explain clearly why they can't do that – it should at least be possible to report IN concentrations required for half maximal binding to the different RNAs).

Response:

Unfortunately, it was not possible to generate reliable fits of the sensorgrams to a standard model, likely due to formation of extended IN multimers when binding to RNA. In addition, both INs aggregate under isotonic conditions, making IN concentration undetermined. For example, the more robust SIVtal IN RNA binding is likely explained by its higher availability in solution. Still, the BLI experiments were useful to demonstrate RNA binding.

2) To the authors' credit they have very nicely explained the functions of the conserved, exposed residues in the

Major Homology Region (MHR). This has been a long-standing question that hasn't ever been adequately resolved (until now) and I suggest that the authors point this out explicitly in the Discussion. The MHR likely does multiple things, especially given its immature assembly phenotypes, but binding IN filaments in the maturing capsid is clearly one of them, and it's an important one. The authors clearly understand this point – they rigorously tested all of the contacts – but I suggest saying it explicitly in the Discussion.

Response:

Thank you for this suggestion; paragraph 2 Discussion was expanded: “Although the MHR has long been recognised for its role in orthoretroviral particle assembly and maturation (27,28), our data clarify that this region of CA also helps to nucleate ribonucleoproteins for IN-mediated incorporation of viral RNA into mature HIV-1 cores. Exciting future work will aim to assess the roles of other retroviral MHRs in IN-mediated encapsulation of their RNA genomes.”

3) Similarly, this study presents the first structure of a subcomplex within the HIV-1 capsid and I suggest noting that explicitly in the Discussion.

Response:

We agree, unfortunately Nature specifically disallows such claims/ terminology (“novel” or “the first”).

4) The purified cores appear to have a lot of p55Gag in them (Supplemental Fig. 9). I think this merits comment. It would also be useful to see negative stained EM images of the core preparations because it's hard to get a sense of core homogeneity from the cryoEM images (although the quality is clearly sufficient to generate high quality reconstructions).

Response:

We were deliberate to not over-purify the cores, subjecting them to minimal manipulation. Representative cryo-EM images in Extended Data Fig. 5a show examples of mature and immature particles, including examples of mature cores that were not boxed by Yolo. Legend to revised Extended Data Fig. 5 provides important details: “The right image in the top row includes immature particles (red arrowheads) as well as mature cores not detected by the initial screen (yellow arrowheads). The presence of immature cores is consistent with Western blot analysis of sucrose gradient fractions (Extended Data Fig. 4c). Initially, particles within each YOLOv11 box were picked using an evenly spaced grid; an example of 12-nm grid picking is shown in the bottom right panel (green circles). Note that YOLOv11 boxes were used both to generate grids for initial picking and to prune Gautomatch results; in contrast, full micrograph areas were picked using Topaz.” We will be happy to share raw cryo-EM images upon request.

5) The authors could do a better job of explaining the relationship between the double-stranded SIVtal IN protein/HIV TAR RNA complex and the single-stranded HIV-1 IN-capsid lattice complex. Specifically, it would be good to be more explicit about: A) The relationship between the two strands of IN octamers in the reconstructed double filaments from the SIVtal IN protein/HIV TAR RNA complex and the single strand of IN octamers within the HIV cores. Which surface from the contributing strands in the SIVtal IN structure is used to contact the HIV capsid hexamer lattice? The CA-IN contacts are nicely explained, but the relationship between the double stranded SIVtal IN/RNA filament and the single stranded HIV-1 IN filament within the capsid is hard to visualize - perhaps just one additional supplemental figure explaining this could be added. B) The relationship between the RNA binding sites in the double-stranded SIVtal IN/RNA filament and the extra “RNA” density in the single stranded HIV-1 IN filament. Perhaps an additional figure that overlaid the RNA positions in one of the two strands from the SIVtal IN/RNA complex (e.g., Fig. 1b) onto the extra “RNA” density shown in Supplemental Fig. 13a. I understand that there is more to learn about how the viral RNA binds the IN filament within the capsid, but I found it hard to even relate the RNA density in the two contexts. It looked to me as though they might not overlap well, but in any event this should be explained better.

Response:

These are excellent points.

A) In the revised manuscript we are explicit about why we considered the single filament to be more relevant, which was well before we observed it inside cores (page 4): “The averaged structure is comprised of two identical linear filaments running parallel to one another and joined by RNA duplexes shared between them (Fig. 1a, Supplementary Movie 1). Each sister filament represents a chain of IN octamers formed around a pair of RNA^{TAR} chains. This assembly depends on the ability of RNA^{TAR} to adopt an extended duplex, multiple copies of which rigidly tether the parallel IN filaments. Concordantly, SIVtal IN formed single filaments (chains of octamers) in the presence of single-stranded oligo-GA ribooligonucleotides, while ordered IN structures failed to form in the absence of RNA (Extended Data Fig. 3a). Although fortuitous formation of the RNA^{TAR}-tethered double IN filament enabled high-resolution structure determination, we would not expect such extended long-range arrangements to occur with the 9-kb viral RNA genome.”

We considered the double filament arrangement a fortuitous nano-crystal, which greatly helped us to refine the structure. Revised manuscript includes images of single IN filaments formed in the presence of (GA)₁₈

oligonucleotide (Extended Data Fig. 3a, middle panel), to emphasise that IN does not need to form the double filament to bind RNA (see also Extended Data Fig. 1) and that filament formation is not restricted to RNA^{TAR} bound form.

B) Same answer as to Reviewer2/Q2: There was a disconnect between our *in vitro* and *in situ* structures primarily because the signal from the RNP was very weak and fragmented in the latter. Therefore, we are excited to include a cryo-ET structure of the filament in native HIV-1 cores produced without overexpression of IN. The subtomogram average reconstructed at 12.6 Å resolution revealed substantial features not explained by CA or IN, which likely correspond to the RNP. These include density penetrating the IN filament at positions close to those of the TAR RNA observed in our *in vitro* SIVtal IN filament (Extended Data Fig. 9f). With this new data, we can be more explicit in explaining how IN filament staples vRNA to core lumen (see page 11): “Remarkably, the cryo-ET reconstruction of the filament in native HIV-1 cores revealed additional density, not explained by CA or IN, that closely matches the position of the RNATAR duplexes in complex with SIVtal IN (Extended Data Fig. 9f), strongly indicating that these positions are occupied by RNA *in vivo*. Capture of variable stem loops in viral RNA by the IN filament would staple the entire RNP to the core lumen. However, more work is required to dissect IN interactions with the entire gamut of viral RNA structures, including single stranded RNA (Extended Data Figs 1 and 3a).”

6) The authors could also do a better job of testing and explaining whether the IN filament binds to preferred site(s) on the conical capsid. It should be possible to map the individual filaments that contribute to the reconstruction back onto a model capsid and thereby reveal whether the filaments bind preferentially to the body of the cone, whether there is a preferred set of positions, etc. It might also be useful to discuss a bit more whether the authors think that the IN filaments promote the shape or assembly of the capsid.

Response:

Mapping particles back onto 2D images of cores would not have been very helpful to answer these questions. The new cryo-ET structure provides a glimpse of what is going on (Extended Data Fig. 9a, right panel). In the revised manuscript we make a carefully worded statement (page 11): “Notably, the filaments that were identified through our subtomogram classification invariably oriented along the longest dimensions of HIV-1 cores (Extended Data Fig 9a). It will be important to determine the properties of the CA lattice configuration that is preferred or perhaps enforced by the IN filament.” Only some IN filaments are clearly discernible in full tomogram reconstructions (a good example is given in Extended Data Fig. 9c), and our template matching biased subvolume picking to the particular CA curvature. More robust template matching (or perhaps template matching with a range of curved lattices) will be used in follow up studies.

Because cone-shaped core-like particles can be formed *in vitro* using CA and IP6 alone, IN would not appear to be a major factor for core assembly. Although cryo-ET analyses of class II IN mutant V165A virions and WT viruses treated with ALLINIs have revealed significant increases in the frequencies of malformed cores (Ref. 1), underlying mechanism is unclear, and we feel it would be overly speculative to suggest that the IN filament is required for proper core formation.

Minor points:

Main Text:

- 1) Line 66. “Supplementary Fig. 3b” should be changed to “Supplementary Fig. 3c” to match the figure legend and updated figure (see below).
- 2) Line 69. Similarly, “Supplementary Fig 3a” should be changed to “Supplementary Fig 3b” to match the figure legend.

Response (points 1 and 2):

Thank you. We made the supplementary figure legend consistent with the main text.

- 3) Line 72. The word “of” is missing (“in the absence or presence OF RNATar”)

Response:

OK

- 4) Line 207. F139D doesn’t seem to exhibit the >100 fold infection defect that is described (although the other mutants E69K and E69A do). F139D looks more like 5-fold.

Response:

We suspect the reviewer failed to appreciate the log₁₀ axis scale; F139D displayed 0.49% of WT HIV-Luc infectivity.

Figures and Captions:

- 1) Figure 2 caption: Should mention that this structure is SIVtal IN

Response:

OK. Done

2) Figure 2c: Residue 2 is labeled "L2", but line 105 refers to Val2. The alignment in Supplementary Figure 3 shows HIV IN with Leu2 and SIV IN with Val2. Please correct the figure to "V2".

Response:
OK. Thank you!

3) Labels for SIVtal and HIV-1 are reversed in the Supplementary Figure 1 caption, line 1094 (vs. the figure itself).

Response:
OK. Thank you!

4) Supplementary Figure 1: use the same bright green color for 125 nM IN in the figure caption as is used in the sensorgrams.

Response:
OK. Thank you!

5) Supplementary Figure 2 caption, Line 1105. Same reversal Supplementary Figure 1 caption; match the order of HIV IN and SIV IN in the figure and the figure caption (left vs right).

Response:
OK.

6) Supplementary Figure 3 caption: line 1113 legend uses terms "total" and "soluble" whereas the figure itself uses "tot" and "sup". Using "sol" in place of "sup" would make more sense.

Response:
OK. Thank you!

7) Supplementary Figure 3: the Size-exclusion chromatograph has no label but is referred to as 'b' in the legend. Then, the IN alignment panel is labeled "b" but is referred to as "c" in the figure caption.

Response:
OK. The chromatograms are now part of Extended Data Fig. 2a (bottom panel).

8) Supplementary Figure 4 and caption: Please define what the shaded area between the red and blue dotted lines represents (perhaps the red/blue average?).

Response:
In the Waters DynamX 3.0 software's butterfly difference pots, the grey area between the dotted red and blue lines represents the standard deviation of the deuterium exchange data, indicating the variability or uncertainty in the measurements across replicates and charge states for each peptide. Essentially, it visually highlights the experimental noise in the measurements. We added the following to revised legend (Extended Data Fig. 2c): "The grey area between the dotted red and blue lines represents the standard deviation of the deuterium exchange data, indicating the variability or uncertainty in the measurements across replicates and charge states for each peptide."

9) Supplementary Figure 6 caption, line 1147: "optimized" is used in the caption, but "Corrected" is used in figure. Should they match?

Response:
OK. We added "corrected" to the caption.

10) Supplementary Figure 6 caption: line 1147 refers to a "horizontal blue line" that appears black in the figure.

Response:
OK. Thank you!

11) Supplementary Figure 6 caption: lines 1153 and 1154 have the same issues as line 1147.

Response:
OK. Thank you!

12) Supplementary Figure 9 legend: line 1189 refers to "underlined residues marking the native N-terminus" but underlined residues aren't evident in the figure.

Response:

OK. Thank you! The residues are now underlined in the revised legend.

13) Supplementary Figure 10c: it would be helpful to add the numbers of particles (or percentages) that went into each 3D class (in addition to the capsid shell class).

Response:

OK, done!

Referee #1 (Remarks to the Author):

The authors have done a very good job addressing the reviewer comments. The additional data on viruses without overexpressed IN is valuable. I congratulate the authors on this impressive, landmark study.

Very minor comments:

1. Lines 187-189 comment that overlapping density weaker in single particle reconstruction: Given the excess of RNA over IN, is the oversupply of IN a reasonable explanation for this difference?

Response: This is a point for interesting discussion. Based on the original CLIP-Seq data from the landmark study by Kessl et al. (Reference #1), IN does have preferred binding sites along the viral genome. Therefore, while there is bulk excess of nucleotides, the preferred binding sites might be saturated when IN is overexpressed. We feel the original careful statement is justified: "The overlaying RNP density was noticeably weaker in our single-particle cryo-EM reconstruction, **potentially** due to the oversupply of IN in those cores (Extended Data Fig. 6b)."

2. The images in extended data 9a appear shifted relative to one another.

Response: Thank you, yes indeed the views were shifted! We readjusted the panels to match the two views. In addition, we labelled the subpanels on the Extended Figure 9a: "Tomogram slice view" and "Tomogram volume view".

3. Lines 232-233 - K170A infectivity and morphology were described by von Schwedler (ref 29), which could be recited here.

Response: Thank you, we added the reference as suggested: "CA K170A was also a severely defective virus²⁹ (Fig. 4b)."

4. If the application of single-particle image processing to the capsid core was inspired by Highland et al. 2023, and/or Stacey et al., 2023, a citation could be added.

Response: Thank you for this suggestion. We added both references (Methods, p. 32): "For structural analysis of the cores with supplemental IN, we employed single-particle cryo-EM approaches that were previously used for 3D reconstruction of the CA lattice within in vitro assembled capsid-like particles⁶⁸⁻⁷⁰."